# Improving the fine structure of intense rainfall forecast by a designed adversarial generation network

Zuliang Fang[1], Qi Zhong[1], Haoming Chen[2], Xiuming Wang[1], Zhicha Zhang[3], and Hongli Liang[1]

[1]China Meteorological Administration Training Center,Beijing 100081, China
[2]Chinese Academy of Meteorological Sciences, Beijing 10081, China
[3]Zhejiang Meteorological Observatory,Hangzhou 310017, China

**Correspondence:** Qi Zhong (zhongq@cma.gov.cn)

**Abstract.** Accurate short-term quantitative precipitation forecasting (QPF) is critical for disaster prevention, mitigation, and socio-economic activities. However, due to the inherent limitations of numerical weather prediction (NWP) models, precipitation forecasts still exhibit substantial inaccuracies. In recent years, deep learning (DL) techniques have been increasingly applied to improve precipitation forecasts, yet these approaches often produce overly smoothed outputs that fail to meet operational requirements for detail and accuracy. In this study, we propose a Generative Fusion Residual Network (GFRNet), a generative adversarial network (GAN)-based framework that integrates multi-source NWP forecasts to generate 3-hourly quantitative precipitation forecasts for North China up to 24 hours in advance. GFRNet employs an adversarial learning mechanism to enhance spatial structure reconstruction, combined with a weighted loss function and carefully designed sampling strategies to address the long-tailed distribution of precipitation and improve model training efficiency. Using independent rainy-season datasets from 2022–2024, we comprehensively evaluate the performance of GFRNet against three NWP models, a linear ensemble method (MSEM), and a deep learning baseline model (FRNet). Results show that GFRNet consistently outperforms the NWPs and baseline models across light, moderate, and heavy rainfall thresholds. Compared to the China Meteorological Administration's highest-resolution regional model (CMA-3KM), GFRNet improves Threat Scores (TS) by 4%, 28%, 35%, and 19% at the 0.1, 10, 20, and 40 mm thresholds, respectively, and improves Fractions Skill Scores (FSS) by 13%, 18%, and 15% at the 10, 20, and 40 mm thresholds. Moreover, GFRNet consistently achieves the highest Multi-Scale Structural Similarity (MS-SSIM) scores and significantly reduces RMSE, demonstrating robust spatial structure recovery, stable intensity control, and strong generalization ability. These advantages are particularly pronounced in systemic high-impact heavy rainfall events, underscoring the model's operational value. FRNet shows advantages in forecasting heavy precipitation but suffers from high BIAS and weaker generalization, limiting its practical applicability. MSEM exhibits robust performance in light and moderate precipitation scenarios but degrades significantly under extreme precipitation conditions. Overall, GFRNet dynamically fuses multi-source NWP information, balances precipitation intensity and spatial structure, and achieves higher forecasting skill and improved forecast quality across diverse precipitation regimes.

# 1 Introduction

Numerical Weather Prediction (NWP) serves as a fundamental tool in routine precipitation forecasting. However, its accuracy is constrained by various factors, including initial condition errors, limited spatial resolution, incomplete physical parameterizations, and approximate boundary conditions, all of which contribute to persistent forecast uncertainties (Sun et al., 2014; Boeing, 2016). As a result, it is challenging for any single numerical model to accurately capture the location, intensity, and structural evolution of precipitation.

In recent years, deep learning (DL), a core technique in artificial intelligence, has been increasingly applied in meteorology, including for NWP post-processing, large-scale data assimilation, super-resolution downscaling, and spatiotemporal prediction (Yang et al., 2022). In the domain of precipitation forecasting, DL has achieved significant progress. For nowcasting (0–6 hours), purely data-driven DL methods based on radar and satellite data have demonstrated substantial superiority over numerical models and optical flow methods (Shi et al., 2015; Wang et al., 2018b; Sønderby et al., 2020; Ayzel et al., 2020; Espeholt et al., 2022; Tan et al., 2024). For short-term forecasting within the 6–24 hour range, precipitation prediction primarily relies on post-processing of NWP outputs. For example, Zhang et al. (2020) developed an LSTM-based correction model for 12-hour accumulated precipitation over eastern China using ECMWF ensemble control forecasts, demonstrating superior performance for both light rain ($< 5$ mm/12 h) and heavy rain ($> 30$ mm/12 h) compared to frequency matching and SVM-based algorithms. Similarly, Chen et al. (2021) constructed an hourly precipitation correction model using a Convolutional Neural Network (CNN) applied to mesoscale forecasts from the East China Regional Numerical Center (CMA-SH9), achieving better skill than probability matching.

Moreover, Zhou et al. (2022) utilized a 3D CNN to learn the nonlinear relationship between basic meteorological variables from the ECMWF's fifth-generation reanalysis dataset (ERA5) and corresponding 3-hour accumulated precipitation. Their model, when applied to ECMWF high-resolution forecasts, significantly improved the Threat Score (TS) at the 20 mm/3 h threshold for lead times up to 72 hours. In another study, Kim et al. (2022) used basic meteorological variables and precipitation from numerical model forecasts as input features for a DL model, achieving positive correction effects for light and moderate precipitation, though the improvements diminished for precipitation exceeding 10 mm. Chen et al. (2023) employed a U-Net architecture with a weighted loss function to correct 6-hour accumulated precipitation predictions from the ECMWF, using 0.25° ERA5 precipitation data as the ground truth. This approach showed improvements across various precipitation intensities, from light rain ($\geq 0.1$ mm/6 h) to rainstorms ($\geq 20$ mm/6 h), in TS scores compared to the ECMWF forecast. Sun et al. (2023) developed a DABU-Net model combining data augmentation with deep learning to improve GFS wintertime precipitation forecasts over southeastern China. The model significantly enhanced Threat Scores (TS) across multiple thresholds, with TS at the 20 mm/day threshold increasing by up to 100% at a 72-hour lead time. Despite these advances, grid-based DL precipitation correction models generally perform better for light to moderate precipitation. Improvements in TS for heavy rainfall are often accompanied by overly smoothed predictions that lack well-defined spatial structures. Additionally, corresponding BIAS scores frequently exceed 1, indicating systematic overestimation and reducing the operational applicability of such methods.

Generative Adversarial Networks (GANs) (Goodfellow et al., 2014), as a typical deep generative model (DGM), have successfully transformed the intractable likelihood function into a neural network framework, enabling the model to optimize its parameters to fit the likelihood function. By learning through competition between a generator and a discriminator, GANs enable the production of outputs that closely resemble the distribution of real data. GANs have been widely successful in image super-resolution tasks (Wang et al., 2018a) and have shown great promise in addressing challenges in short-term forecasting, such as excessive smoothing and the degradation of intensity over time (Ravuri et al., 2021; Zhang et al., 2023). GANs have also demonstrated strong performance in statistical downscaling within the meteorological field (Leinonen et al., 2021; Price and Rasp, 2022; Singh et al., 2019). Recent studies have explored the use of GANs in post-processing NWP-based precipitation forecasts. For example, Price and Rasp (2022) utilized a 4 km resolution radar precipitation product to train a conditional GAN (CGAN) model for correcting and downscaling 6-hour precipitation forecasts from the 32 km ECMWF ensemble. The CGAN model outperformed CNN baselines and achieved skill comparable to high-resolution regional ensemble forecasts, especially for heavy precipitation events ($\geq 30$ mm/6 h). Similarly, Harris et al. (2022) aimed to generate high-resolution ensemble precipitation forecasts by post-processing ECMWF forecasts at 10 km resolution using GAN and VAE-GAN methods, targeting 1-hour accumulated precipitation products at 1 km resolution. Compared to traditional methods, the GAN approach showed significant advantages in preserving precipitation structure and predicting heavy precipitation ($\geq 5$ mm/1 h). However, most existing applications of GANs focus on probabilistic ensemble forecasts rather than deterministic quantitative precipitation forecasts, and few studies directly address severe storm precipitation, an area of critical operational importance due to the associated risks.

Short-term heavy precipitation is typically characterized by sudden onset, short duration, small spatial scale, and high localization. These features demand precipitation forecasts with finer temporal and spatial resolutions to meet operational needs. In this study, we employ a GAN-based model, GFRNet, to generate deterministic forecasts of 3-hour accumulated precipitation over the next 24 hours in North China, using multiple NWP model outputs as input and targeting a resolution of 5 km. Compared with previous research, this study introduces the following key advancements:

– GAN-based generative fusion framework. We propose GFRNet, a novel GAN-based model that dynamically integrates multi-source NWP forecasts (ECMWF, CMA-SH9, CMA-3KM), which enhances fine-scale precipitation structure reconstruction and mitigating the blurriness common in deep learning precipitation forecasts.

– Targeted evaluation of high-impact precipitation. Beyond conventional thresholds, this study adopts a stringent 40 mm/3 h criterion and introduces a Top 10% coverage-based subset to explicitly assess model performance in organized, high-impact precipitation events.

– Comprehensive multi-year validation and statistical analysis. GFRNet is systematically evaluated across three independent summer seasons (2022–2024) using diverse metrics (TS, FSS, RMSE, MS-SSIM) and paired t-tests, providing robust evidence of skill improvements and clarifying the sources of these gains.

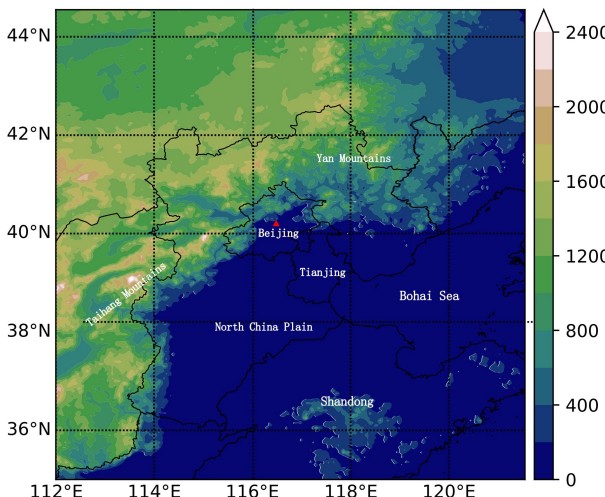

**Figure 1.** Topography distribution (shaded; in units of m) of North China domain (35° – 45°N, 112° – 122°E.) . The vast area with an altitude of less than 400m in the middle and southeast of the figure is the North China Plain, which reaches the southern foot of Yanshan Mountain in the north, leans on Taihang Mountain in the west, and borders the Bohai Sea in the east. It includes Beijing (Red Triangle), Tianjin, Shandong, and most of Hebei.

## 2 Data and method

### 2.1 Data

This study focuses on North China (35°N – 44.55°N, 112°E – 121.55°E), as illustrated in Figure 1. Administratively, this region includes Beijing, Tianjin, Hebei, Shanxi, and the Inner Mongolia Autonomous Region, with the southeastern part encompassing Shandong and the Bohai Sea region. The target area features complex topography, dominated by the Taihang Mountains, which extend from the southwest to the northeast. To the southeast lies the North China Plain, characterized by an average elevation below 400 meters. West of the Taihang Mountains is the Loess Plateau, and to the north is the Inner Mongolia Plateau, with elevations exceeding 800 meters and local peaks reaching up to 2000 meters.

This study utilizes CMA Multi-source merged Precipitation Analysis System (CMPAS) as the ground truth for precipitation fields. CMPAS is a comprehensive precipitation product developed by the National Meteorological Information Center of the China Meteorological Administration. It integrates ground automatic station data, satellite observations, and radar observations using methods such as Probability Density Function (PDF), Bayesian Model Averaging (BMA), Optimal Interpolation (OI) and Downscaling (DS)(Pan et al., 2018). CMPAS provides hourly temporal resolution and a spatial resolution of 0.05° × 0.05°.

For numerical models, considering the operational usage, model resolution, and performance, this study uses the precipitation forecast of the following three NWPs. The high-resolution global model forecast from the European Centre for Medium-Range Weather Forecasts (ECMWF), with a horizontal resolution of approximately 9 km in the China region and a temporal

resolution of 3 hours. The mesoscale forecast from the East China Regional Numerical Center (CMA-SH9) (Zhang et al., 2021), with a horizontal resolution of 9 km and a temporal resolution of 1 hour. The high-resolution regional numerical forecast independently developed by the Numerical Prediction Center of the China Meteorological Administration (CMA-3KM) (Shen et al., 2020), with a horizontal spatial resolution of about 3 km and a temporal resolution of 1 hour. Forecasts are taken from the initial times of 00 UTC and 12 UTC, retaining a 24 hours forecast range. Spatially, numerical model forecasts are interpolated to a uniform grid of $0.05° \times 0.05°$ using a bilinear interpolation algorithm, corresponding to a target area size of $192 \times 192$ grid points.

Based on the data described earlier, we performed a 3-hour accumulated precipitation (r3) forecast for the next 24 hours. Table 1 details the specific feature selection process, which includes five sources of features. Let r3(T) denote the 3-hour accumulated precipitation at time T, where the learning target is the corresponding CMPAS r3(T) observation. The input features consist of r3(T) and r3(T–3) from ECMWF, CMA-SH9, and CMA-3KM. Given that precipitation formation, development, and movement are closely linked to topography and location, META features including elevation, latitude, and longitude are also incorporated into the model. The performance of numerical model forecasts varies depending on the forecast cycle and lead time. To account for this, temporal information such as forecast cycle and lead hour is encoded using trigonometric functions and included as features in the deep learning model. The cycle values range from [0, 1], corresponding to the initial forecast times of 00 UTC and 12 UTC for the numerical models. For each cycle, only the forecast lead times at 3, 6, 9, 12, 15, 18, 21, and 24 hours are considered.

**Table 1.** Data sources and features used in the model

|       | Source   | Feature                                              |
|-------|----------|------------------------------------------------------|
|       | ECMWF    | r3(T-3), r3(T)                                        |
|       | CMA-SH9  | r3(T-3), r3(T)                                        |
| Input | CMA-3KM  | r3(T-3), r3(T)                                        |
|       | META     | Elevation, Latitude, Longitude                       |
|       | Time     | Cos(cycle), Sin(cycle), Cos(lead hour), Sin(lead hour) |
| Label | CMPAS    | r3(T)                                                 |

This study uses precipitation data from 2019 to 2022 and divides the dataset into training, validation, and test sets. In North China, precipitation is predominantly concentrated in the summer months, particularly July and August. Therefore, the period from July 10 to August 20, 2021, was designated as the validation set (637 samples) for model tuning and parameter selection, and the period from June 16 to August 31, 2022, was designated as the initial test set (1,196 samples). To further evaluate the model's generalization, we also added the rainy seasons of 2023 and 2024 (1,093 and 1,211 samples, respectively) as independent test sets.

The training set spans multiple summer periods and initially contains many non-precipitation images, which would otherwise cause the model to waste computational resources and potentially degrade its learning. Therefore, we applied an image-level sampling strategy: for each image, if the proportion of pixels exceeding a rainfall threshold $t$ is below a predefined ratio $r$, the image is discarded; otherwise, it is retained. We set $t = 1$ mm and $r = 2\%$ in this study, ensuring that low-level precipitation is captured while maintaining sufficient sample representativeness. After sampling, the training set was reduced from 4,645 to 2,885 samples. It is important to note that no sampling or filtering was applied to the validation or test sets (Table 2).

**Table 2.** Sample distribution across training, validation, and test sets. The training set (2019–2022) is subject to image-level sampling to increase the proportion of samples containing measurable precipitation. The validation set (2021) and test sets (2022–2024) were not sampled to objectively assess model performance and generalization.

| Dataset | Time Period | Samples | |
|---|---|---|---|
| | | Pre-sampling | Post-sampling |
| Training set | 2019-06-01 - 2019-10-10<br>2020-06-01 - 2020-10-10<br>2021-03-15 - 2021-07-09<br>2021-08-21 - 2021-10-10<br>2022-03-15 - 2022-06-14 | 4645 | 2885 |
| Validation set | 2021-07-10 - 2021-08-20 | 637 | No sampling |
| 2022 Rainy season | 2022-06-16 - 2022-08-31 | 1196 | No sampling |
| 2023 Rainy season | 2023-06-16 - 2023-08-31 | 1093 | No sampling |
| 2024 Rainy season | 2024-06-16 - 2024-08-31 | 1211 | No sampling |

Since the sampling was performed at the image level, we categorized precipitation intensity for each image based on the proportion of pixels exceeding specific thresholds. Specifically, an image is classified as *light rain* if more than 10% of its pixels have precipitation $\geq 0.1$ mm, as *moderate rain* if more than 0.5% of its pixels have precipitation $\geq 10$ mm, as *heavy rain* if more than 0.2% of its pixels have precipitation $\geq 20$ mm, and as a *rainstorm* if more than 0.1% of its pixels have precipitation $\geq 40$ mm.

Figure 2a illustrates how the proportion of different image-level precipitation categories changed after sampling: the proportion of **light rain** samples increased from 40% to 63%, **moderate rain** samples increased from 25% to 40%, and **heavy rain** and **rainstorm** samples rose from 16% and 7% to 26% and 11%, respectively. This adjustment in sample proportions is expected to facilitate more stable and efficient model training by increasing the representation of precipitation cases across different intensity levels.

However, Figure 2b shows the pixel-level precipitation distribution, revealing a persistent **long-tail pattern**: the proportion of pixels with precipitation $\geq 20$ mm and $\geq 40$ mm remains extremely low (only 0.29% and 0.05%, respectively). While sampling

improved the composition of training images, it cannot fundamentally change the imbalance of precipitation intensity at the pixel level. This imbalance motivated the design of the customized weighted loss function (Section 2.2.3).

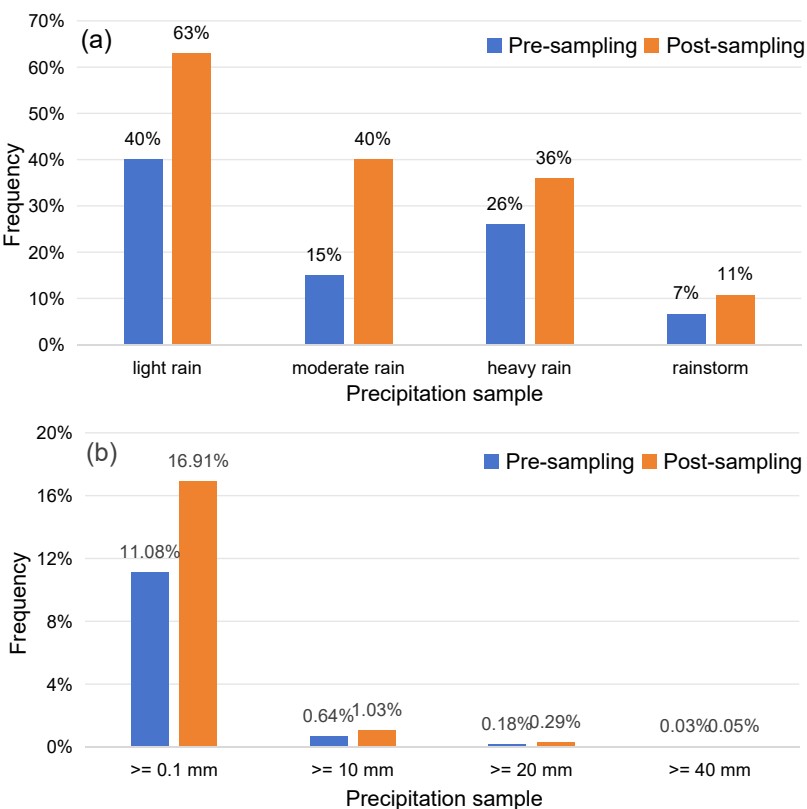

**Figure 2.** Changes in precipitation sample composition in the training set before and after sampling: (a) image-level sample proportions across different precipitation intensities; (b) pixel-level precipitation distribution, showing the persistent long-tail characteristic.

## 2.2   Methodology

### 2.2.1   GFRNet

The core idea of Generative Adversarial Networks (GANs) (Goodfellow et al., 2014) is to use adversarial training to enable
the Generator (G) to learn the distribution of real data and generate synthetic data that closely approximates real data. Simultaneously, the Discriminator (D) strives to improve its ability to distinguish between real data from the dataset and data generated by the generator. In this study, we propose a Generative Fusion Rain Net (GFRNet) for multi-NWP precipitation post-processing. As illustrated in Figure 3, GFRNet consists of two main components: the Generator and the Discriminator.

The core structure of the Generator in GFRNet is inspired by a U-Net with an encoder–decoder architecture (Ronneberger
et al., 2015). The input to the model is a tensor of size $13 \times 192 \times 192$, and the output is a tensor of size $1 \times 192 \times 192$.

The encoder comprises four Down-ConvBlocks, which gradually reduce the spatial dimensions of the feature maps while extracting deep feature information. The decoder, conversely, consists of four Up-ConvBlocks that progressively restore the spatial dimensions of the feature maps through upsampling operations. The specific sizes of the feature maps are illustrated in Figure 3a. Skip connections are introduced between the encoder and decoder, connecting the output of a layer in the encoder directly to the input of the corresponding layer in the decoder, which helps better preserve and reuse the features extracted at different levels. The activation function of the generator's final layer is set to ReLU (Agarap, 2019) for regression predictions. Each ConvBlock module integrates four key components:

– Convolution operation: This transforms the size of the feature map and is used for either upsampling or downsampling.

– Batch Normalization (BN) (Ioffe and Szegedy, 2015), ReLU, and Dropout (Srivastava et al., 2014) layers: These are used to accelerate the training process, improve model robustness, and prevent overfitting.

– Residual module (He et al., 2015): This backbone consists of two convolutional layers with BN and a dropout layer in between. The final output is obtained by adding the block input to the output of the second convolutional layer through a skip connection.

– SE-Block: This is a channel-attention module composed of two sub-modules—Squeeze and Excitation (Hu et al., 2019). The squeeze operation compresses the feature map of each channel via global pooling to obtain channel-wise importance coefficients, and the excitation operation reweights each channel according to these coefficients.

The Generator's U-Net-like structure can effectively capture the geographic and spatial dependencies of precipitation distribution. The residual structure in the ConvBlock helps prevent gradient disappearance and explosion in deep-layer networks, enhancing model performance and accelerating training. Moreover, it improves the reuse and transmission of features. The SE attention mechanism helps the model focus on the feature channels that contribute significantly to the prediction of precipitation.

Radford et al. (2016) significantly improved the training stability of GANs and the quality of generated images by introducing the Deep Convolutional GAN (DCGAN) structure. Inspired by DCGAN, the main architecture of our discriminator consists of four ConvBlocks that perform progressive spatial downsampling and channel expansion on the single input image of size $192 \times 192 \times 1$, enabling richer semantic feature extraction. This is followed by a Dense layer and a Sigmoid activation, which outputs the probability that the input image is a real sample.

### 2.2.2 Baseline Model: MSEM

Traditional multi-model fusion correction methods have undergone extensive development and validation, demonstrating widespread application and reliable performance in numerical weather prediction (DAI et al., 2018). To provide a meaningful baseline for evaluating the proposed GFRNet, we introduce the Multi-Model Similarity Ensemble Method (MSEM). This method mimics forecasters' reasoning process by quantifying the similarity among different forecasts and assigning higher weights to ensemble members that exhibit higher similarity.

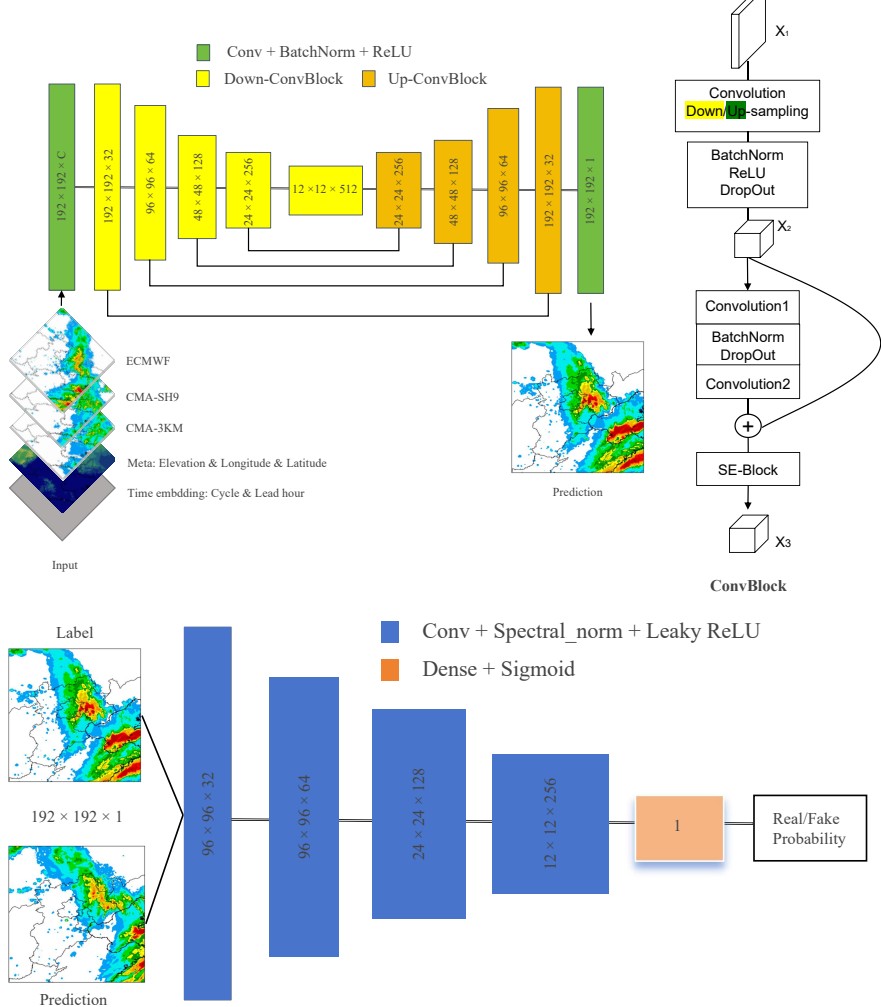

**Figure 3.** Model architecture (a)the generator of GFRNet model, also named FRNet and (b)Discriminator of GFRNet model.

Let $\hat{y}_i \in \mathbb{R}^n$ denote the flattened forecast field of the $i$-th ensemble member ($i = 1, 2, \ldots, N$), where $n$ is the total number of grid points. MSEM begins by constructing a similarity matrix $S \in \mathbb{R}^{N \times N}$ among all ensemble members using cosine similarity:

$$S_{ij} = \frac{\hat{y}_i \cdot \hat{y}_j}{\|\hat{y}_i\| \|\hat{y}_j\|} \tag{1}$$

Next, the similarity-based weight $w_i$ for the $i$-th member is computed as the average similarity between this member and all others:

$$w_i = \frac{1}{N} \sum_{j=1}^{N} S_{ij}, \quad \text{normalized as} \quad \tilde{w}_i = \frac{w_i}{\sum_{k=1}^{N} w_k} \tag{2}$$

Finally, the ensemble prediction is obtained through a weighted average:

$$\hat{y}_{\text{MSEM}} = \sum_{i=1}^{N} \tilde{w}_i \hat{y}_i \tag{3}$$

This approach adaptively emphasizes ensemble members with higher similarity, which are assumed to be more reliable. Compared to standard ensemble averaging, MSEM provides enhanced robustness and interpretability, particularly in capturing localized high-impact rainfall events, as evidenced in previous studies (Chen et al., 2005; Lin et al., 2013). It serves as a strong, physically motivated baseline for evaluating GAN-based correction methods such as GFRNet.

### 2.2.3 Model Training and Optimization

During the GAN training process, the generator and discriminator continuously compete and collaborate, driving mutual evolution. The generator aims to produce samples that resemble real data, while the discriminator receives both real and generated data as input and outputs a probability value indicating its confidence in the input being real. In this study, the optimization objectives for the discriminator and generator are as follows:

$$\min_{\theta_D} E_{y,\hat{y}} \left[ L_D \left( y, \hat{y}; \theta_D \right) \right] \tag{4}$$

$$\min_{\theta_G} E_{y,\hat{y},x} \left[ L_G \left( y, \hat{y}, x; \theta_G \right) \right] \tag{5}$$

$L_D$ and $L_G$ represent the loss functions of the discriminator and generator, respectively. The parameters of the corresponding neural networks are denoted by $\theta_D$ and $\theta_G$. The input to the generator and the predicted results are represented by $x$ and $\hat{y}$, respectively, while $y$ denotes the real labels. Wasserstein GANs (WGANs) (Arjovsky et al., 2017; Gulrajani et al., 2017) address the gradient vanishing problem commonly encountered in traditional GANs. Following the principles of WGAN, we adopt a loss function with gradient penalty term to optimize the discriminator. As shown in Equation 6, $D(y)$ and $D(G(\hat{y}|x))$ denote the scores assigned by the discriminator to real samples and samples generated by the generator, respectively. The latter part of the equation represents the gradient penalty term, where the weight $\gamma$ is set to 10, and the samples $\tilde{y}$ are randomly weighted averages of the real label $y$ and the generator's prediction $\hat{y}$, with $\varepsilon$ drawn from a randomly sampled value from a uniform distribution between 0 and 1.

$$L_D \left( y, \hat{y}; \theta_D \right) = 1 - D(y) + D(G(\hat{y} \mid x)) + \underbrace{\gamma \left( \left\| \nabla_{\tilde{x}} D(\tilde{y} \mid x) \right\|_2 - 1 \right)^2}_{gradient\ penalty} \tag{6}$$

$$\widetilde{y} = \varepsilon y + (1 - \varepsilon)G(\hat{y} \mid x) \tag{7}$$

The loss function $L_G$ for the generator consists of two components. The first part, $D(G(\hat{y} \mid x))$ is the confidence score given by the discriminator, indicating how closely the generated images resemble real samples. We aim for this score to be as high as possible. The second part, $L_{content}$ is the content loss, which is a weighted combination of Mean Squared Error (MSE) and Mean Absolute Error (MAE) loss functions. By setting the weight $\lambda$ to 50, we ensure that the values of both loss components are of the same magnitude.

$$L_G(y, \hat{y}, x, \theta_G) = 1 - D(G(\hat{y} \mid x)) + \lambda L_{\text{content}} \tag{8}$$

$$L_{\text{content}} = L_{\text{wmse}} + L_{wmae} = \sum_{i=1}^{192}\sum_{j=1}^{192} w_{i,j}(y_{i,j} - \hat{y}_{i,j})^2 + \sum_{i=1}^{192}\sum_{j=1}^{192} w_{i,j}||y_{i,j} - \hat{y}_{i,j}|| \tag{9}$$

$$w_{i,j} = \exp(a y_{i,j} + b) \tag{10}$$

In $L_{content}$, the MSE part emphasizes larger errors and provides a smoother gradient, while the MAE is less affected by outliers. Combining MSE and MAE helps balance large and small errors, enhancing the model's robustness and stability. Additionally, considering the long-tail distribution of r3 intensity as shown in Figure 2, where significant precipitation events are rare but critical, it is crucial to assign higher loss weights to samples with strong precipitation intensity (Bihlo, 2023). This strategy mitigates gradient vanishing or explosion and ensures the model learns to predict these rare, high-impact events effectively. As shown in Equation 10, we empirically found that using an exponential loss weighting function with parameters $a = 4.3$ and $b = 0.8$ yields optimal performance. To determine these values, we first tuned $a$ and $b$ on the FRNet configuration (generator-only) to balance performance across moderate-to-heavy and extreme rainfall events, selecting the combination that improved both heavy rainfall detection performance without overestimating light rain. After fixing these weights, we introduced the discriminator to form GFRNet and subsequently adjusted the gradient penalty coefficient $\gamma$ to ensure stable adversarial training, selecting a value that produced a smooth and consistent decline in the generator's loss curve.

Both the generator and discriminator are optimized using the Adam optimizer (Kingma and Ba, 2017) with betas set to (0.9, 0.999) and a weight decay of 0.01. The learning rate follows a CosineAnnealingLR schedule (Loshchilov and Hutter, 2017), oscillating between 0.001 and 0 over a period of 20 epochs. During training, we observed that the discriminator initially improved slowly, necessitating a reduction in the generator's update frequency. Experimental results showed that updating the generator every 9 steps stabilized training for both networks. Model training was monitored using the validation loss, and early stopping was employed. Training was halted if the validation loss failed to decrease for 30 consecutive epochs. All evaluation results presented below are based on the model checkpoint with the minimum validation loss.

The generator and discriminator contain 4.46M and 0.72M parameters, respectively. Training and inference were conducted using the NVIDIA CUDA library and Tesla GPUs. With a single NVIDIA A100 GPU, the training process completes in approximately 3 hours, and inference for 1,000 samples takes just 2 minutes, satisfying operational time constraints.

To further evaluate GFRNet, we conducted an ablation study using only the generator, without adversarial training, referred to as FRNet. The content loss, dataset, and training strategies for FRNet remain consistent with those used for GFRNet.

## 2.3 Evaluation Metrics

To comprehensively evaluate the predictive performance of the proposed model, we adopt a combination of categorical, neighborhood-based, and continuous/structural verification metrics. This diverse set of metrics enables an in-depth assessment from multiple perspectives, including precipitation occurrence, magnitude, and spatial structural realism.

### 2.3.1 Binary Verification Metrics

The binary metrics (TS, POD, FAR, and BIAS) are calculated based on a confusion matrix constructed at a given precipitation threshold. These metrics provide insight into the forecast's ability to correctly detect rainfall events. The specific definitions are as follows:

$$TS = \frac{h}{h+f+m}, \quad POD = \frac{h}{h+m}, \quad FAR = \frac{f}{h+f}, \quad BIAS = \frac{h+f}{h+m} \tag{11}$$

The definition of $h$, $f$, $m$ aligns with the confusion matrix shown in Table 3. The TS, POD, and FAR values range between 0 and 1. Higher TS and POD values and lower FAR values indicate better forecast performance. A BIAS value of 1 indicates an unbiased forecast, while values between 0 and 1 indicate under-prediction, and values greater than 1 indicate over-prediction.

**Table 3.** Confusion matrix to calculate metrics.True or False is determined by the chosen threshold

| Confusion matrix | | Observation | |
|---|---|---|---|
| | | True | False |
| Prediction | True | Hit(h) | False alarm(f) |
| | False | Miss(m) | True negative(tn) |

### 2.3.2 Neighborhood-Based Metric: FSS

The above binary metrics are all measured by comparing individual pixel values. Even if the predicted rainfall structure and intensity match the actual conditions, a slight positional deviation in the predicted rainfall band from the observed location can result in a high FAR and a lower POD, leading to a lower TS score, a limitation that cannot objectively reflect the true forecasting ability of the model. To address this, neighborhood spatial verification methods like the Fraction Skill Score (FSS)

(Roberts and Lean, 2008) have been developed. FSS evaluates forecast performance by comparing the fraction of grid points exceeding a certain threshold within a neighborhood in both forecast and observation fields. This approach enables a more objective assessment of high-resolution models' ability to capture spatial structures. Additionally, FSS is easy to implement and is not sensitive to parameters such as threshold filters or smoothing radii, which contributes to consistent evaluation results. FSS is now widely used and has been adopted by ECMWF as a standard metric for precipitation evaluation, replacing many traditional skill scores. The FSS is derived from the Fractional Brier Score (FBS) and is calculated as follows:

$$FBS = \frac{1}{N} \sum_{i=1}^{N} (O_r - M_r)^2 \tag{12}$$

$$FSS = 1 - \frac{FBS}{\frac{1}{N} \left( \sum_{i=1}^{N} O_r{}^2 + \sum_{i=1}^{N} M_r{}^2 \right)} \tag{13}$$

Here, $N$ is the total number of grid points within the evaluation domain, and $M_r$ and $O_r$ represent the ratio of grid points exceeding a threshold to the total number of grid points within a given window size for the forecast and observation fields, respectively. First, we use a modified Brier score to compare the precipitation frequency between forecasts and observations, known as the Fraction Brier Score (FBS). Then, employing the variance skill score concept, we derive the Fraction Skill Score (FSS), which ranges from 0 to 1, where 0 indicates no match and 1 indicates a perfect match. FSS typically increases with larger neighborhood sizes. From the definitions of FBS and BIAS, it can be observed that if the BIAS within the given window is significantly greater or less than 1, the FBS value increases, leading to a lower FSS score. This indicates that FSS penalizes both under-prediction ($BIAS < 1$) and over-prediction ($BIAS > 1$).

### 2.3.3  Continuous and Structural Metrics: RMSE and MS-SSIM

To evaluate the overall prediction error in a continuous manner, we use the Root Mean Square Error (RMSE):

$$RMSE = \sqrt{\frac{1}{N} \sum_{i=1}^{N} (\hat{y}_i - y_i)^2} \tag{14}$$

In addition, to assess the spatial structural consistency between predicted and observed precipitation, we adopt the Multi-Scale Structural Similarity Index (MS-SSIM) (Wang et al., 2003, 2004). Unlike RMSE, which only reflects pixel-wise magnitude differences, MS-SSIM evaluates perceptual similarity in luminance, contrast, and spatial structure. It is particularly suited for high-resolution precipitation forecasts. The full formulation is:

$$\text{MS-SSIM}(x,y) = \prod_{j=1}^{M} [l_j(x,y)]^{\alpha_j} \cdot [c_j(x,y)]^{\beta_j} \cdot [s_j(x,y)]^{\gamma_j} \tag{15}$$

where $x$ and $y$ are the predicted and observed fields at scale $j$ (typically $M = 5$). The three image components are defined as:

$$l(x,y) = \frac{2\mu_x\mu_y + C_1}{\mu_x^2 + \mu_y^2 + C_1}, \quad c(x,y) = \frac{2\sigma_x\sigma_y + C_2}{\sigma_x^2 + \sigma_y^2 + C_2}, \quad s(x,y) = \frac{\sigma_{xy} + C_3}{\sigma_x\sigma_y + C_3} \tag{16}$$

where $\mu_x$, $\mu_y$ are local means, $\sigma_x$, $\sigma_y$ are standard deviations, and $\sigma_{xy}$ is the covariance. Constants $C_1$, $C_2$, and $C_3$ are small values to avoid instability.

MS-SSIM values range from 0 to 1. A higher MS-SSIM value indicates stronger agreement in precipitation spatial structure and better preservation of morphology. MS-SSIM has been widely used in nowcasting and precipitation forecasting as a metric for evaluating the spatial quality of forecasts, and some studies have even applied it as a loss function to further improve model outputs (Yin et al., 2021; Tan et al., 2024). Compared to RMSE and BIAS, MS-SSIM provides a more perceptually aligned evaluation of spatial realism.

For the categorical and neighborhood-based evaluations (i.e., TS, POD, FAR, BIAS, and FSS), we uniformly apply four precipitation thresholds—0.1, 10, 20, and 40 mm per 3 hours—corresponding to light rain, moderate rain, heavy rain, and rainstorm events, respectively. These thresholds are used to assess the model's ability to detect and spatially represent different intensities of precipitation, and to ensure consistent and interpretable comparisons across all models and rainfall regimes.

## 3    Results and Analysis

The statistical evaluation results on the test set are given below.

### 3.1    Overall Performance Evaluation

Figure 4 presents a comprehensive evaluation of six models over the rainy seasons of 2022, 2023, and 2024 for 3-hour accumulated precipitation forecasts. The evaluation metrics include the Fractions Skill Score (FSS), Threat Score (TS), Probability of Detection (POD), 1–False Alarm Ratio (1–FAR), and BIAS (scaled by 0.5 for visual consistency). These metrics are computed across four precipitation thresholds (0.1, 10, 20, and 40 mm), reflecting each model's performance in terms of spatial pattern reconstruction, intensity detection accuracy, and generalization stability across rainfall regimes. For detailed metric scores of each model across the flood seasons of 2022–2024, please refer to the numerical tables provided in Appendix A.

For numerical weather prediction (NWP) models, ECMWF demonstrates overall stability, particularly at the light precipitation threshold (0.1 mm), with relatively high FSS and TS, reflecting its strength in capturing large-scale weak precipitation. However, its performance deteriorates significantly for moderate to heavy precipitation ($r3 \geq 10$ mm), where both POD and BIAS decline, indicating a systematic bias of overforecasting light rain and underforecasting heavy rainfall. CMA-SH9 exhibits a consistent overestimation tendency across all thresholds, likely due to an overly aggressive deep convection parameterization scheme. CMA-3KM benefits from higher spatial resolution, achieving better FSS and TS than the other NWPs, with relatively reasonable BIAS. Nevertheless, its skill in detecting heavy precipitation remains limited.

The MSEM model, a similarity-based ensemble constructed from multiple NWP forecasts, shows notable and stable improvements for light to moderate precipitation. Its FSS and TS at the 0.1 mm and 10 mm thresholds outperform all three NWP models. At 20 mm, MSEM maintains competitive TS and POD scores, slightly surpassing CMA-3KM, highlighting the effectiveness of its weighted integration strategy under moderate rainfall conditions. However, under the 40 mm threshold, MSEM's performance declines, constrained by the limitations of its base NWP forecasts. Its TS and FSS scores fall behind deep learning

models, especially when the underlying NWP (e.g., CMA-3KM) struggles with heavy precipitation. BIAS analysis reveals a layered behavior: MSEM tends to overpredict light precipitation (0.1 mm), stays near 1 for moderate thresholds (10–20 mm), and underestimates heavy rainfall ($r3 \geq 40$ mm), highlighting its limited responsiveness to extreme events.

    FRNet consistently achieves the highest TS and POD scores across the three years, indicating strong detection ability for moderate to heavy precipitation. However, it suffers from systematically high BIAS across all thresholds — notably, BIAS

values at 20 mm and 40 mm reached as high as 2.191 and 2.480 in 2024, respectively — indicating significant overforecasting. This issue leads to weaker FSS and 1–FAR compared to GFRNet. Its performance in 2024 deteriorates noticeably, suggesting reduced generalization across years.

    GFRNet offers the most balanced performance across all evaluation metrics. Its FSS scores remain consistently superior across all precipitation levels. While TS scores in 2022 were slightly lower than FRNet, GFRNet matched or outperformed

FRNet in 2023 and 2024. Taking 2024 as an example, GFRNet achieves TS values of 0.237, 0.173, and 0.101 for the 10 mm, 20 mm, and 40 mm thresholds, representing improvements of 22.8%, 38.4%, and 46.4% over CMA-3KM. Corresponding FSS scores are 0.570, 0.488, and 0.350, exceeding CMA-3KM by 9.6%, 30.1%, and 37.8%, respectively. GFRNet also maintains reliable intensity estimation: its BIAS scores at the 20 mm and 40 mm thresholds are 1.067 and 0.911, significantly better than FRNet and CMA-SH9, avoiding both systematic overprediction and underdetection.

GFRNet's outstanding performance is attributed to its generative adversarial framework, which introduces a discriminator to enforce distributional similarity between the predicted and observed precipitation fields. This helps improve the structural realism of rare and intense convective rainfall events. More importantly, GFRNet exhibits minimal interannual fluctuations in FSS and TS, highlighting its robust generalization across years.

    We further evaluate the temporal stability of model performance over different lead times (3–24 h) in the 2024 rainy season

as shown in 5. All models exhibit decreasing FSS and TS with increasing lead time, consistent with the accumulation of forecast errors over time. GFRNet maintains consistently high FSS and TS across all lead times, especially for moderate and heavy rainfall. Its BIAS remains within a stable and reasonable range (0.8–1.2), indicating robust intensity estimation under varying temporal conditions.

    In contrast, NWP models (ECMWF, CMA-SH9, CMA-3KM) exhibit rapid performance degradation at longer lead times,

especially for higher thresholds. FRNet performs well in TS for short lead times but consistently shows high BIAS ($> 1.5$). As a result, its FSS scores under the 20 mm and 40 mm thresholds lag behind MSEM, and even fall below CMA-3KM during some lead hours (e.g., 15–21 h), suggesting limited structural reconstruction capability. MSEM retains advantages under light to moderate thresholds (0.1–20 mm) but is clearly outperformed at 40 mm, where it falls behind CMA-3KM, again reflecting its limited skill in extreme rainfall scenarios.

GFRNet stands out with consistently superior TS and FSS across precipitation thresholds and lead times, along with robust BIAS control. It exhibits the best overall generalization. FRNet, while strong in detection (TS), suffers from high BIAS and limited spatial accuracy. MSEM remains effective in moderate rainfall but lacks responsiveness in extreme cases.

**Table 4.** The RMSE and MS-SSIM of ECMWF, CMA-SH9, CMA-3KM, MSEM, FRNet, and GFRNet for 3-hourly precipitation predictions over 2022–2024 rainy seasons. The best, second-best, and third-best scores for each metric are shown in bold, underlined, and italic, respectively.

| Model | 2022 | | 2023 | | 2024 | |
|---|---|---|---|---|---|---|
| | RMSE | MS-SSIM | RMSE | MS-SSIM | RMSE | MS-SSIM |
| ECMWF | **2.208** | 0.653 | **1.803** | 0.693 | **2.693** | 0.607 |
| CMA-SH9 | 3.049 | *0.717* | 2.737 | *0.747* | 3.945 | *0.687* |
| CMA-3KM | 2.826 | 0.714 | 2.358 | 0.745 | 3.652 | 0.662 |
| MSEM | 2.221 | 0.642 | *1.909* | 0.678 | 2.821 | 0.593 |
| FRNet | 2.459 | 0.754 | 2.083 | 0.784 | 3.322 | 0.714 |
| GFRNet | *2.264* | **0.763** | 1.883 | **0.794** | *2.857* | **0.728** |

     To further evaluate each model's overall ability to capture precipitation intensity and spatial structure, 4 summarizes the RMSE and multi-scale structural similarity index (MS-SSIM) scores for all models during the rainy seasons of 2022–2024.

ECMWF consistently achieves the lowest RMSE in 2022–2023, reflecting reliable average intensity forecasts in weak precipitation regimes. However, its MS-SSIM remains low (maximum of 0.693), indicating spatial structure mismatches under convective conditions. CMA-SH9 shows high RMSE and limited MS-SSIM improvements, consistent with its overestimation tendency and parameterization biases. CMA-3KM benefits from resolution (MS-SSIM is 0.754 in 2022) but suffers from poor RMSE in 2024 (3.652), suggesting accumulated errors in convective scenarios.

MSEM maintains second-tier RMSE across years, indicating robust average intensity control. However, its MS-SSIM consistently ranks low, suggesting poor spatial structure reconstruction. FRNet achieves strong MS-SSIM (0.754–0.784), reflecting its ability to represent convective-scale structures, but suffers from a high RMSE, consistent with its high BIAS and overforecasting. GFRNet demonstrates the best balance: top MS-SSIM scores in all three years and competitive RMSE (2022–2024: 2.264–2.857), validating its ability to capture both spatial structure and precipitation magnitude, with strong generalization

capability.

### 3.2   Spatial Performance Analysis

To further evaluate the spatial performance of different models, Figure 6 presents the spatial distribution of FSS scores for each model during the 2024 rainy season across four precipitation thresholds (0.1 mm, 10 mm, 20 mm, and 40 mm). Figure 7 shows the FSS improvement of GFRNet relative to the other reference models.

At the light precipitation threshold of 0.1 mm, all models exhibit relatively high FSS values with spatially uniform distributions and minor inter-model differences. ECMWF shows slightly lower scores over the Taihang Mountains, while the CMA

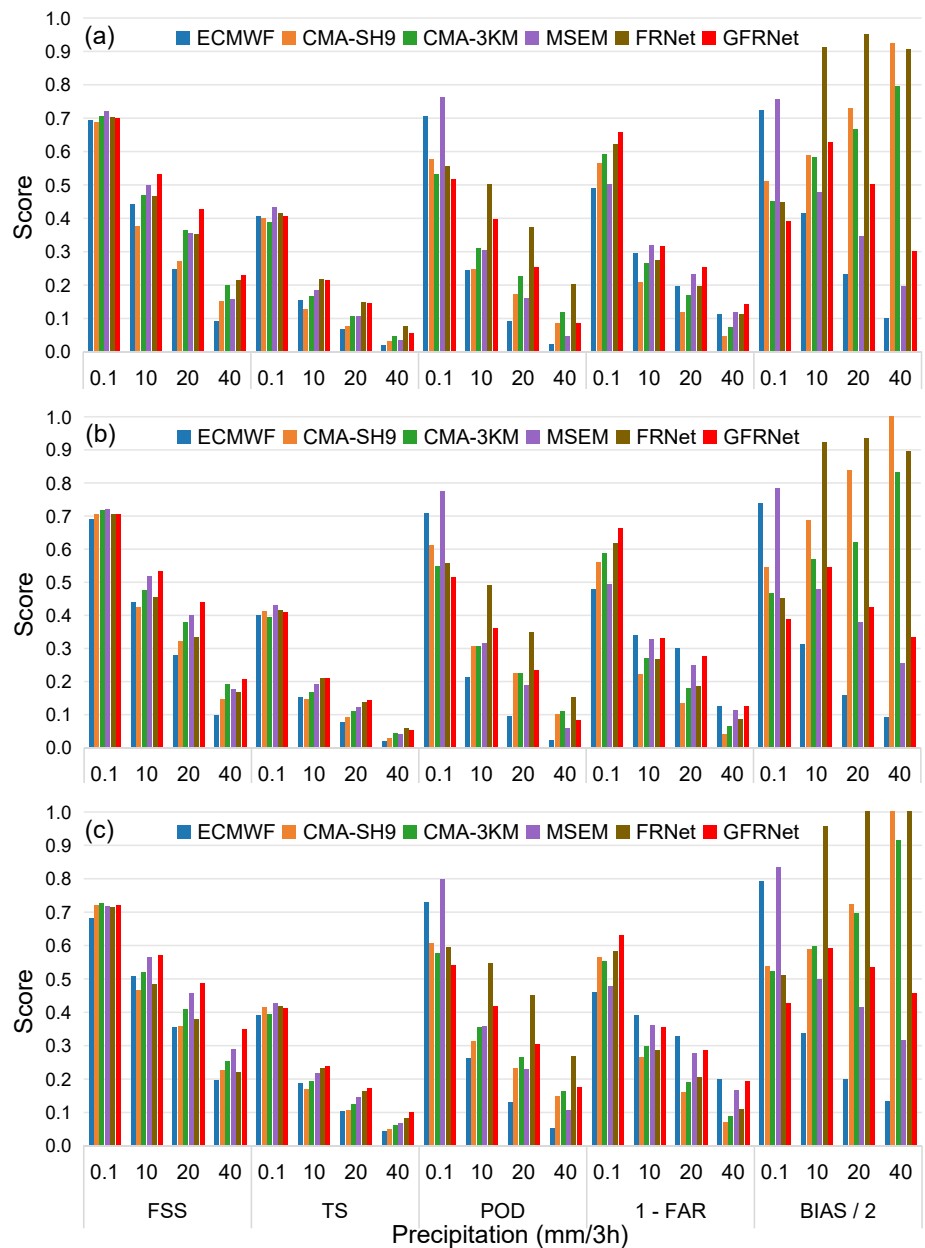

**Figure 4.** Evaluation scores of different models during the (a) 2022, (b) 2023, and (c) 2024 rainy seasons for 3-hour accumulated precipitation forecasts. Metrics include the Fraction Skill Score (FSS), Threat Score (TS), Probability of Detection (POD), False Alarm Ratio (shown as $1 - \text{FAR}$), and Bias score (scaled by 0.5 for visualization consistency). Evaluations are conducted at multiple precipitation thresholds (0.1, 10, 20, and 40 mm/3h) across six models: ECMWF, CMA-SH9, CMA-3KM, MSEM, FRNet, and GFRNet.

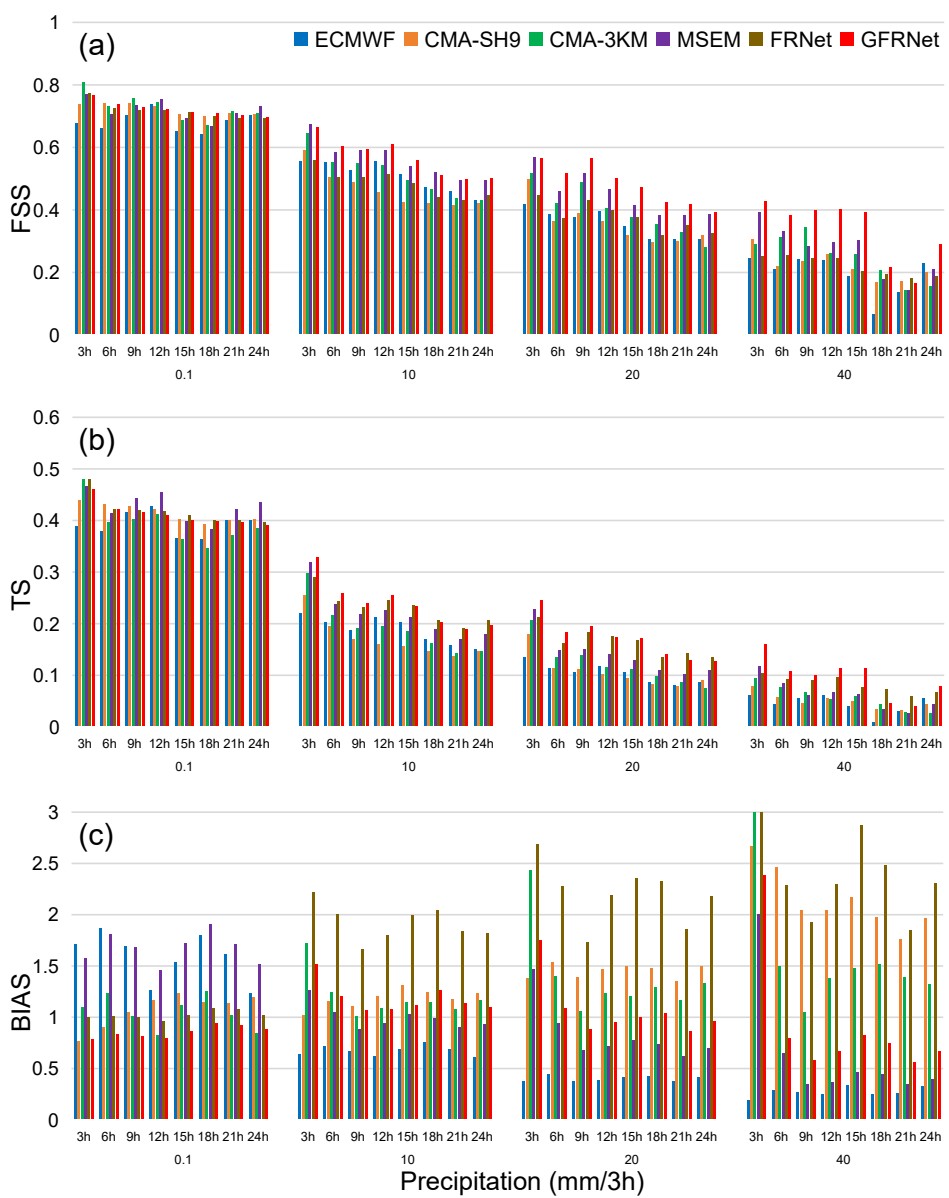

**Figure 5.** Temporal evolution of (a) FSS, (b) TS, and (c) BIAS scores across different precipitation thresholds (0.1, 10, 20, and 40 mm/3h) during the 2024 rainy season. Results are shown for six models at 3–24 h lead times (3 h interval). GFRNet consistently maintains high FSS and TS with relatively stable BIAS across precipitation intensities and forecast ranges.

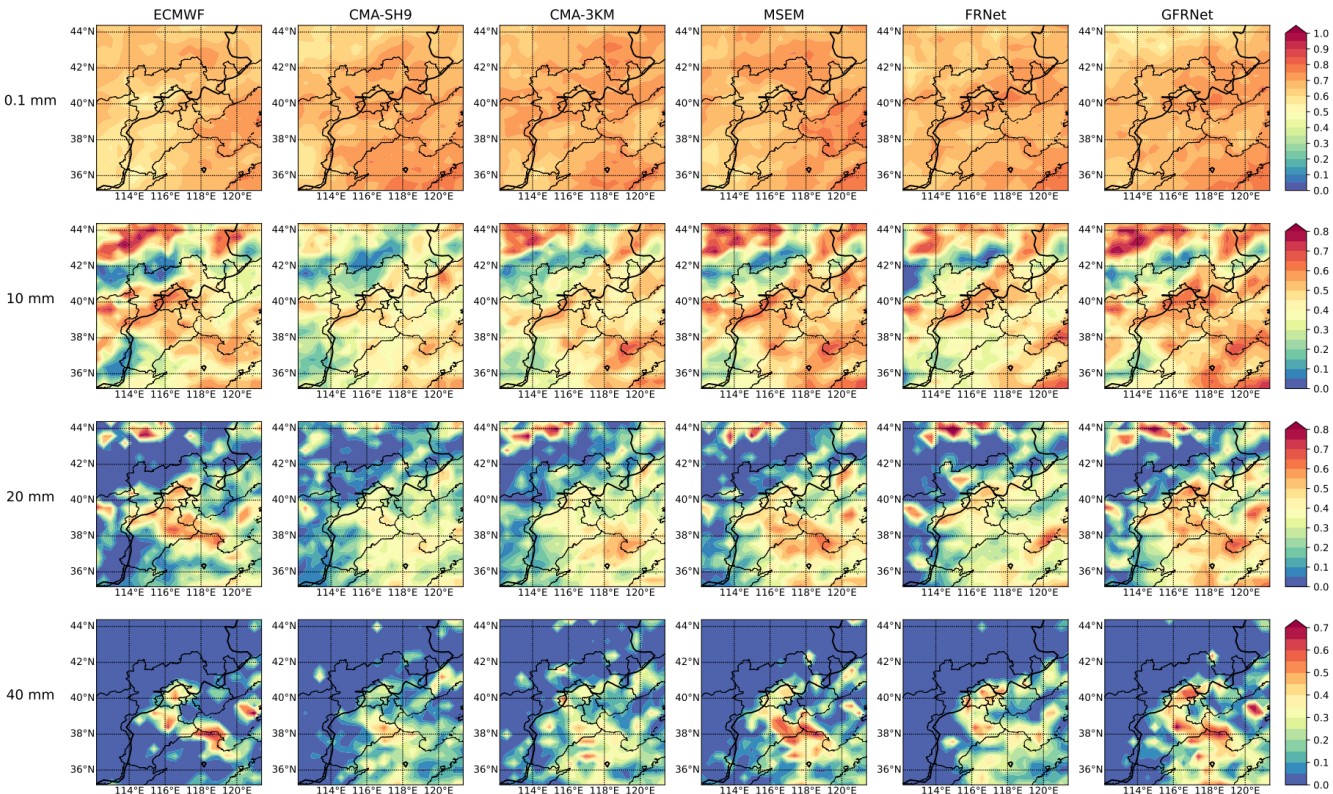

**Figure 6.** Spatial distribution of Fraction Skill Score (FSS) during the 2024 rainy season across four precipitation thresholds (0.1, 10, 20, and 40 mm/3h, from top to bottom). The six columns correspond to ECMWF, CMA–SH9, CMA–3KM, MSEM, FRNet, and GFRNet. Higher FSS values indicate better spatial consistency between predictions and observations. The black lines show 500-meter elevation contours.

series and deep learning models (FRNet and GFRNet) perform more stably, indicating that all models effectively capture light rainfall patterns.

As the threshold increases to 10 mm (moderate rain), spatial performance differences among models become more pro-
380 nounced. In general, NWP models achieve higher FSS scores in plains and coastal regions than in mountainous areas, high-lighting the challenge of modeling precipitation over complex terrain. ECMWF maintains competitive performance in the Taihang Mountain region, and CMA-3KM and CMA-SH9 show stable performance over the North China Plain and Shandong Peninsula. They also perform well over the gently varying topography in the northeastern highlands (upper-right region of the figures).

At the 20 mm threshold (heavy rain), spatial differentiation becomes more significant. ECMWF retains relatively high scores in central regions but degrades elsewhere. CMA-SH9 and CMA-3KM continue to show stable performance in the cen-tral–eastern regions, reflecting their ability to capture mesoscale precipitation structures. In contrast, MSEM and deep learning

models begin to demonstrate advantages, effectively fusing multi-source forecast information to enhance the representation of localized heavy precipitation.

Under the 40 mm threshold (rainstorm), the spatial coverage of high FSS scores from NWP models shrinks noticeably, and the well-performing regions become sparse. Overall FSS values drop significantly, indicating the persistent challenges these models face in forecasting extreme rainfall events. In contrast, GFRNet maintains relatively high scores across multiple key regions, particularly in southern Hebei, western Shandong, and the eastern foothills of Shanxi, suggesting robust spatial generalization and capability for modeling high-impact events.

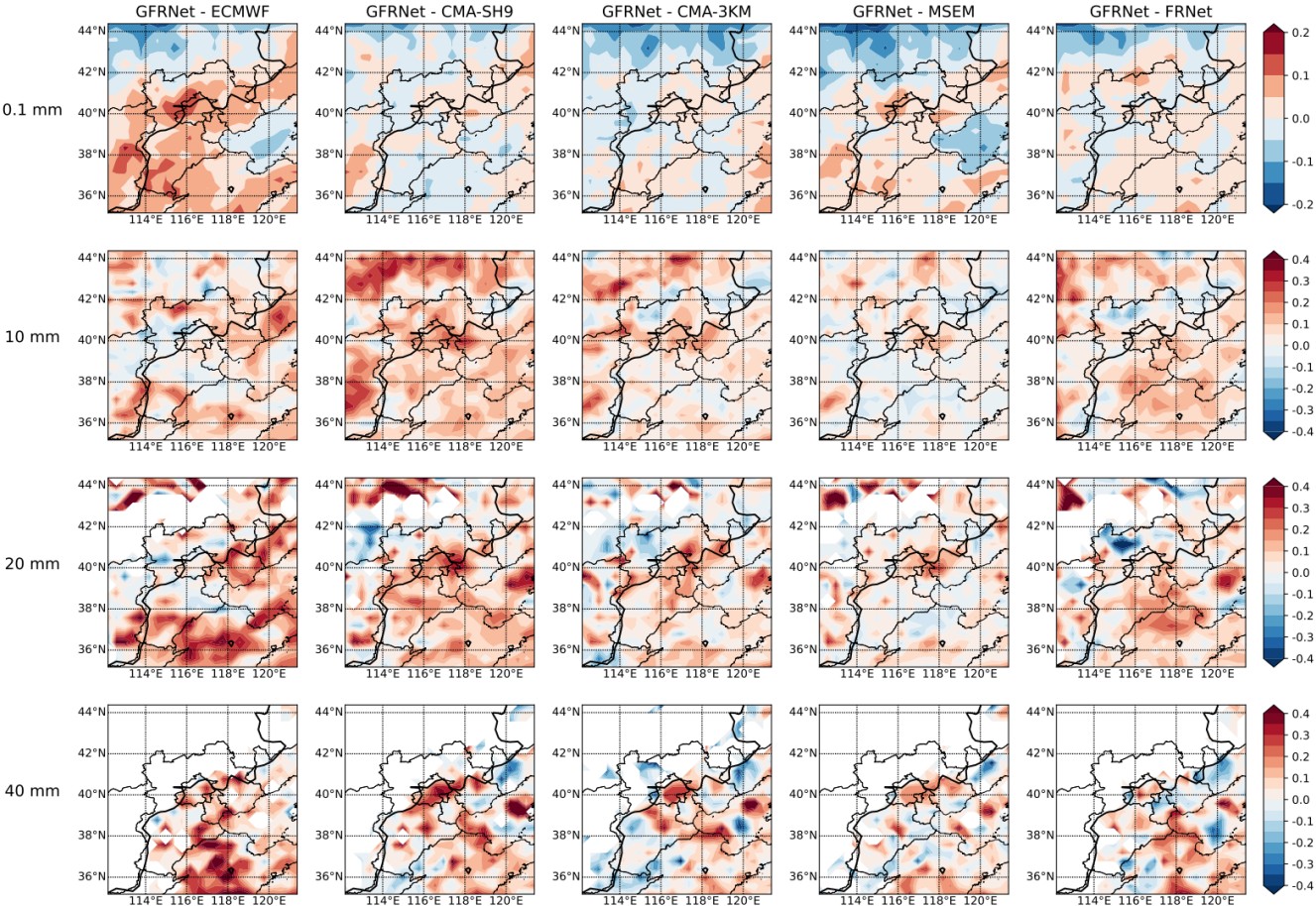

**Figure 7.** Spatial distribution of FSS improvement by GFRNet over other models during the 2024 rainy season. Each column represents the FSS difference between GFRNet and a baseline model (from left to right: ECMWF, CMA-SH9, CMA-3KM, MSEM, and FRNet). Each row corresponds to a rainfall threshold (0.1 mm, 10 mm, 20 mm, and 40 mm per 3h, from top to bottom). Red areas indicate regions where GFRNet outperforms the corresponding model, while blue areas indicate regions where it underperforms. The black lines show 500-meter elevation contours.

From the perspective of multi-model integration, MSEM, FRNet, and GFRNet all demonstrate the ability to leverage NWP guidance at moderate to heavy rainfall levels. However, MSEM tends to be more conservative as precipitation intensity increases, and FRNet shows diminished learning capability under extreme events. GFRNet, by contrast, consistently exhibits superior spatial adaptability and integration capability across all rainfall thresholds, with notably higher FSS scores at 10 mm, 20 mm, and 40 mm compared to other models.

The FSS improvement maps in Figure 7 further illustrate the spatial regions where GFRNet improves over the reference models. At thresholds above 10 mm, GFRNet shows substantial positive gains (highlighted in red) across key areas such as most of Shandong, eastern and central Hebei, the eastern foothills of the Taihang Mountains, and gently elevated plateau regions. Some of these areas are known to be particularly challenging for NWP models due to their complex precipitation structures and less reliable forecasts. Notably, even in regions where ECMWF and the CMA models already perform well,

GFRNet still provides consistent gains, underscoring its robustness and adaptability under diverse geographical conditions.

### 3.3 Case Studies of Heavy Rainfall Events

### 3.3.1 Heavy Rainfall Event on 5 July 2022

On 5 July 2022, a significant precipitation event impacted southern Hebei and western Shandong in North China. This event was associated with the interaction between a weakening tropical cyclone (the remnants of Typhoon Chaba) and an upper-level

trough. The rainfall exhibited both convective and stratiform characteristics and was distributed across a broad region, posing substantial challenges for accurate forecasting, particularly regarding the initiation and development of convective systems.

  To comprehensively evaluate model performance, we analyzed the 3-hourly accumulated precipitation forecasts from +3 h to +24 h lead times (Figure 8) and compared the results using standard verification metrics (Figure 9)

  In the early forecast stages (from +3 h to +12 h), observations indicated scattered convective rainfall over southwestern

Shandong and southern Hebei. These localized convective cells gradually evolved into a narrow southwest–northeast-oriented rainband. GFRNet effectively capture the core locations and general evolution of the precipitation at this early stage. In contrast, ECMWF and MSEM significantly underestimated both the intensity and spatial extent of the rainfall. CMA-SH9 and CMA-3KM produced reasonable forecasts but exhibited slight spatial deviations in the initial precipitation patterns.

  From +15 h onward, the rainfall band intensified and propagated northeastward. By +21 h and +24 h, it split into two distinct

clusters, forming a clear double-center structure. Most models captured this structural evolution to varying degrees.ECMWF exhibited a delayed response to moderate and heavy rainfall, failing to forecast the northern rain cluster but reasonably predicting southern rainfall at later lead times (e.g., +18 h to +24 h), albeit with weakened intensity. CMA-SH9 and CMA-3KM effectively reproduced the spatial distribution and evolution of the two rainfall centers, although both models showed positional shifts relative to observations. MSEM provided spatially smoothed forecasts with moderate accuracy but relatively low TS and

FSS scores. FRNet precisely located the southern rainband but tended to over-smooth rainfall structures, leading to higher false alarm rates and limited generalization at higher thresholds. GFRNet consistently captured both the spatial structure and inten-

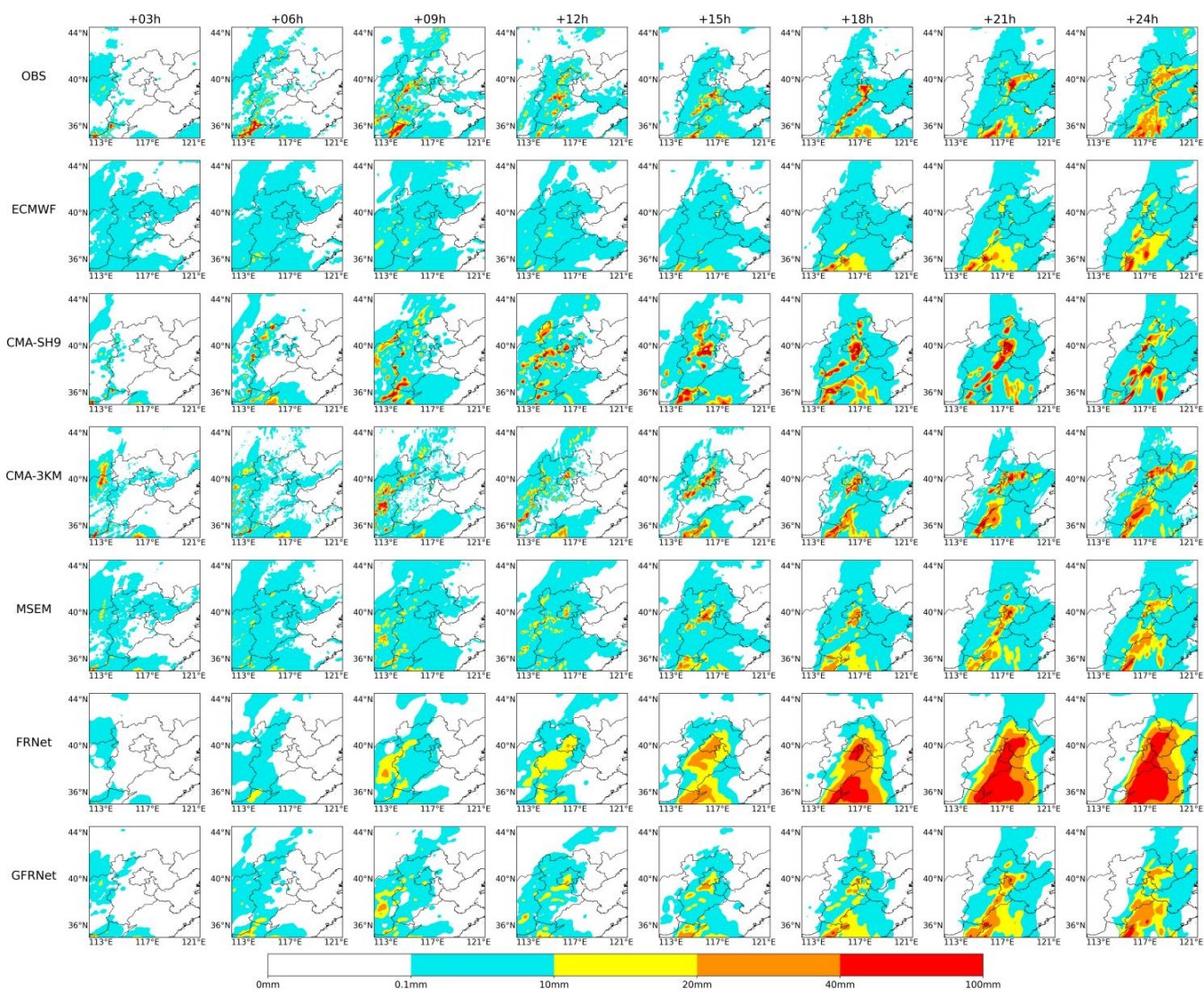

**Figure 8.** Precipitation forecasts of all models initialized at 0000 UTC on 5 July 2022. Panels show 3-hour accumulated precipitation at +3 h to +24 h lead times from observations (CMPAS) and six forecast models (ECMWF, CMA-SH9, CMA-3KM, MSEM, FRNet, and GFRNet). This event was associated with a weakening extratropical cyclone and an upper-level trough over North China, resulting in a widespread heavy rainfall event affecting parts of Shandong and Hebei provinces.

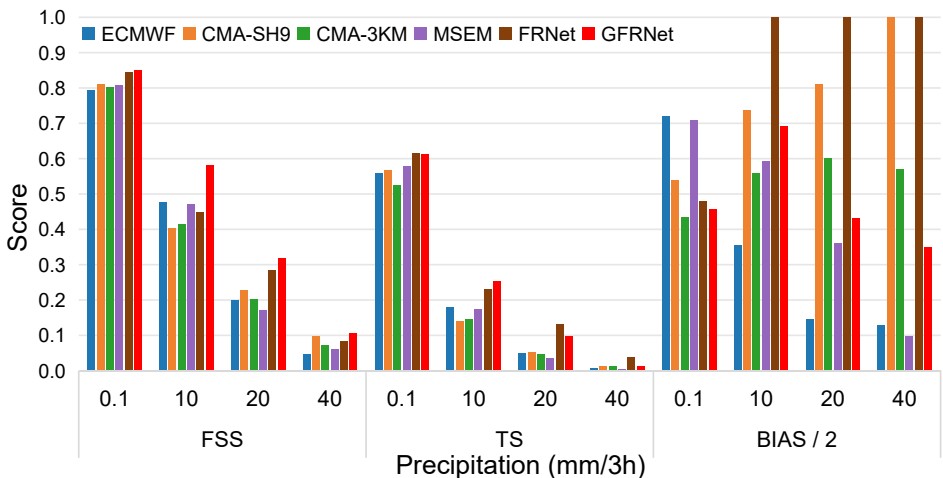

**Figure 9.** Verification scores of all models for the precipitation event on 5 July 2022, evaluated over four thresholds (0.1, 10, 20, and 40 mm per 3 h). Metrics include the Fractions Skill Score (FSS), Threat Score (TS), and Bias Score (BIAS). To maintain visual comparability across different metrics, BIAS values are scaled by a factor of 0.5 (i.e., BIAS/2 is shown). For BIAS, a value closer to 1 indicates better performance

sity evolution, accurately reproducing the development of the double-center rainfall pattern and achieving improved spatial fidelity.

It is important to note that GFRNet's forecasting capability is built upon the input guidance of three NWP models. For instance, the ECMWF forecast provided a reasonably accurate depiction of the southern rainfall location after +18 h, but with substantially weaker intensity. In contrast, CMA-SH9 and CMA-3KM both provided relatively better predictions of intensity but with moderate spatial shifts. GFRNet effectively leveraged the complementary strengths of these models by dynamically learning and integrating both spatial and intensity-related features. This fusion mechanism allowed GFRNet to provide a more accurate and balanced forecast, particularly for the evolving rainband structures at later lead times.

The verification scores presented in Figure 9 further support these findings. GFRNet achieved the highest TS and FSS values across all precipitation thresholds, particularly at 20 mm and 40 mm per 3 h, indicating its advantage in forecasting moderate to heavy rainfall. Its BIAS values remained close to 1 (noting that BIAS/2 is shown in the figure), suggesting balanced precipitation intensity forecasts. In contrast, CMA-SH9 and FRNet showed a stronger tendency toward overestimation at higher thresholds, while ECMWF consistently underestimated precipitation.

Overall, this case study demonstrates that GFRNet is capable of accurately forecasting both the spatial distribution and intensity of precipitation in a challenging convective environment. This capability stems from its dynamic assimilation and integration of multi-source NWP information, yielding improvements over traditional NWP models and baseline deep learning approaches.

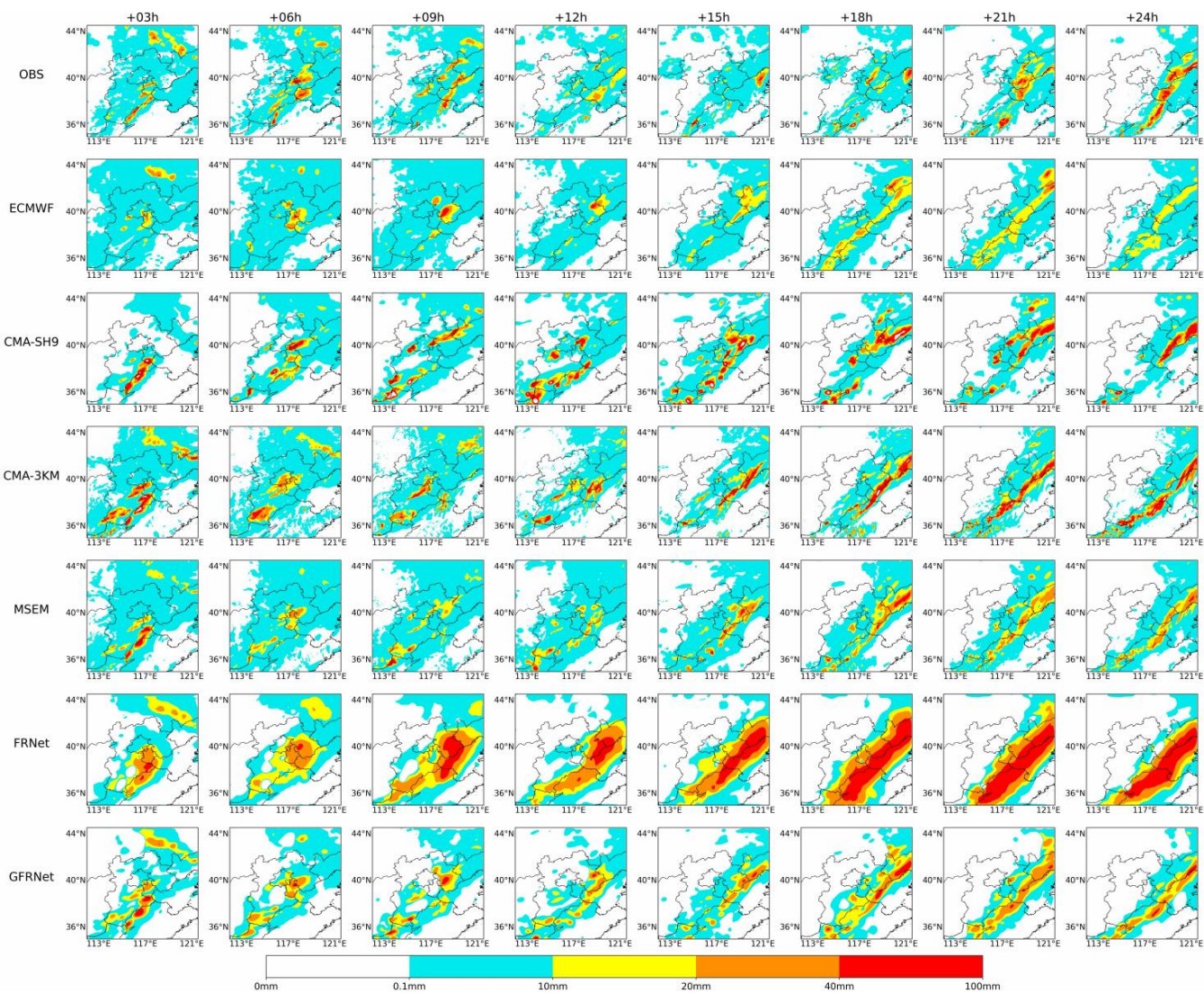

**Figure 10.** Precipitation forecasts of all models initialized at 0000 UTC on 25 July 2024. Panels show 3-hour accumulated precipitation at +3 h to +24 h lead times from observations (CMPAS) and six forecast models (ECMWF, CMA-SH9, CMA-3KM, MSEM, FRNet, and GFRNet).

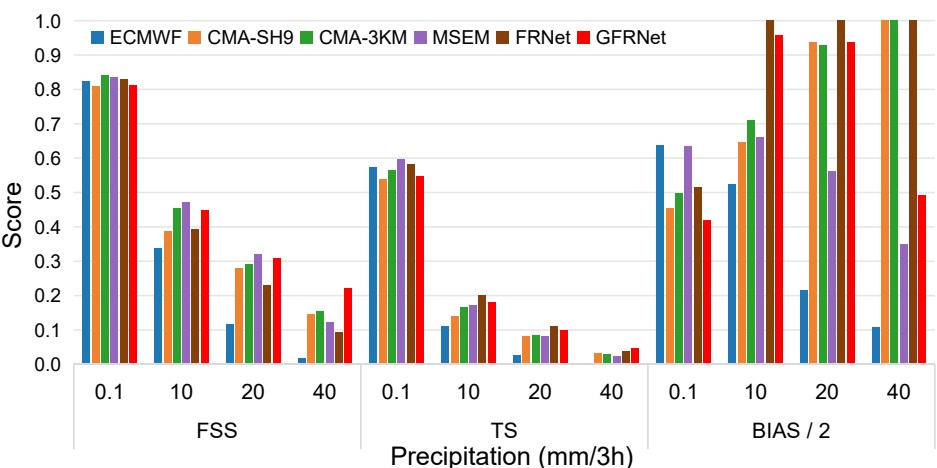

**Figure 11.** Verification scores of all models for the precipitation event on 25 July 2024, evaluated over four thresholds (0.1, 10, 20, and 40 mm per 3 h). Metrics include the Fractions Skill Score (FSS), Threat Score (TS), and Bias Score (BIAS). To maintain visual comparability across different metrics, BIAS values are scaled by a factor of 0.5 (i.e., BIAS/2 is shown). For BIAS, a value closer to 1 indicates better performance

### 3.3.2 Organized Rainstorm over Beijing-Tianjin-Hebei Region on 25 July 2024

This case focuses on a typical summer heavy precipitation event that occurred between 00:00 UTC on 24 July and 00:00 UTC on 25 July 2024, significantly impacting the Beijing–Tianjin–Hebei region. The event was characterized by high organization, abrupt onset, and extremity. The formation of this heavy rainfall process was driven by a combination of favorable large-scale conditions: persistent control of the subtropical high, continuous moisture transport from the outer circulation of Typhoon *Gaemi*, the eastward progression of a mid-latitude trough, the presence of a low-level shear line, and orographic lifting as-

sociated with the *Taihang* and *Yanshan* Mountains. The 5880 *gpm* ridge of the subtropical high remained quasi-stationary over northern China, facilitating sustained moisture accumulation and convective instability. Meanwhile, southeasterly flow at 850 *hPa*, with specific humidity values reaching 16–18 g kg$^{-1}$, provided abundant water vapor from the East China Sea. The superposition of the mid-level trough, low-level convergence, and orographic forcing contributed to the rapid development and structural organization of the convection.

The evolution of the precipitation system can be broadly divided into two stages: the early stage (+03 h to +12 h) was dominated by scattered deep convection, while the later stage (+18 h to +24 h) transitioned into a well-organized, banded precipitation system. Observations show that by +03 h, multiple localized heavy rainfall centers emerged in southeastern Hebei and northeastern parts of the domain. By +06 h, two convective cells developed in eastern Hebei, which further organized into two southwest–northeast (SW–NE) oriented narrow rainbands by +09 h. At +12 h, the western band weakened, and the eastern

band moved offshore. By +15 h, the heavy rainfall temporarily ceased. Subsequently, from +18 h to +24 h, new convective cells developed rapidly over northeastern Hebei, Tianjin, and western Shandong. By +24 h, these cells merged to form a prominent

SW–NE oriented rainband spanning multiple provinces, illustrating the high degree of organization and rapid evolution of this event (Fig. 10).

During this process, numerical weather prediction (NWP) models exhibited stage-dependent performance. In the early phase characterized by scattered convection, ECMWF forecasts generally underestimated precipitation intensity and failed to capture convective development. In contrast, CMA-SH9 and CMA-3KM tended to overestimate rainfall intensity and exhibited substantial spatial biases, with premature development and false alarms in certain regions. In the later stage with more organized precipitation, all three NWP models predicted the emergence of the SW–NE oriented rainband as early as +15 h, while in reality the structure was not observed until after +21 h. This indicates a common "premature triggering" issue in system-scale precipitation forecasting. Among them, ECMWF provided more accurate spatial placement but underestimated intensity, while CMA-SH9 and CMA-3KM captured stronger precipitation but suffered from high bias and spatial overextension.

The MSEM method, based on weighted ensemble integration of three NWP models using inter-model similarity, demonstrated robust performance in light to moderate precipitation scenarios. It achieved the highest FSS scores at the 0.1 mm, 10 mm, and 20 mm thresholds (Fig. 11), highlighting the advantage of ensemble averaging in mitigating individual model biases under moderate conditions. However, due to the absence of structural correction and nonlinear representation capabilities, MSEM significantly underestimated extreme rainfall, with notably lower TS and FSS scores at the 40 mm threshold, revealing limited generalization to high-impact events.

FRNet outperformed most NWPs in terms of TS scores, yet it exhibited a clear tendency toward systematic overprediction. Strong and extensive rainfall belts emerged as early as +06 h and +09 h, with further intensification during +21 h and +24 h. These features led to a substantial positive bias, indicating excessive spatial coverage and rainfall intensity. While FRNet enhanced structural representation, it lacked sufficient physical constraints on extreme rainfall, making it prone to overfitting under severe weather conditions.

GFRNet, by dynamically integrating ECMWF's strength in spatial placement with CMA-type models' responsiveness to heavy rainfall, achieved more balanced and physically realistic forecasts. It accurately captured the dual-band structure at +06 h and +09 h, and its eastern rainband at +12 h closely matched observations. Although residual spurious rainfall persisted at +15 h, it was markedly weaker than in other models. During the +21 h to +24 h period, GFRNet effectively reconstructed the newly formed main rainband, both in structure and intensity, without the inflated patterns observed in FRNet. As shown in Fig. 11, GFRNet achieved the highest TS and FSS scores at the 40 mm threshold, while maintaining a near-unity BIAS value, validating its generalization capability and forecast stability in extreme rainfall scenarios. However, similar to the NWPs, GFRNet also exhibited a tendency to predict the emergence of organized rainbands prematurely (around +15 h), indicating that its temporal modeling still inherits timing biases from the input NWP forecasts and thus requires further refinement.

This case study highlights the key characteristics, strengths, and limitations of the different models under a complex extreme rainfall scenario, and demonstrates the enhanced structural and intensity prediction capabilities of GFRNet under a multi-source fusion framework.

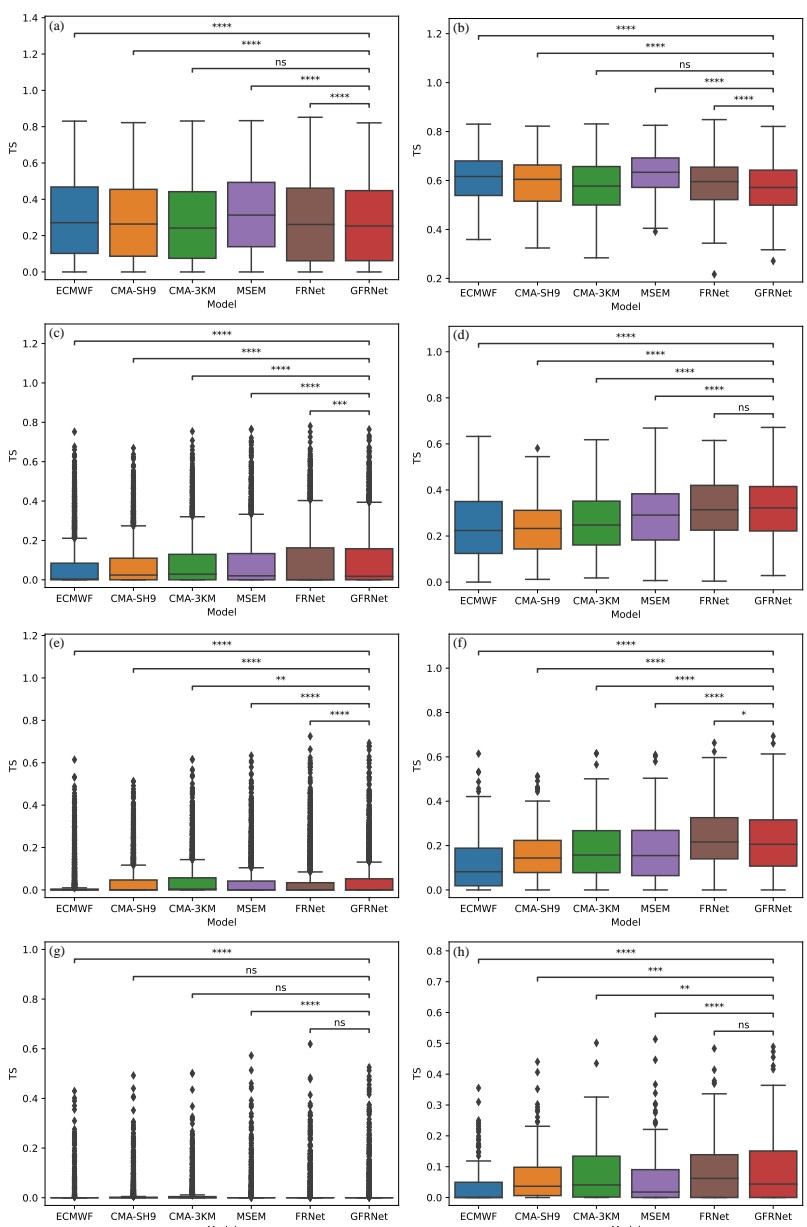

**Figure 12.** Boxplots of TS distributions for all models at four precipitation thresholds (0.1, 10, 20, and 40 mm). Panels (a)(c)(e)(g) show results for the *all-sample set*, including all samples containing at least one pixel above the given threshold. Panels (b)(d)(f)(h) present results for the *Top 10% coverage subset*, defined as the top 10% of samples ranked by the number of pixels exceeding the threshold, representing organized high-impact rainfall events. Stars indicate significance levels from paired t-tests between models (∗: $p < 0.05$, ∗∗: $p < 0.01$, ∗∗∗: $p < 0.001$). Compared to NWPs and MSEM, GFRNet shows statistically significant advantages in the Top 10% subset for 20–40 mm thresholds, while differences are less pronounced for light precipitation cases.

## 3.4 Statistical Significance Testing and Sample Stratification Analysis

In the overall statistical evaluation of the verification metrics, the deep learning models and MSEM consistently outperformed the numerical models across multiple precipitation thresholds. However, to further verify whether these performance differences are statistically significant and to clarify their primary sources, we conducted a statistical significance analysis. In precipitation forecast evaluation, aggregate scores are often dominated by a large number of weak-precipitation samples. These weak samples typically consist of only scattered or marginal precipitation pixels; although numerous, they contribute little to disaster prevention and mitigation and can therefore "dilute" the models' demonstrated performance in key precipitation events (e.g., organized rainbands and frontal rainfall systems).

To more precisely assess model skill in high-impact precipitation scenarios, we stratified the analysis into two levels:

– **All-sample set**: Includes all samples with at least one pixel exceeding the specified precipitation threshold.

– **Top 10% coverage subset**: Within the all-sample set, samples were ranked by the number of pixels exceeding the threshold, and the top 10% were selected. These samples represent cases with the largest precipitation coverage and clearest signals at each threshold, providing a focused evaluation of model performance in major precipitation events.

For the 0.1, 10, 20, and 40 mm thresholds, the all-sample set contained 3,471, 2,812, 2,435, and 1,813 samples, respectively, with corresponding Top 10% subsets of 347, 281, 243, and 181 samples. Paired t-tests were applied to the scores within each sample set to assess statistical significance between models, and significance levels ($p < 0.05, 0.01, 0.001$) were annotated in the boxplots with star markers.

Figure 12 presents the TS (Threat Score) distributions for the models at thresholds of 0.1, 10, 20, and 40 mm, for both the all-sample and Top 10% subsets, along with the corresponding significance annotations. At the **light precipitation** level ($\geq 0.1$ mm), MSEM showed the most stable performance, with TS scores clearly higher than most models, followed by ECMWF. FRNet and GFRNet exhibited no obvious advantage at this level, suggesting that their generalization in weak-precipitation contexts remains limited.

For moderate and heavy rainfall, GFRNet already demonstrated superior performance over the NWP models and MSEM in the all-sample set, and this advantage became even more pronounced in the Top 10% subset. FRNet also showed significant gains in the Top 10% subset, but for heavy rainfall in the all-sample set, its TS median and upper quartile were slightly lower than those of CMA-3KM, indicating that its higher TS scores largely stem from a small number of high-impact samples.

At the extreme rainfall threshold ($\geq 40$ mm), all models performed poorly for dispersed strong-precipitation samples. CMA-3KM, benefiting from its higher resolution, performed slightly better than others in the all-sample set. However, in the Top 10% subset (representing typical organized extreme rainfall events), FRNet and GFRNet achieved median and upper-quartile TS values significantly higher than those of the CMA models and MSEM, underscoring their advantage in organized extreme rainfall situations.

From this analysis, several important insights emerge. Firstly, in comparing the three NWP models, we find that for moderate and heavy rainfall events, CMA-3KM consistently demonstrates stable and superior performance across both the all-sample and

Top 10% subsets, underscoring the robustness and advantages of high-resolution mesoscale modeling, with CMA-SH9 ranking second. In contrast, ECMWF shows markedly weaker forecast skill for rainfall above 20 mm, highlighting the limitations of the global model in these scenarios. Secondly, deep learning models show a clear advantage for organized extreme rainfall. GFRNet's strong performance in the Top 10% subset suggests that its generative fusion strategy is particularly effective for complex precipitation processes, such as typhoon rainbands and Mei-yu fronts, leading to substantial improvements in these high-impact scenarios. However, dispersed heavy rainfall remains challenging. In diverse, complex background conditions, FRNet has not demonstrated consistent advantages, and GFRNet also shows weaknesses in handling weak-signal samples, even in extreme rainfall cases ($\geq 40$ mm). This suggests that future work should target weak-precipitation scenarios with dedicated training strategies or loss function designs, to prevent the model from becoming overly "aggressive" or neglecting weak signals.

Furthermore, mean scores can obscure differentiated capabilities. Model evaluation should not rely solely on overall scores; it is essential to incorporate analyses of the Top 10% subset to avoid average statistics masking a model's true strengths in high-impact events. Differentiating between the all-sample and Top 10% subsets helps diagnose the model's "core capability." Finally, future development should consider incorporating additional thermodynamic and dynamic predictors to enable the model to better characterize precipitation generation when NWP guidance is weak or fails, thereby enhancing its generalization in weak-precipitation scenarios.

## 4    Discussion and Conclusions

This study developed the GFRNet model, a generative adversarial network (GAN)-based framework designed to produce 3-hourly quantitative precipitation forecasts (QPFs) for northern China up to 24 hours ahead. GFRNet ingests forecasts from one global model (ECMWF) and two regional models (CMA-SH9 and CMA-3KM) as inputs, and employs a tailored sampling strategy alongside a weighted loss function to improve model training efficiency and address precipitation's long-tailed distribution. We systematically evaluated GFRNet's performance for the 2022, 2023, and 2024 summer rainy seasons, comparing it against three NWP models (ECMWF, CMA-SH9, CMA-3KM), a similarity-based ensemble approach (MSEM), and a non-generative deep learning baseline (FRNet). Across three independent rainy seasons, GFRNet demonstrated consistently superior spatial structure reconstruction and robust intensity control, achieving higher TS, FSS, and MS-SSIM scores and lower RMSE than other approaches, highlighting its strong generalization capability and operational applicability. FRNet exhibited better detection of heavy precipitation but suffered from high BIAS and weaker generalization, while MSEM performed well in moderate rainfall but deteriorated in extreme precipitation conditions.

The comparative analyses highlight phase-dependent strengths and weaknesses of the NWP models. For scattered, convective precipitation, all models exhibited notable spatial displacement and intensity biases; the higher-resolution CMA-3KM captured localized convection more effectively but remained prone to false triggers and overestimation. In contrast, during organized rainfall events, NWPs captured banded structures relatively well, yet often predicted them too early or too late and with biased intensity. Among the NWP models, CMA-3KM consistently delivered the most reliable performance across both scat-

tered and organized rainfall, demonstrating the value of high-resolution regional models; CMA-SH9 followed, while ECMWF underperformed for events exceeding 20 mm but retained relatively accurate positional guidance due to its large-scale control fields. Compared to the NWPs, GFRNet delivered improvements across moderate to heavy rainfall events, with particularly clear gains for organized high-impact rainfall, demonstrating that its generative fusion mechanism is especially well suited for complex systems (e.g., typhoon outer rainbands and Mei-yu fronts). However, its performance in non-organized, highly localized heavy rainfall remains an area for improvement.

The sources of model skill provide further insights. MSEM, a similarity-weighted ensemble method, showed stable performance in the 0.1–20 mm range but declined sharply for $\geq 40$ mm events due to its lack of structural correction and nonlinear representation, limiting its generalization to extreme precipitation. FRNet, trained with content-based loss optimization, primarily focused on systematic rainfall events, often adopting an "aggressive fusion" strategy that over-predicted rainfall to maximize detection. While this yielded higher POD and TS scores, it resulted in systematic overforecasting (elevated BIAS) and distorted precipitation structures, and gave insufficient attention to rare, localized heavy rainfall. GFRNet, by introducing adversarial training, reinforced learning of realistic spatial structures and dynamically fused complementary strengths of multiple NWPs, reducing false precipitation while retaining fine-scale structure. This contributed to its stronger spatial fidelity and generalization, which remained stable even on independent 2023–2024 rainy season data. Spatial FSS maps further confirmed this: GFRNet consistently improved FSS both in areas where NWPs struggled and where they already performed well.

Despite these advances, several limitations and future research directions emerge. First, the evaluation metric system warrants refinement. While the Fraction Skill Score (FSS) better reflects spatial displacement and structure errors than TS, it can overreward overly smooth precipitation fields, potentially distorting assessments. A multidimensional metric framework integrating pixel-wise accuracy with structural fidelity would provide more robust evaluations. Furthermore, aggregate metrics can mask model-specific strengths and weaknesses: averaged scores risk "diluting" performance in high-impact rainfall scenarios. Evaluations should explicitly distinguish between all samples and subsets representing the most influential rainfall events to better diagnose model "core competence."

Second, the current loss functions remain pixel-level MSE and MAE variants. Even with weighting schemes, they tend to neglect weak, isolated rainfall signals during training, leading models to either over-aggressively forecast or under-represent weak signals. Tailored loss functions and training strategies are required to address this gap.

Third, while GFRNet nonlinearly integrates multi-NWP information, its performance is ultimately constrained by NWP guidance. In cases where none of the three NWP models capture precipitation, GFRNet similarly fails to reconstruct realistic structures, illustrating that current deep learning post-processing is still largely dependent on the underlying NWPs. To overcome this limitation, future steps include introducing more thermodynamic and dynamic variables (e.g., temperature, humidity, wind fields, geopotential height) as auxiliary inputs, enabling the model to directly learn the complex nonlinear relationships between physical factors and precipitation generation, thereby enhancing its capabilities in forecasting nascent convection and systematic organizational structures.

Finally, limitations inherent to GAN-based frameworks merit attention. GAN training can suffer from mode collapse, instability, and difficulty learning rare-event distributions, which are critical for extreme rainfall. Future research could leverage

diffusion models—next-generation generative frameworks that have demonstrated superior stability and distribution learning in image reconstruction and remote sensing. Conditional diffusion models, which iteratively "denoise" toward realistic outputs under NWP constraints, could gradually generate refined precipitation fields and naturally support probabilistic outputs, enabling uncertainty quantification. Hybrid GAN–diffusion architectures may balance GAN's efficiency with diffusion's stability, improving realism without compromising speed.

It is worth noting that, although MSEM provides a strong and interpretable baseline for multi-model fusion, this study has not yet carried out a systematic comparison against a broader set of classical statistical or regression-based methods (e.g., locally weighted regression, analogue techniques, or topography-aware interpolation schemes). A more comprehensive benchmark including such lower-complexity and more transparent approaches would further clarify the practical added value of GAN-based post-processing. We regard this as an important direction for future work, particularly in the context of operational implementation and user-facing interpretability.

In summary, GFRNet illustrates the potential of generative modeling for precipitation correction, delivering marked gains for organized and high-impact rainfall events. Future work combining more informative physical predictors (e.g., thermodynamic and dynamic fields), advanced generative architectures (e.g., conditional diffusion), and probabilistic output frameworks offers a clear path to further advance GFRNet's capabilities, enhancing its ability to forecast a wider range of rainfall types and intensities with greater fidelity.

*Code and data availability.* The gridded precipitation ground truth data and model forecast outputs used in this study are freely accessible at https://doi.org/10.57760/sciencedb.09821 (Zuliang and Qi, 2024). The codes for training GFRNet and FRNet, as well as for evaluating model performance, are available at https://zenodo.org/records/14652556 (Fang and Zhong, 2025).

**Appendix A: Performance Tables for Rainy Seasons (2022–2024)**

This appendix provides detailed evaluation scores for model performance during the 2022, 2023, and 2024 rainy seasons. The tables present verification results for 3-hour precipitation forecasts using a set of categorical and neighborhood-based metrics: Threat Score (TS), Bias Score (BIAS), False Alarm Ratio (FAR), Probability of Detection (POD), and Fraction Skill Score (FSS). Evaluations are performed at four thresholds: 0.1, 10, 20, and 40 mm/3 h, corresponding to light, moderate, heavy, and extreme rainfall.For each metric, the best and second-best scores are shown in bold and underlined text, respectively.

**Appendix B: Ablation Study**

**B1   Training Process Ablation Analysis**

Ablation experiments were conducted to assess the effects of two architectural components in GFRNet: the Squeeze-and-Excitation (SE) block and the weighted loss function. Specifically, the performance of GFRNet was compared with that of a

**Table A1. 2022 Rainy Season:** Evaluation results of **ECMWF**, **CMA–SH9**, **CMA–3KM**, **MSEM**, **FRNet**, and **GFRNet** for $r3$ prediction. TS, BIAS, FAR, POD, and FSS are listed. For each metric, the **best**, second-best, and *third-best* scores are highlighted using bold, underline, and italic font styles, respectively.

| Precipitation | Model | FSS | TS | BIAS | FAR | POD |
|---|---|---|---|---|---|---|
| $r3 \geq 0.1$ mm | ECMWF | 0.693 | 0.405 | 1.444 | 0.511 | 0.706 |
| | CMA–SH9 | 0.689 | 0.400 | **1.024** | 0.436 | *0.578* |
| | CMA–3KM | 0.704 | 0.389 | 0.899 | *0.409* | 0.532 |
| | MSEM | **0.720** | **0.434** | 1.514 | 0.497 | **0.761** |
| | FRNet | *0.702* | 0.416 | *0.896* | 0.379 | 0.557 |
| | GFRNet | 0.700 | *0.406* | 0.784 | **0.343** | 0.515 |
| $r3 \geq 10$ mm | ECMWF | 0.443 | 0.155 | 0.829 | *0.704* | 0.245 |
| | CMA–SH9 | 0.376 | 0.128 | 1.774 | 0.790 | 0.246 |
| | CMA–3KM | *0.469* | 0.167 | 1.167 | 0.730 | *0.310* |
| | MSEM | 0.499 | *0.184* | **0.958** | **0.682** | 0.305 |
| | FRNet | 0.465 | **0.216** | 1.822 | 0.725 | **0.501** |
| | GFRNet | **0.530** | 0.214 | *1.254* | 0.683 | 0.398 |
| $r3 \geq 20$ mm | ECMWF | 0.248 | 0.066 | 0.466 | 0.805 | 0.091 |
| | CMA–SH9 | 0.270 | 0.075 | 1.458 | 0.882 | 0.171 |
| | CMA–3KM | 0.363 | *0.108* | **1.333** | 0.830 | *0.227* |
| | MSEM | *0.356* | 0.105 | *0.692* | 0.768 | 0.161 |
| | FRNet | 0.352 | **0.147** | 1.901 | *0.804* | **0.373** |
| | GFRNet | **0.427** | 0.145 | 1.006 | **0.748** | 0.254 |
| $r3 \geq 40$ mm | ECMWF | 0.092 | 0.019 | 0.200 | *0.887* | 0.023 |
| | CMA–SH9 | 0.154 | 0.031 | 1.850 | 0.953 | *0.086* |
| | CMA–3KM | *0.198* | *0.047* | **1.588** | 0.926 | 0.117 |
| | MSEM | 0.157 | 0.035 | *0.394* | 0.881 | 0.047 |
| | FRNet | 0.215 | **0.077** | 1.810 | 0.889 | **0.201** |
| | GFRNet | **0.228** | 0.056 | 0.603 | 0.858 | 0.085 |

**Table A2. 2023 Rainy Season:** Evaluation results of **ECMWF**, **CMA–SH9**, **CMA–3KM**, **MSEM**, **FRNet**, and **GFRNet** for $r3$ prediction. FSS, TS, BIAS, FAR, and POD are listed. For each metric, the **best**, second-best, and *third-best* scores are highlighted using bold, underline, and italic font styles, respectively.

| Precipitation | Model | FSS | TS | BIAS | FAR | POD |
|---|---|---|---|---|---|---|
| $r3 \geq 0.1$ mm | ECMWF | 0.689 | 0.401 | 1.478 | 0.520 | 0.710 |
| | CMA–SH9 | 0.706 | *0.412* | **1.092** | 0.441 | *0.611* |
| | CMA–3KM | 0.718 | 0.395 | 0.933 | *0.413* | 0.548 |
| | MSEM | **0.720** | **0.432** | 1.571 | 0.507 | **0.775** |
| | FRNet | 0.706 | 0.415 | 0.896 | 0.381 | 0.558 |
| | GFRNet | *0.707* | 0.409 | 0.777 | **0.336** | 0.516 |
| $r3 \geq 10$ mm | ECMWF | 0.439 | 0.151 | **0.628** | **0.660** | 0.214 |
| | CMA–SH9 | 0.424 | 0.148 | 1.377 | 0.778 | 0.306 |
| | CMA–3KM | *0.476* | 0.167 | *1.139* | 0.731 | 0.306 |
| | MSEM | 0.519 | *0.192* | **0.959** | *0.671* | *0.315* |
| | FRNet | 0.454 | 0.209 | 1.843 | 0.733 | **0.492** |
| | GFRNet | **0.533** | **0.209** | 1.088 | 0.668 | 0.361 |
| $r3 \geq 20$ mm | ECMWF | 0.278 | 0.077 | 0.315 | **0.701** | 0.094 |
| | CMA–SH9 | 0.322 | 0.092 | 1.675 | 0.865 | *0.226* |
| | CMA–3KM | *0.378* | 0.111 | 1.243 | 0.819 | 0.225 |
| | MSEM | 0.399 | *0.121* | *0.749* | *0.749* | 0.190 |
| | FRNet | 0.334 | 0.138 | 1.871 | 0.814 | **0.348** |
| | GFRNet | **0.439** | **0.145** | **0.847** | 0.724 | 0.234 |
| $r3 \geq 40$ mm | ECMWF | 0.098 | 0.020 | 0.184 | 0.874 | 0.023 |
| | CMA–SH9 | 0.146 | 0.030 | 2.550 | 0.960 | *0.102* |
| | CMA–3KM | 0.193 | *0.043* | *1.662* | 0.934 | 0.110 |
| | MSEM | *0.177* | 0.040 | 0.511 | *0.888* | 0.057 |
| | FRNet | 0.167 | **0.058** | 1.789 | 0.914 | **0.153** |
| | GFRNet | **0.207** | 0.053 | **0.665** | **0.873** | 0.084 |

**Table A3. 2024 Rainy Season:** Evaluation results of **ECMWF**, **CMA–SH9**, **CMA–3KM**, **MSEM**, **FRNet**, and **GFRNet** for $r3$ prediction. *FSS*, TS, BIAS, FAR, and POD are listed. For each metric, the **best**, second-best, and *third-best* scores are highlighted using bold, underline, and italic font styles, respectively.

| Precipitation | Model | FSS | TS | BIAS | FAR | POD |
|---|---|---|---|---|---|---|
| $r3 \geq 0.1$ mm | ECMWF | 0.682 | 0.392 | 1.587 | 0.541 | 0.729 |
| | CMA–SH9 | *0.720* | *0.414* | *1.075* | *0.435* | *0.607* |
| | CMA–3KM | **0.727** | 0.392 | 1.045 | 0.449 | 0.576 |
| | MSEM | 0.719 | **0.425** | 1.670 | 0.523 | **0.797** |
| | FRNet | 0.715 | 0.417 | **1.021** | 0.418 | 0.594 |
| | GFRNet | 0.720 | 0.410 | 0.856 | **0.369** | 0.540 |
| $r3 \geq 10$ mm | ECMWF | 0.508 | 0.187 | **0.672** | **0.609** | 0.263 |
| | CMA–SH9 | 0.464 | 0.168 | 1.179 | 0.734 | 0.313 |
| | CMA–3KM | *0.520* | 0.193 | 1.194 | 0.703 | 0.354 |
| | MSEM | 0.564 | *0.218* | **0.995** | 0.640 | *0.358* |
| | FRNet | 0.485 | 0.231 | 1.914 | 0.715 | **0.547** |
| | GFRNet | **0.570** | **0.237** | *1.182* | *0.646* | 0.419 |
| $r3 \geq 20$ mm | ECMWF | 0.355 | 0.104 | 0.400 | **0.671** | 0.131 |
| | CMA–SH9 | 0.392 | 0.105 | 1.446 | 0.839 | 0.233 |
| | CMA–3KM | *0.408* | 0.125 | *1.393* | 0.810 | *0.265* |
| | MSEM | 0.458 | *0.144* | 0.829 | *0.722* | **0.230** |
| | FRNet | 0.378 | 0.165 | 2.191 | 0.794 | **0.451** |
| | GFRNet | **0.488** | **0.173** | **1.067** | 0.714 | 0.305 |
| $r3 \geq 40$ mm | ECMWF | 0.195 | 0.044 | 0.268 | **0.800** | 0.054 |
| | CMA–SH9 | 0.226 | 0.050 | 2.132 | 0.930 | 0.149 |
| | CMA–3KM | *0.254* | 0.062 | *1.832* | 0.910 | *0.164* |
| | MSEM | 0.290 | *0.069* | 0.635 | *0.834* | 0.105 |
| | FRNet | 0.221 | 0.083 | 2.480 | 0.882 | **0.268** |
| | GFRNet | **0.350** | **0.101** | **0.911** | 0.807 | 0.176 |

variant without SE blocks (GFRNet_wo_SE) and another using standard MSE/MAE loss instead of the weighted loss (GFR-Net_wo_WeightedLoss). The results are summarised in Table B1.

The SE blocks were found to have limited influence on light rain prediction, as reflected by the comparable TS scores of GFRNet and GFRNet_wo_SE (0.406 vs. 0.408). However, for thresholds of 10 mm and above, GFRNet consistently outperformed the variant without SE blocks. For instance, at the 20 mm and 40 mm thresholds, the TS scores increased from 0.134

and 0.052 to 0.145 and 0.056, respectively. These results suggest that SE blocks play a notable role in capturing the structural details associated with heavier precipitation. In contrast, the use of standard loss functions led to improved TS scores for light rain (0.431 vs. 0.406), indicating better performance in this regime. Nevertheless, for higher thresholds, the weighted loss function significantly enhanced model accuracy. At the 20 mm and 40 mm thresholds, the TS scores of GFRNet_wo_WeightedLoss dropped to 0.115 and 0.028, compared to 0.145 and 0.056 for GFRNet. This demonstrates the effectiveness of the weighted

loss in improving the model's sensitivity to moderate and heavy rainfall.

**Table B1.** TS scores for different rain thresholds in blocks ablation experiments.Note: The best score is indicated in bold and the second-best score is underlined

| Model/Threshold | 0.1 mm | 10 mm | 20 mm | 40 mm |
|---|---|---|---|---|
| GFRNet | 0.406 | **0.214** | **0.145** | **0.056** |
| GFRNet_wo_SE | 0.408 | 0.195 | 0.134 | 0.052 |
| GFRNet_wo_WeightedLoss | **0.431** | 0.191 | 0.115 | 0.028 |

In summary, both SE blocks and the weighted loss function are essential to GFRNet's performance in forecasting moderate to heavy precipitation. The SE blocks enhance spatial feature representation, while the weighted loss strengthens the model's focus on high-impact events. These findings confirm the utility of the proposed components in improving the robustness and accuracy of precipitation forecasts.

## B2  Input Source Contribution Analysis

We conducted a series of ablation experiments to systematically evaluate the contribution of each input source to the precipitation forecasting performance of GFRNet. The influence of each input was quantified using a Relative Importance Score (RIS), which is defined as follows:

$$\text{Relative Importance Score}(x) = \frac{\text{TS(GFRNet)} - \text{TS(GFRNet\_wo\_x)}}{\text{TS(GFRNet)}} \tag{B1}$$

The TS scores of the ablation experiments are presented in Table B2. The results indicate that, even with the removal of any single input, GFRNet consistently outperforms the three NWP baselines in forecasting moderate, heavy, and storm precipitation. This demonstrates the model's robustness and its capacity to produce reliable corrections even when certain data sources are unavailable.

**Table B2.** TS scores for different rain thresholds in source ablation experiments

| Model/Threshold | 0.1 mm | 10 mm | 20 mm | 40 mm |
|---|---|---|---|---|
| wo_ECMWF | 0.397 | 0.186 | 0.127 | 0.053 |
| wo_CMA-SH9 | 0.406 | 0.201 | 0.137 | 0.055 |
| wo_CMA-3KM | 0.405 | 0.183 | 0.124 | 0.051 |
| wo_META | **0.418** | 0.203 | 0.134 | 0.049 |
| wo_Time | 0.410 | 0.198 | 0.126 | 0.050 |
| GFRNet | 0.406 | **0.214** | **0.145** | **0.056** |

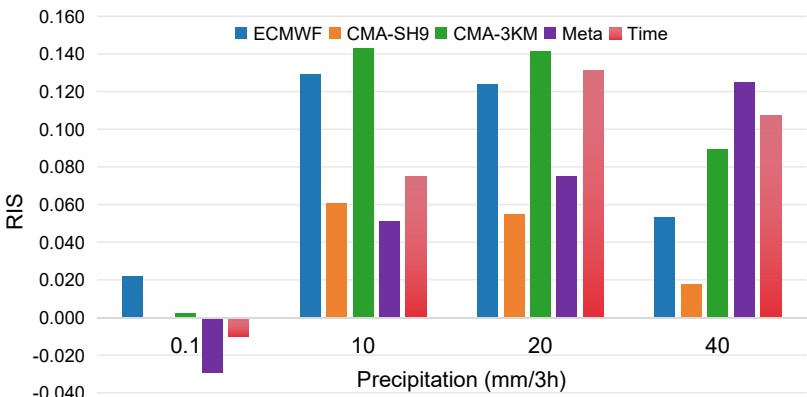

**Figure B1.** Relative importance score (RIS) of each input source (ECMWF, CMA-SH9, CMA-3KM, Meta, and Time) at different precipitation thresholds (0.1, 10, 20, and 40 mm/3h). RIS values quantify the contribution of each source to GFRNet's performance; positive values indicate a beneficial effect on model forecasts, while negative values indicate a slight degradation.

Further analysis of the RIS values (Figure B1) shows that, except for META and temporal features—which exhibit a minor
negative effect on light rain forecasts—all inputs contribute positively across precipitation categories. The ECMWF input is particularly beneficial for moderate and heavy rainfall, although its contribution is smaller in light and storm precipitation forecasts. This is consistent with its status as a global high-resolution model with advanced physical parameterizations (e.g., cloud microphysics and boundary-layer schemes), which enhance its skill in simulating mesoscale precipitation processes.

The CMA-3KM input yields substantial improvements across moderate, heavy, and storm precipitation forecasts, with par-
655 ticularly strong impact on moderate and heavy rain. As a high-resolution regional model, CMA-3KM is capable of resolving finer-scale convective structures and local precipitation evolution, thereby enhancing forecast accuracy in these regimes. In contrast, CMA-SH9 contributes modestly to moderate and heavy rainfall forecasts, but its impact on light and storm precipitation is limited—likely due to its lower spatial resolution and less detailed physical process representations.

META and temporal features improve forecasts for moderate to storm precipitation but slightly degrade performance for light rainfall, possibly due to increased noise. Heavier precipitation events tend to exhibit clearer spatial patterns and more distinct temporal evolution, which can be effectively leveraged by topographic and temporal features.

Overall, by integrating multiple NWP model outputs and auxiliary features, GFRNet substantially improves the accuracy and resolution of precipitation forecasts. The ablation results highlight the model's effectiveness in forecasting moderate to extreme precipitation and demonstrate its robustness to missing input sources, further underscoring its practical applicability.

*Author contributions.* ZLF, QZ, HMC, and XMW initiated the study, and QZ supervised and administered the project. ZLF, ZZC, and HLL prepared all the data and wrote the training and evaluation scripts together. All authors contributed to the writing and editing of the paper.

*Competing interests.* The authors declare that they have no conflict of interest.

*Acknowledgements.* We extend our heartfelt thanks to Zhang Dan for her patient guidance in writing, the Tianhe team for providing computational resources and technical support, and the China Meteorological Administration for providing valuable data. This work was supported by the National Natural Science Foundation of China (Grant Nos. U2142214 and 42030611), the CMA Innovation Foundation (CXFZ2023J001), and the Open Grants of the State Key Laboratory of Severe Weather (2023LASW-B05).

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
