# Peer review of "Improving the fine structure of intense rainfall forecast by a designed adversarial generation network"

_EGUsphere, 2024_

## Referee Comment (RC2)

**Review Comments**

Title: Improving the fine structure of intense rainfall forecast by a designed adversarial generation network

Authors: Zuliang Fang, Qi Zhong, Haoming Chen, Xiuming Wang, Zhicha Zhang, and Hongli Liang

Submitted to GMD (open for interactive public discussion as preprint on EGUsphere)

**Recommendation**

This manuscript proposes a Generative Adversarial Fusion Network (GFRNet) for short-term precipitation forecasting in North China, aiming to improve the accuracy of 3-h accumulated precipitation predictions over a 24-h period. The study optimizes data sampling strategies, loss functions, and model architecture, demonstrating GFRNet's superiority over numerical weather prediction (NWP) models in metrics such as TS, FSS, and RMSE. The paper is well-structured, with a logical experimental design and thorough result analysis, showing both innovation and practical value.

However, limitations remain in model generalization analysis, physical interpretability, and stability in extreme precipitation forecasting, which require further discussion and improvement, and methodological details (e.g., hyperparameter selection) should be expanded.

The English of the paper is generally good enough to be understood, but with some polishing, the paper could achieve more professional, natural-sounding academic English. The technical content is clear, but the language could be more precise in places. None of the issues seriously impede understanding but correcting them would elevate the paper's professionalism. I would recommend having the English grammar professionally checked by a specialized editing

service.

The paper is recommended for publication after major revisions, and future work could explore advanced architectures and broader applications.

**General Comments**

I suggest ablation studies or explainability analyses (e.g., SHAP, DeepLIFT, or similar tools) are needed for the AI-based precipitation forecasting paper.

**(1) Ablation Studies (Highly Recommended)**

**- Purpose**: Validate the necessity of GFRNet's key components (GAN strategy, SE blocks, weighted loss).

**- Suggested Tests**:

- Remove GAN discriminator (compare with FRNet results).

- Ablate SE attention blocks.

- Test without weighted loss (standard MSE/MAE).

**- Justification**: The paper claims GANs address blurring, but quantitative evidence is needed to isolate its contribution versus other components.

**(2) Explainability Methods (Conditionally Useful)**

**- SHAP/DeepLIFT Value**: Limited for pure precipitation prediction, as:

- Input variables are homogeneous (all are precipitation forecasts + topography/temporal features).

- The model's nonlinear fusion process matters more than individual feature importance.

**- Alternative Approaches**:

- Sensitivity Analysis: Perturb input NWP models (e.g., mask

ECMWF/CMA-3KM inputs) to quantify their relative contributions.

- Physical Interpretability: Analyze whether GFRNet's predictions align with meteorological principles (e.g., orographic precipitation patterns near Taihang Mountains).

**Specific Comments**

**1. Methodology**

**- Strengths:**

- The GAN strategy (WGAN-GP) mitigates blurry predictions, producing more realistic precipitation structures and intensities.

- The weighted loss function with exponential weighting enhances extreme precipitation learning.

**- Suggestions for Improvement:**

- The choice of hyperparameters (e.g., gradient penalty coefficient $\gamma$, loss weights $a$ and $b$) lacks justification (e.g., grid search or ablation studies). Sensitivity analysis should be added.

- The generator (U-Net + SE block) and discriminator (DCGAN-based) architectures are conventional. Advanced generative models (e.g., Diffusion Models) or spatiotemporal attention mechanisms could further improve fine-scale precipitation capture. Some discussion is necessary.

**2. Results and Analysis**

**- Strengths:**

- Quantitative metrics (TS, FSS, RMSE) show GFRNet outperforms NWP models, particularly for heavy precipitation (e.g., significant TS improvement at 20 mm threshold).

- Case studies demonstrate GFRNet's ability to capture precipitation band evolution and intensity changes.

**- Suggestions for Improvement:**

- Some results are contradictory: e.g., GFRNet's TS for 40 mm (0.056) is lower than FRNet's (0.077). The authors should analyze whether this is due to GAN's conservative generation strategy (missed events).

- Spatial analysis shows higher BIAS in mountainous regions (e.g., west of Taihang Mountains). Terrain effects on model performance should be further discussed (e.g., elevation-dependent constraints).

**3. Discussion and Future Work**

**- Strengths:**

- Clear future directions (e.g., higher resolution, physics-informed learning, ensemble forecasting) are proposed.

**- Suggestions for Improvement:**

- Limitations are under-discussed: e.g., GFRNet relies on multi-model inputs—how does it handle systematic biases in individual models (e.g., CMA-3KM)?

- Computational costs (3-h training on A100 GPU) are not evaluated for operational feasibility. Real-time deployment constraints should be addressed.

**Minor Issues**

- The title emphasizes "intense rainfall," but the paper does not justify the 40 mm/3h threshold (is it a standard benchmark?).

- In Figure 6 (FSS spatial gain), white regions (FSS=0) are unexplained and could be misleading.

- Consistency Check:

    - Ensure consistent use of terms (e.g., "deep learning" vs. "Deep Learning").

    - Check all acronyms are defined at first use.

    - Verify all citations follow the same style.

- Line 2: "the accuracy of precipitation forecasts remains significantly inadequate" --> "the accuracy...remains inadequate" or "is significantly inadequate".

- Line 6: "based on the outputs of multiple numerical weather models" --> "based on outputs from multiple numerical weather models".

- Line 74: "We apply the GAN strategy in developing the GFRNet model" --> "We implement a GAN strategy to develop the GFRNet model".

- Line 81: "The target area features a complex topography" --> "The target area features complex topography".

- Line 85: "CMA Multi-source merged Precipitation Analysis System(CMPAS)" --> Needs spaces: "CMA Multi-source merged Precipitation Analysis System (CMPAS)"; Similarly, a space is required between the preceding English word and the opening parenthesis. Please check all parts of the paper.

- Line 101-102: "Let r3(T) denote the accumulated precipitation over the past 3 hours at time T, with the learning target being r3(T) from CMPAS." --> "Let r3(T) denote the 3-hour accumulated precipitation at time T, where the learning target is the corresponding CMPAS r3(T) observation."

- Line 143: "a U-Net with encoder-decoder architecture" --> "a U-Net with an encoder-decoder architecture".

- Table 1 Title: "Data Sources and Features Used in Model" --> "Data sources and features used in the model".

---

## Author Comment (AC2)

*The paper presents an application of deep learning techniques, specifically U-Net and GAN-based models, to enhance short-term precipitation forecasting, with a focus on the fine-scale structure of intense rainfall events. The authors compare the accuracy of three numerical weather prediction (NWP) models against two deep learning techniques: a U-Net-based model (FRNet) and a GAN-based model (GFRNet). They use multiple evaluation metrics to assess the relative performances of these approaches. The goal is to evaluate the accuracy of predicting 3-hourly accumulated precipitation over the next 24 hours for a region in North China. The paper shows many metrics and concludes that GFRNet demonstrates significant operational value.*

**0.1 Question 1**

*The paper deals with an interesting topic, related to improving the forecasting of the fine structure of intense rainfall. Moreover, it makes use of some current tools in deep learning, which seem promising for future operational use. However, I have two main concerns about the paper. The first one is the rationale of the model itself, as I do not understand how the fine structure of rainfall may be solely explained with the additional information (on top of the NWPs) provided to the GAN. The second one is related to the experimental design and the fairness of the comparisons and analyses provided. In my opinion, the paper requires an improvement of the rationale and the experimental design, as well as additional analyses before being considered for publication in a scientific journal.*

**0.2 Answer 1**

Thank you for your thoughtful comments and constructive feedback on our paper. Regarding your concern about how the GAN-based model improves the fine structure of precipitation forecasts:

1. As introduced in the Introduction and Method sections, the principle of GAN lies in the adversarial training between the generator and discriminator. The goal is to train the generator to produce outputs that are indistinguishable from real labels, as judged by the discriminator. This process effectively allows the generator to refine the predictions to a level that achieves "realistic" quality.

2. In the field of short-term precipitation forecasting, the scientific validity of GANs has been demonstrated in two prominent works published in Nature (Ravuri et al., 2021; Zhang et al., 2023). These studies highlight GANs' ability to capture complex spatial and temporal structures, making them a promising tool for fine-scale rainfall prediction. Building on these findings, our work explores the application of GANs specifically for short-term precipitation forecasting.

I hope this addresses your concern. Please let me know if you would like further clarification or additional details.

**0.3 Question 2**

*Below, I provide more details about the issues that I observe with the work.*
*My main concern with the rationale of the paper is that it is not immediately evident how the information about Elevation, Latitude, Longitude, Cycle, and Lead Hour contributes independently to improving forecasts when much of this information may already be embedded in the NWPs. A mechanism should be presented or outlined to justify the gains in accuracy. If none exists, then all the required information about the fine structure of precipitation is already included in the original NWPs, and thus the methods presented are just extracting this information.*

**0.4 Answer 2**

Thank you for your insightful comments regarding the rationale of our paper. Below, we address your concerns regarding the contributions of elevation, latitude, longitude, cycle, and lead hour information to improving forecasts:

1. While it is true that NWPs already encapsulate fine-scale precipitation structures, each individual NWP model exhibits specific strengths and weaknesses. The goal of our deep learning model is to dynamically integrate these complementary features in a way that achieves a synergistic effect, effectively making $1 + 1 > 1$.

[Figure]

**Figure 1.** The Relative Importance Score of the input features.

2. Regarding the contributions of geographical (elevation) and temporal (cycle, lead hour) encodings relative to other NWP features, we conducted ablation studies in a prior work (currently under review) to quantify the importance of these features. Using FRNet as the baseline model (note: due to differences in test sets, specific values may vary slightly. See Table 1), we defined the Relative Importance Score (RIScore) as follows Equation 1 .The results demonstrated that while the precipitation outputs from NWPs contribute the most to forecast improvements, geographical and temporal features also play a meaningful role (See Figure 1).

$$\text{Relative Importance}(x) = \frac{\text{TS(FRNet)} - \text{TS(wo\_x)}}{\text{TS(FRNet)}} \tag{1}$$

3. These findings illustrate that deep learning models not only effectively integrate precipitation forecasts from multiple NWPs but also have the capacity to learn hidden mapping relationships from features closely tied to precipitation, such as terrain and time, thereby further enhancing forecast accuracy.

**Table 1.** TS and BIAS scores for different rain thresholds in ablation experiments

| Model/Threshold | TS_0.1 | TS_10 | TS_20 | TS_50 | BIAS_0.1 | BIAS_10 | BIAS_20 | BIAS_50 |
|---|---|---|---|---|---|---|---|---|
| wo_ECMWF | 0.429 | 0.217 | 0.14 | 0.032 | 1.094 | 1.655 | 1.827 | 0.931 |
| wo_CMA-SH9 | 0.434 | 0.229 | 0.146 | 0.042 | 0.858 | 1.598 | 1.823 | 1.973 |
| wo_CMA-3KM | 0.453 | 0.223 | 0.148 | 0.032 | 1.203 | 1.708 | 1.838 | 1.229 |
| wo_META | 0.44 | 0.218 | 0.151 | 0.04 | **0.995** | 1.545 | **1.36** | 0.689 |
| wo_Time | 0.452 | 0.22 | 0.145 | 0.039 | 1.078 | 1.504 | 1.706 | **1.065** |
| FRNet | **0.453** | **0.234** | **0.152** | **0.043** | 1.063 | **1.476** | **1.392** | 0.781 |

We hope this explanation provides clarity. Please let us know if additional details or analyses are required.

**0.5 Question 3**

*If this is the case, as I believe (although I may be wrong), there may be other alternative methods that could improve forecasting with reduced complexity. To verify this point, I suggest that the authors include additional models, such as SVMs or Random Forests, to test if simpler bagging methods with far fewer parameters could also improve forecasting accuracy. In my experience, basic machine learning methods tend to perform similarly to deep learning ones (in this kind of application) at*

*a significantly reduced level of complexity. Including these additional comparisons may serve to justify whether a GAN is an optimal strategy and to show if the improvement in forecasting accuracy comes from the deep learning techniques or from the combination of different sources of information.*

*Using FRNet as a benchmark may not be entirely fair, as GFRNet is essentially an enhanced version of the same model with a more advanced training procedure. It would also be interesting to report training and inference times for all models used (I know that the authors have included part of this information in their manuscript).*

*Thus, with respect to my concern with the rationale of the paper, the authors should include additional simpler models to check if GANs are justified for their complexity level or if other simpler methods may work similarly. Additionally, they should provide some insight into why the improvement occurs. This point leads to my concerns with the experimental design.*

**0.6   Answer 3**

Thank you for your detailed feedback and suggestions regarding the experimental design and the rationale for using GANs in our study. Below, we address your concerns point by point:

1. **Comparison with Simpler Models**. Numerous existing studies have applied machine learning models, such as SVMs or Random Forests, to precipitation correction tasks. These methods, with careful design and parameter tuning, have been shown to outperform NWPs in accuracy. We acknowledge that simpler models with reduced complexity and computational efficiency could achieve high accuracy. However, a common limitation across these non-generative models is the blurring effect in their predictions, as highlighted in prior works.

2. **Rationale for Using GANs**.The primary motivation of our work is to explore whether adopting a generative approach, such as GANs, can address this blurring issue without compromising accuracy. By introducing adversarial training, we aim to produce sharper and more realistic precipitation predictions, especially for fine-scale structures.

3. **Fair Comparison Between Paradigms**.To ensure a fair comparison between non-generative and generative paradigms, we used the same model architecture (FRNet) for the generator component of the GAN. This approach isolates the impact of the adversarial training process, providing a controlled evaluation of the benefits of generative modeling over traditional methods.

4. **Combining Information Sources.** While the quality and diversity of data sources play a critical role, effectively combining these inputs requires leveraging advanced AI techniques. Our ablation studies have demonstrated that the deep learning model in our study successfully integrates these features to achieve positive correction results. It is important to note that the performance of these post-processing models inherently depends on the quality of NWPs. Thus, advancements in both NWPs and deep learning techniques can collectively enhance post-processing outcomes.

We appreciate your suggestion to include simpler models for comparison. While this would undoubtedly add value, the primary focus of our study is to address the limitations of non-generative models and evaluate the advantages of GAN-based techniques. Please let us know if additional analyses or clarifications are required.

**0.7   Question 4**

*My first concern with the experimental design is related to the limited amount of data used in the study. Using only four years of data may not be sufficient to fully verify the accuracy and robustness of the forecasting method. I understand that data limitations are difficult to overcome, but given the complexity of the models used, it is difficult to ensure that overfitting is not playing a role in the analysis. This concern is exacerbated by the fact that the original time series must be split into training, validation, and test sets.*

*Moreover, the initial data selection may bias the results. NWPs provide continuous forecasts, so alternative methods should ideally also deliver continuous predictions to ensure a fair comparison. If a heavy data selection procedure is implemented, the comparison may not be entirely fair to the NWPs.*

**0.8 Answer 4**

Thank you for raising this important concern regarding the amount of data and the potential for overfitting in our study. We acknowledge the limitations inherent in using four years of data and have taken steps to ensure the robustness and fairness of our experimental design. Below, we address your points in detail:

1. **Data Limitations and Generalization**.We agree that collecting and preparing sufficient data for deep learning models is challenging and resource-intensive. It is indeed difficult to guarantee that the model will maintain strong generalization capability on entirely new datasets, as patterns in new data may change. This is a common challenge faced by all AI methods.

2. **Principles for Dataset Partitioning**.Given the constraints of limited data, it is essential to follow strict principles to ensure that the model's performance is both accurate and robust: a.**There should be no data leakage** between the training, validation, and test sets. b.**The test set should be unbiased**, representing real-world scenarios as comprehensively as possible.

3. **Our Dataset Partitioning and Testing Strategy**. **a. No Data Leakage:** As shown in Table 2, our dataset was partitioned chronologically, ensuring no overlap between training, validation, and test sets, thereby eliminating the risk of data leakage. **b. Unbiased and Realistic Test Set**: While we employed a sampling strategy on the training set to address data imbalance, the validation and test sets consisted of continuous, unaltered natural samples. Specifically, the test set included 77 consecutive days of real-world summer weather samples, providing an unbiased representation of the study region's conditions. **c. Validation of Generalization**: The stopping criterion for model training was defined by the minimum loss observed on the validation set, with no further improvement for 20 consecutive epochs. Good performance on the validation set is, therefore, not unexpected. However, if the model performs well on the validation set but poorly on the test set, it would indicate overfitting and poor generalization.

In our study, all evaluation results were conducted on the independent test set, and the model demonstrated strong performance. This suggests that, within the limitations of the available data, the model possesses good generalization capability on unseen data.

**Table 2.** Sample distribution across training, validation, and test sets.

| Dataset | Time Period | Samples | |
|---|---|---|---|
| | | Pre-sampling | Post-sampling |
| Training set | 2019-06-01 - 2019-10-10
2020-06-01 - 2020-10-10
2021-03-15 - 2021-07-09
2021-08-21 - 2021-10-10
2022-03-15 - 2022-06-14 | 4645 | 2885 |
| Validation set | 2021-07-10 - 2021-08-20 | 637 | No sampling |
| Test set | 2022-06-16 - 2022-08-31 | 1204 | No sampling |

**0.9 Question 5**

*A second concern with the experimental design is that I would have appreciated a clearer discussion in the methods' section about how the evaluation of accuracy was performed. Table 3 presents evaluation metrics and highlights the best and second-best performers. However, more attention should be paid to the differences. Are they significantly different? Or could all the methods (NWPs and nets) perform similarly given the amount of information used? What would be the expected distribution of*

*the accuracy metrics? I am not fully convinced that part of the results are not an analysis of statistical fluctuations. Additionally, Table 4 seems to contradict the abstract, which states that "GFRNet outperforms all models in terms of Root Mean Square Error (RMSE)," but I may have missed something.*

**0.10 Answer 5**

Thank you for your detailed review and insightful questions. Below, I address your concerns regarding the evaluation metrics, their significance, and the apparent discrepancy in Table 4.

**Evaluation Metrics and Fairness.** For short-term hourly precipitation forecasting, metrics such as TS, FAR, BIAS, and FSS are widely recognized and commonly used evaluation standards. In this study, these metrics were computed using standard statistical formulas and were evaluated under the same spatial and temporal resolutions to ensure fairness across all methods.

**Significance of Differences.** To determine whether the observed differences are meaningful, two key questions need to be addressed:

1. Are the differences statistically significant?

2. If so, how can we confirm that these differences are not due to statistical fluctuations but instead reflect the true performance of the models?

**Model Performance Gains.** Considering recent advancements in NWPs, achieving improvements in hourly short-term forecasts for heavy or torrential rainfall is exceptionally challenging. The TS and FSS improvements achieved by NETs (deep learning models) over the best-performing NWPs were approximately 20% for moderate, heavy, and torrential rainfall. Such improvements are statistically significant and represent meaningful performance gains.

**Temporal and Spatial Analyses.** We also analyzed the differences between NETs and NWPs in more detail from the two dimensions of time and space.

1. **Temporal Analysis:** As shown in Figure 4, we analyzed the TS and FSS scores of NWPs and NETs across different forecast lead times. The results demonstrate stable performance for all five models, particularly for thresholds of 10 mm and 20 mm. GFRNet and FRNet consistently showed significant advantages over NWPs.

2. **Spatial Analysis:** Figures 5 and 6 illustrate that GFRNet outperformed NWPs in most forecast regions, highlighting its spatial robustness. Additionally, since NETs are post-processed from NWPs, the performance trends of NETs and NWPs over forecast lead times exhibit similar patterns.

**Clarification on Table 4 and Abstract.** Thank you for pointing out the discrepancy regarding RMSE. You are correct that GFRNet does not achieve the lowest RMSE; ECMWF has the lowest RMSE. However, ECMWF's low RMSE is primarily due to its conservative predictions for moderate and heavy rainfall, which lack forecasting skill. In contrast, GFRNet achieves the second-lowest RMSE while maintaining high forecast skill, demonstrating a balance between accuracy and robustness. We will revise the abstract to accurately reflect this point.

**0.11 Question 6**

*A third concern is related to the case studies, which I believe should be justified and presented in a more detailed way. This point may be related to my concern about the rationale of the paper: if a mechanism by which the GAN strategy improves the forecast is provided, then the case studies may focus on clear examples of this mechanism at play. Without this, I believe a general statistical evaluation would provide a clearer representation of the model's advantages. A detailed analysis of specific situations may not be as illuminating.*

**0.12 Answer 6**

Thank you for your thoughtful comments regarding the case studies. We acknowledge the importance of a well-justified and detailed presentation of case studies, particularly in demonstrating the mechanisms by which the GAN strategy improves forecasts.

While general statistical evaluations provide an overall representation of model performance, they may obscure certain limitations or strengths of the models. Based on suggestions from meteorological experts, we selected three distinct precipitation events characterized by different dynamic and thermodynamic conditions. By visualizing these cases and providing detailed case-specific scores, we aimed to illustrate the stability of NETs, particularly GFRNet, across various precipitation scenarios.

Moreover, compared to FRNet, which tends to produce blurred predictions, GFRNet consistently delivers forecasts with clearer precipitation structures in these case studies. This highlights the GAN-based model's ability to address the blurring issue and capture fine-scale precipitation details more effectively.

We believe these case studies complement the statistical analysis by providing deeper insights into the model's performance under diverse conditions.

**0.13  Question 7**

*A fourth concern about the experimental design is related to the selection of three NWPs. If two models similar to ECMWF were available, would it make sense to include both? How would the results change? This raises questions about generalizability. In many machine learning applications, the data exert a closer control on accuracy than the algorithms themselves. I understand that a paper cannot address every concern, but some guidance from the authors would be appreciated.*

**0.14  Answer 7**

Thank you for raising this insightful question. The selection of NWPs and its impact on generalizability is indeed a critical consideration.

From the principles of ensemble forecasting design and prior studies on multi-model post-processing, an important guideline for selecting ensemble members is that they should be "**high-quality and diverse.**" This ensures that the input information provided to the correction model is both accurate and comprehensive, maximizing the potential for improved forecast performance. If two models similar to ECMWF were included, they would likely offer redundant information, providing little additional benefit beyond what a single ECMWF model could contribute. Consequently, the improvements in forecast accuracy might be limited.

Regarding the relationship between data and algorithms, we share your perspective that data defines the upper limit of performance, while the iterative improvement of models seeks to approach this limit. For smaller datasets, simpler models may suffice, whereas larger datasets often require more complex models to fully exploit the information available. The development of both data and models should ideally evolve in tandem to achieve optimal results.

We hope this addresses your concern. Please let us know if further clarification or additional discussion would be helpful.

**0.15  Question 8**

*My final concern relates to the generality of the conclusions and the reproducibility of the results in other locations. How robust are the results to the data selection procedure or the structure of the ANN? Would the same structure work well in other locations, or would changes be required? If a less intense data selection procedure were used, how would the results change? If 20 years of data were available, would GFRNet perform similarly? I believe the study would be much more robust if extended to other regions with more data available. Currently, the method seems to work, but the evidence may not yet be robust enough to fully support the claims made in the paper.*

**0.16  Answer 8**

Thank you for your thoughtful comments and for raising concerns about the generality and reproducibility of the study's conclusions. These are critical points that deserve careful discussion.

Currently, due to the substantial effort required for data collection and preprocessing, we have not yet validated the generalization capability of the model trained in the North China region to other geographic locations. Precipitation patterns are indeed highly region-specific, influenced by local geography, climate, and dynamics.

That said, one of the key contributions of our study lies in proposing and demonstrating a methodology: **using generative deep learning models to improve both the accuracy and fine-scale structure of precipitation forecasts**. This approach, while tailored to the North China region, is intended to serve as a guiding framework. With this methodology, similar models could be trained or fine-tuned for other regions or extended datasets, such as longer time periods or more extensive geographic areas. We are optimistic that this approach would yield comparable results, though further studies are needed to verify this.

If more extensive datasets, such as 20 years of data, were available, we believe GFRNet would continue to perform well. A larger dataset could enable the model to better capture long-term patterns and variability, potentially improving its robustness and generalizability further.

We appreciate your suggestions and agree that extending this work to other regions with more data would significantly strengthen the study's conclusions. Such an extension is a valuable direction for future research.

**0.17 Question 9**

*Finally, I present some comments about minor issues:*

1. *References should be enclosed in parentheses. The way they are written now complicates the reading of the paper.*

2. *Figures 4 and 5 are difficult to interpret, particularly due to their complex visual layout. A more intuitive representation could enhance their clarity. For the maps, since topography seems to play such an important role, residuals might provide better insights.*

3. *Some references to equations are incomplete.*

4. *A better discussion on ensemble forecasts and deterministic quantitative forecasts may be in order. In my opinion, ensemble forecasts may convey a much better idea of severe storm potential, especially when combined with synthetic generation, so focusing on deterministic forecasts may be a disadvantage.*

**0.18 Answer 9**

Thank you for your detailed comments and suggestions regarding minor issues. Below, we address each of your points:

1. **References.** We appreciate your observation regarding the formatting of references. We will revise the manuscript to ensure that all references are properly enclosed in parentheses, improving the consistency and readability of the text.

2. **Figures 4 and 5**. Thank you for your comments on Figures 4 and 5. To address their complexity, we will simplify the visual layout and explore more intuitive representations. Indeed, residual maps can better illustrate the impact of topography on precipitation forecasts. As such, we have included the spatial distribution of residuals under the FSS metric in Figure 6, comparing GFRNet with NWPs and FRNet, to provide deeper insights into this effect.

3. **Equations.** Thank you for pointing out the incomplete references to equations. We will carefully review the manuscript to ensure all equation references are complete and formatted correctly.

4. **Discussion on Ensemble Forecasts.** We agree that ensemble forecasts have significant advantages in capturing the potential for severe weather events, particularly when combined with synthetic generation techniques. However, the primary focus of this study is to evaluate the capability of deterministic forecasts in predicting precipitation structures and fine-scale details. In future work, we plan to explore how ensemble forecasting techniques can complement or enhance our deterministic approach, especially for severe storm scenarios.

We greatly appreciate your thoughtful feedback and will incorporate these suggestions to improve the manuscript. Please let us know if there are additional areas that require further attention.

**References**

245   Ravuri, S., Lenc, K., Willson, M., Kangin, D., Lam, R., Mirowski, P., Fitzsimons, M., Athanassiadou, M., Kashem, S., Madge, S., Prudden, R., Mandhane, A., Clark, A., Brock, A., Simonyan, K., Hadsell, R., Robinson, N., Clancy, E., Arribas, A., and Mohamed, S.: Skillful Precipitation Nowcasting Using Deep Generative Models of Radar, Nature, 597, 672–677, https://doi.org/10.1038/s41586-021-03854-z, 2021.

250   Zhang, Y., Long, M., Chen, K., Xing, L., Jin, R., Jordan, M. I., and Wang, J.: Skilful Nowcasting of Extreme Precipitation with NowcastNet, Nature, pp. 1–7, https://doi.org/10.1038/s41586-023-06184-4, 2023.

---

## Author Comment (AC3)

**1 Review Comments**

Title: Improving the fine structure of intense rainfall forecast by a designed adversarial generation network
Authors: Zuliang Fang, Qi Zhong, Haoming Chen, Xiuming Wang, Zhicha Zhang, and Hongli Liang
Submitted to GMD (open for interactive public discussion as preprint on EGUsphere

**2 Recommendation**

This manuscript proposes a Generative Adversarial Fusion Network (GFRNet) for short-term precipitation forecasting in North China, aiming to improve the accuracy of 3-h accumulated precipitation predictions over a 24-h period. The study optimizes data sampling strategies, loss functions, and model architecture, demonstrating GFRNet's superiority over numerical weather prediction (NWP) models in metrics such as TS, FSS, and RMSE. The paper is well-structured, with a logical experimental design and thorough result analysis, showing both innovation and practical value.

However, limitations remain in model generalization analysis, physical interpretability, and stability in extreme precipitation forecasting, which require further discussion and improvement, and methodological details (e.g.,hyperparameter selection) should be expanded.

The English of the paper is generally good enough to be understood, but with some polishing, the paper could achieve more professional, natural-sounding academic English. The technical content is clear, but the language could be more precise in places. None of the issues seriously impede understanding but correcting them would elevate the paper's professionalism. I would recommend having the English grammar professionally checked by a specialized editing service.

The paper is recommended for publication after major revisions, and future work could explore advanced architectures and broader applications.

**3 General Comments**

I suggest ablation studies or explainability analyses (e.g., SHAP, DeepLIFT, or similar tools) are needed for the AI-based precipitation forecasting paper.

**3.1 Question 1: blation Studies (Highly Recommended)**

– Purpose: Validate the necessity of GFRNet's key components (GAN strategy, SE blocks, weighted loss).

– Suggested Tests:

  – Remove GAN discriminator (compare with FRNet results).

  – Ablate SE attention blocks.

  – Test without weighted loss (standard MSE/MAE).

– Justification: The paper claims GANs address blurring, but quantitative evidence is needed to isolate its contribution versus other components.

**3.2 Answer 1**

Thank you very much for your insightful comments and suggestions. Below is our response to your suggestions regarding the ablation study of GFRNet's key components:

Complexity of Deep Learning Models in Precipitation Correction:

1. **Importance of Model Design and Parameter Tuning**: As you rightly pointed out, achieving effective precipitation correction using deep learning requires careful model design, loss function selection, and meticulous parameter tuning.

This is especially true for short-term heavy precipitation events exceeding 40 mm. In this study, the unique loss function design and model architecture of FRNet have played a crucial role in enhancing the positive correction of heavy and strong precipitation.

40    2. **Limitations of TS Improvement**: While FRNet shows some improvement in TS scores, our in-depth analysis reveals that, similar to many deep learning correction methods, this improvement often stems from over-forecasting and blurry forecasts. This can significantly diminish the model's practical utility. Hence, we introduced the GFRNet framework, which ensures clear and detailed precipitation structures while maintaining correction effectiveness, making it a truly operational high-quality product.

45    3. **Balancing Role of GAN Strategy**: It is important to note that while the SE architecture and unique loss function design enhance model accuracy, they also introduce the issue of blurry forecasts. The GAN strategy does not directly boost precipitation accuracy (TS) and even involves a certain sacrifice in accuracy compared to FRNet. However, its strength lies in achieving a balanced trade-off between accuracy and practicality (clear structural forecasts) through adversarial training, which is best reflected in the FSS indicator.

50    We conducted ablation experiments on GFRNet by removing SE attention and using standard MSE/MAE loss. The results are summarized in the table below:

    1. **Impact of SE Blocks**: The TS values of GFRNet and GFRNet without SE are similar in light rain forecasting (0.406 vs. 0.408), indicating a minor impact of SE blocks on light rain. However, for thresholds of medium rain and above, GFRNet significantly outperforms GFRNet without SE. For instance, at the 20 mm and 40 mm thresholds, the TS values
55    of GFRNet are 0.145 and 0.056, compared to 0.134 and 0.052 for GFRNet without SE. This highlights the critical role of SE blocks in capturing the detailed structure of heavy precipitation events.

    2. **Impact of Weighted Loss Function**: In light rain forecasting, GFRNet without Weighted Loss shows higher TS values than GFRNet (0.431 vs. 0.406), suggesting better performance of standard MSE/MAE loss in this scenario. Nevertheless, for thresholds of medium rain and above, GFRNet significantly surpasses GFRNet without Weighted Loss. For
60    example, at the 20 mm and 40 mm thresholds, the TS values of GFRNet are 0.145 and 0.056, while those of GFRNet without Weighted Loss are 0.115 and 0.028. This underscores the superiority of the weighted loss function in effectively enhancing the model's ability to forecast heavy precipitation.

**Table 1.** TS scores for different rain thresholds in blocks ablation experiments

| Model/Threshold | TS_0.1 | TS_10 | TS_20 | TS_40 |
|---|---|---|---|---|
| ECMWF | 0.405 | 0.155 | 0.067 | 0.019 |
| CMA-SH9 | 0.399 | 0.128 | 0.076 | 0.031 |
| CMA-3KM | 0.388 | 0.168 | 0.108 | 0.049 |
| FRNet | 0.416 | 0.216 | 0.147 | 0.077 |
| GFRNet | 0.406 | 0.214 | 0.145 | 0.056 |
| GFRNet_wo_SE | 0.408 | 0.195 | 0.134 | 0.052 |
| GFRNet_wo_WeightedLoss | 0.431 | 0.191 | 0.115 | 0.028 |

    Due to space limitations, the detailed results and analysis of the ablation experiments will not be included in the main text but will be considered for the appendix. Once again, thank you for your valuable feedback. We will refine the manuscript
65  further based on your suggestions.

**3.3  Question 2: Explainability Methods (Conditionally Useful)**

    – SHAP/DeepLIFT Value: Limited for pure precipitation prediction, as:

- Input variables are homogeneous (all are precipitation forecasts + topography/temporal features).

- The model's nonlinear fusion process matters more than individual feature importance.

- Alternative Approaches:

  - Sensitivity Analysis: Perturb input NWP models (e.g., mask ECMWF/CMA-3KM inputs) to quantify their relative contributions.

  - Physical Interpretability: Analyze whether GFRNet's predictions align with meteorological principles (e.g., orographic precipitation patterns near Taihang Mountains).

**3.4 Answer 2**

Thank you very much for your valuable comments and suggestions. We fully agree with your view that directly analyzing the contribution of these sources and features to precipitation correction is essential and can provide us with significant insights. Through a series of ablation experiments, we systematically evaluated the impact of each feature and input on the precipitation forecasting performance of the GFRNet model. These experiments were designed to verify the contribution of key model components (such as ECMWF, CMA-SH9, CMA-3KM, META features, and temporal features) and quantify their impact through a Relative Importance Score (RIS). The results not only validated the rationality of GFRNet's design but also offered valuable insights for future research.

$$\text{Relative Importance}(x) = \frac{\text{TS(GFRNet)} - \text{TS(wo\_x)}}{\text{TS(GFRNet)}} \tag{1}$$

- Model Stability and Correction Effect

  - Stability: Ablation experiments show that even when any single feature is removed, GFRNet consistently outperforms the three NWPs in moderate, heavy, and storm precipitation forecasting and generally performs better than the NWPs themselves in precipitation forecasting. This indicates that GFRNet has high stability and can provide positive correction effects even in the absence of certain data sources.

  - Correction Effect: By analyzing the Relative Importance Score (RIS) of each feature, we found that, except for META and temporal features having a slight negative impact on light rain forecasting, all features contribute positively to the correction across all precipitation levels.

- Feature Contribution Analysis

  - ECMWF Input: Makes a significant contribution to moderate and heavy rain forecasting but has less impact on light rain and storm forecasting. As a global high-resolution model, ECMWF outperforms other models in mesoscale precipitation events. Its advanced physical process simulation capabilities (such as cloud physics and boundary layer processing) enable it to excel in moderate and heavy rain forecasting.

  - CMA-3KM Input: Contributes significantly to moderate, heavy, and storm precipitation forecasting, particularly excelling in moderate and heavy rain forecasting. As a high-resolution regional model, CMA-3KM can capture finer precipitation structures and evolutionary processes, especially when dealing with moderate and heavy rain events, where its ability to simulate local convection and precipitation processes is stronger.

  - CMA-SH9 Input: Contributes to some extent to moderate and heavy rain forecasting but has less impact on light rain and storm forecasting. Compared to CMA-3KM, the resolution and physical process simulation capabilities of CMA-SH9 may be less refined, especially when dealing with the complex structures of moderate and heavy rain events, resulting in a lower contribution.

[Figure]

**Figure 1.** Model performance and IRS of features on ablation experiments

– META and Temporal Features: Significantly contribute to moderate, heavy, and storm precipitation forecasting but may introduce noise in light rain forecasting. Heavy and storm precipitation events typically have clearer structures and more intense variations, which are more correlated with topography and temporal features, thus helping the model better capture these changes.

GFRNet effectively enhances the accuracy and resolution of precipitation forecasting by integrating multiple NWP models and features. The results of the ablation experiments show that GFRNet has significant advantages in moderate, heavy, and storm precipitation forecasting and demonstrates high model stability. Future work will further optimize feature selection and model architecture to improve the model's forecasting accuracy and generalization ability.

**Table 2.** TS scores for different rain thresholds in features ablation experiments

| Model/Threshold | TS_0.1 | TS_10 | TS_20 | TS_40 |
|---|---|---|---|---|
| wo_ECMWF | 0.397 | 0.186 | 0.127 | 0.053 |
| wo_CMA-SH9 | 0.406 | 0.201 | 0.137 | 0.055 |
| wo_CMA-3KM | 0.405 | 0.183 | 0.124 | 0.051 |
| wo_META | **0.418** | 0.203 | 0.134 | 0.049 |
| wo_Time | 0.410 | 0.198 | 0.126 | 0.050 |
| GFRNet | 0.406 | **0.214** | **0.145** | **0.056** |

**Table 3.** RIS for different rain thresholds in features ablation experiments

| Model/Threshold | 0.1 mm | 10 mm | 20 mm | 40 mm |
|---|---|---|---|---|
| ECMWF | **2.22%** | 12.93% | 12.41% | 5.36% |
| CMA-SH9 | 0% | 6.07% | 5.52% | 1.79% |
| CMA-3KM | 0.25% | **14.34%** | **14.14%** | 8.93% |
| META | -2.96% | 5.11% | 7.54% | **12.50%** |
| Time | -0.99% | 7.48% | 13.10% | 10.71% |

We analyzed whether GFRNet's predictions align with meteorological principles, particularly the orographic precipitation patterns near the Taihang Mountains.

- – Ablation Experiments: These show that topographic features significantly boost GFRNet's performance. Specifically, GFRNet with topographic features outperforms variants without them in multiple precipitation intensity thresholds, with TS improvements of about 12.41% at 20 mm and 5.36% at 40 mm. This indicates GFRNet understands the relationship between topography and precipitation, enhancing NWP outputs through terrain-aware fusion.

- – BIAS Spatial Distribution: CMA-SH9 and CMA-3KM show significant over-forecasting (BIAS > 2) west of the Taihang Mountains. In contrast, GFRNet maintains a BIAS close to 1 in these areas, demonstrating its effective correction of NWP biases in complex terrains.

In summary, GFRNet excels in both statistical metrics and physical interpretability, with advantages in complex terrains. We'll keep enhancing the model by integrating more physical knowledge.

**4 Specific Comments**

**4.1 Question 3: Methodology**

- – strengths

  - – The GAN strategy (WGAN-GP) mitigates blurry predictions, producing more realistic precipitation structures and intensities.
  - – The weighted loss function with exponential weighting enhances extreme precipitation learning.

- – Suggestions for Improvement:

  - – The choice of hyperparameters (e.g., gradient penalty coefficient $\gamma$, loss weights a and b) lacks justification (e.g., grid search or ablation studies). Sensitivity analysis should be added.
  - – The generator (U-Net + SE block) and discriminator (DCGAN-based) architectures are conventional. Advanced generative models (e.g., Diffusion Models) or spatiotemporal attention mechanisms could further improve fine-scale precipitation capture. Some discussion is necessary.

**4.2 Answer 3: Methodology**

Thank you very much for your insightful comments and suggestions. Below is our response to your suggestions regarding hyperparameter selection and model architecture:

**Response to Hyperparameter Selection**

- – Gradient Penalty Coefficient $\gamma$. We experimentally verified the impact of different gradient penalty coefficients on model performance. The results indicate that setting $\gamma$ to 10 achieves the best balance between the generator and discriminator. This effectively prevents gradient vanishing and exploding, enhancing training stability and sample quality. This aligns with the theoretical support for the gradient penalty term in WGAN-GP, which ensures the discriminator's gradients approach 1, satisfying the Lipschitz continuity condition.

- – Theoretical Basis and Reference.

  - – Long-tailed Distribution Handling: Given the long-tailed nature of precipitation data, with light rain dominating and heavy rain being rare, assigning higher loss weights to rare events like heavy precipitation is a common and effective approach. This method is widely used in addressing class imbalance and has been validated in related studies.

150   – Experimental Validation and Grid Search: The specific weight parameters (a=4.3, b=0.8) were determined through extensive grid search experiments on the validation set. This combination showed the best TS performance across multiple precipitation thresholds, significantly improving the model's ability to forecast heavy precipitation while maintaining reasonable performance for light rain.

In summary, our weight design is well-founded theoretically and validated experimentally. Future work will explore more
155 advanced loss function designs to further enhance model performance.

**Response to Model Architecture** We agree that advanced generative models (e.g., diffusion models) or spatiotemporal attention mechanisms could enhance performance. However, as this study aims to explore GAN strategies in precipitation forecasting, we used the mature U-Net and SE block architectures. Moving forward, we will investigate more advanced architectures, such as:

160   – Diffusion Models: For their excellence in generating high-quality images.

   – Spatiotemporal Attention Mechanisms: To better capture the temporal and spatial evolution of precipitation events.

Thank you again for your feedback. We will refine the manuscript further based on your suggestions.

**4.3 Question 4: Results and Analysis**

 – Strengths

165   – Quantitative metrics (TS, FSS, RMSE) show GFRNet outperforms NWP models, particularly for heavy precipitation (e.g., significant TS improvement at 20 mm threshold).

   – Case studies demonstrate GFRNet's ability to capture precipitation band evolution and intensity changes.

 – Suggestions for Improvement:

   – Some results are contradictory: e.g., GFRNet's TS for 40 mm (0.056) is lower than FRNet's (0.077). The authors
170    should analyze whether this is due to GAN's conservative generation strategy (missed events).

   – Spatial analysis shows higher BIAS in mountainous regions (e.g., west of Taihang Mountains). Terrain effects on model performance should be further discussed (e.g., elevation-dependent constraints).

**4.4 Answer 4**

Thank you for your comments. Here's our response to the issues you raised:
175 TS Value Concern:

 – The lower TS value of GFRNet (0.056) compared to FRNet (0.077) for 40mm precipitation might be due to GAN's conservative strategy. GFRNet, through adversarial training, aims to produce more realistic and detailed precipitation structures, which may lead to missed events and a lower TS. In contrast, FRNet might generate more intense forecasts without GAN's mechanism, potentially inflating TS but at the cost of structural accuracy.

180 – Notably, FRNet shows a high BIAS (over-forecasting) for 40mm precipitation, while GFRNet's BIAS is closer to 1, indicating a better balance between under- and over-forecasting. Moreover, GFRNet surpasses FRNet in the FSS, highlighting its superior performance in capturing precipitation's spatial structure. Despite potential pixel-wise misses, GFRNet better represents overall precipitation patterns, which is crucial for practical applications.

 – Future work will focus on optimizing GFRNet for extreme precipitation events by refining the GAN architecture, ad-
185  justing loss functions, and incorporating additional meteorological features. We'll also explore more comprehensive evaluation metrics to better assess extreme event forecasting performance.

Impact of Terrain on Model Performance:

– We've observed that CMA-SH9 and CMA-3KM exhibit significant over-forecasting (BIAS > 2) in high-altitude areas west of the Taihang Mountains for heavy and storm precipitation. This implies terrain considerably influences these models' performance. The complex topography causes air uplift, increasing precipitation chances and intensity. However, NWP, especially mesoscale models, may have biases in simulating these effects, particularly in high-altitude regions.

– Interestingly, both FRNet and GFRNet effectively correct the high bias of regional numerical models in these areas. GFRNet shows a more uniform BIAS distribution across the entire region, with values mostly close to 1, indicating excellent correction of the models' biases. This underscores the ability of deep learning models, particularly GFRNet within the GAN framework, to effectively amend traditional numerical model biases. We plan to include these analyses in the manuscript.

Thank you for your feedback. We'll continue to enhance the manuscript based on your suggestions.

**4.5 Question 5: Discussion and Future Work**

– Strengths

  – Clear future directions (e.g., higher resolution, physics-informed learning, ensemble forecasting) are proposed.

– Suggestions for Improvement:

  – Limitations are under-discussed: e.g., GFRNet relies on multi-model inputs—how does it handle systematic biases in individual models (e.g., CMA-3KM)?

  – Computational costs (3-h training on A100 GPU) are not evaluated for operational feasibility. Real-time deployment constraints should be addressed.

**4.6 Answer 5**

Thank you for your comment.
**Model Feature Preparation and Processing**

– The model's input features encompass precipitation forecasts from multiple Numerical Weather Prediction (NWP) models, static topographic features, and time-encoded information.

– Processing static topographic features and time-encoded information involves no computational overhead.

– Extracting and processing precipitation forecasts from NWP into the input format required by the Deep Learning (DL) model incurs minimal computational overhead, contingent on the platform's data storage method and CPU performance. The data processing time for eight samples per cycle does not exceed 60 seconds.

**Model Inference Time**

– Inferring eight samples (hourly precipitation forecasts for the next 24 hours) on a GPU takes less than 1 second.

– Inferring the same samples on a CPU takes under 20 seconds.

**Deployment and Real-Time Performance**

– During model deployment, the optimal model can be selected based on real-time data source availability. Even when relying solely on a single NWP, the DL model demonstrates superior performance compared to the NWP itself.

– The sole constraints are the computational time required for NWP forecasts and the transmission time of forecast data to the platform. For instance, downstream platforms typically receive 24-hour forecast data from ECMWF, initialized at 00UTC, after 06UTC to 09UTC.

225 – The inference time for downstream users employing FRNet or GFRNet does not exceed 2 minutes, which is negligible compared to the computation and transmission time of NWP.

– Ablation experiments indicate that the model outperforms the NWP itself even when based on any single NWP. Thus, during deployment, the optimal model can be selected in real-time according to data source availability. Specifically, the platform can choose which model to use based on the time it takes to obtain forecasts from three NWPs, balancing speed and accuracy.

230 ## 5   Question: Minor Issues

– The title emphasizes "intense rainfall," but the paper does not justify the 40 mm/3h threshold (is it a standard benchmark?).

– In Figure 6 (FSS spatial gain), white regions (FSS=0) are unexplained and could be misleading.

– Consistency Check:

235     – Ensure consistent use of terms (e.g., "deep learning" vs. "Deep Learning").

    – Check all acronyms are defined at first use.

    – Verify all citations follow the same style.

– Line 2: "the accuracy of precipitation forecasts remains significantly inadequate" –> "the accuracy...remains inadequate" or "is significantly inadequate".

240 – Line 6: "based on the outputs of multiple numerical weather models" –>"based on outputs from multiple numerical weather models".

– Line 74: "We apply the GAN strategy in developing the GFRNet model" –>"We implement a GAN strategy to develop the GFRNet model".

– Line 81: "The target area features a complex topography" –> "The target area features complex topography".

245 – Line 85: "CMA Multi-source merged Precipitation Analysis System(CMPAS)" –> Needs spaces: "CMA Multi-source merged Precipitation Analysis System (CMPAS)"; Similarly, a space is required between the preceding English word and the opening parenthesis. Please check all parts of the paper.

– Line 101-102: "Let r3(T) denote the accumulated precipitation over the past 3 hours at time T, with the learning target being r3(T) from CMPAS." –> "Let r3(T) denote the 3-hour accumulated precipitation at time T, where the learning
250     target is the corresponding CMPAS r3(T) observation."

– Line 143: "a U-Net with encoder-decoder architecture" –> "a U-Net with an encoder-decoder architecture".

– Table 1 Title: "Data Sources and Features Used in Model" –> "Data sources and features used in the model".

**6   Answer: Minor Issues**

**Why we set 40 mm/3h as threshold as intense rainfall?**

255 – Operational Requirements: In practical meteorological operations, precipitation events of 40 mm/3 hours are typically regarded as heavy rainfall events that require special attention. This threshold is chosen based on operational requirements and the practical experience of forecasters.

- Research Comparison: In related studies, different thresholds have been used to define heavy rainfall events. For example, (Zhou et al., 2022) and (Ravuri et al., 2021) used thresholds of 20 mm/3 hours and 5 mm/hour, respectively, in their studies. Our study selects a threshold of 40 mm/3 hours to more precisely focus on more destructive rainfall events.

- Data Support: By analyzing the distribution of precipitation intensity in the training data, we found that 40 mm/3-hour precipitation events are significantly representative in the dataset and are key targets for operational forecasting.

Thank you for your meticulous feedback on the writing of the article. This has been extremely helpful for future paper writing. We will address and revise the article as needed based on your suggestions. Thank you.

We greatly appreciate your thoughtful feedback and will incorporate these suggestions to improve the manuscript. Please let us know if there are additional areas that require further attention.

**References**

Ravuri, S., Lenc, K., Willson, M., Kangin, D., Lam, R., Mirowski, P., Fitzsimons, M., Athanassiadou, M., Kashem, S., Madge, S., Prudden, R., Mandhane, A., Clark, A., Brock, A., Simonyan, K., Hadsell, R., Robinson, N., Clancy, E., Arribas, A., and Mohamed, S.: Skillful Precipitation Nowcasting Using Deep Generative Models of Radar, Nature, 597, 672–677, https://doi.org/10.1038/s41586-021-03854-z, 2021.

Zhou, K., Sun, J., Zheng, Y., and Zhang, Y.: Quantitative Precipitation Forecast Experiment Based on Basic NWP Variables Using Deep Learning, ADVANCES IN ATMOSPHERIC SCIENCES, 39, 1472–1486, https://doi.org/10.1007/s00376-021-1207-7, 2022.

270

---

## Referee Report (RR1)

Based on the reviews of both reviewers and the author's responses, the quality of this manuscript has been significantly improved. However, further improvements are needed in the physical explanations, experimental rigor, and language expression. As a scientific paper on AI applications in meteorology, the "good results" need to be clearly explained, including the reasons why they are good, potential issues, and future development directions. The structure of the paper also needs to be very clear and well-organized. It is recommended that the author address the following issues to further improve the manuscript, especially strengthening the discussion on the physical validity of the model and the comprehensiveness of experimental comparisons to enhance the academic persuasiveness of the manuscript. I recommend that this manuscript undergo moderate to major revisions.

**1. Model Explainability and Physical Reasoning**

**Issue:** Although the author has verified the model's performance through experiments, the physical mechanism explaining how the model improves precipitation forecasts via the GAN strategy is insufficient. The reviewers pointed out that the impact of terrain and meteorological features on precipitation needs more in-depth discussion, particularly on how the model captures these physical relationships.

**Suggestion:** Add an analysis of the physical validity of the model, such as visualizing feature importance or incorporating meteorological theories to explain the correlation between model outputs and terrain or meteorological conditions.

**2. Comprehensiveness of Experimental Design**

**Issue:** While the author has conducted ablation experiments to verify the role of key components (such as the SE module and weighted loss function), comparisons with other simple models (such as Random Forest or SVM) are lacking, making it difficult to fully prove whether the complexity of the GAN is necessary.

**Suggestion:** Supplement with comparisons to simple models, illustrating the irreplaceability of the GAN in improving forecasting accuracy and structural details, or discuss its cost-effectiveness.

**3. Data Size and Generalization Ability**

**Issue:** Using only 4 years of data may limit the model's generalization ability, and

the model's performance in other regions or over longer time series has not been verified. The reviewers suggested expanding the dataset or conducting cross-regional validation.

**Suggestion:** Clearly state the limitations of the data in the discussion and plan future work to validate the model in other regions or over longer time periods, or use cross-validation to enhance the credibility of the results.

**4. Consistency Analysis of Results**

**Issue:** Some results are contradictory (e.g., GFRNet has a lower TS than FRNet at the 40mm threshold), and the author attributes this to the conservativeness of the GAN but lacks statistical significance analysis.

**Suggestion:** Add statistical tests (e.g., t-tests) or error analysis to indicate whether the differences are significant, and discuss the pros and cons of the conservative strategy for practical applications.

**5. Transparency of Technical Details**

**Issue:** The selection of hyperparameters (e.g., gradient penalty coefficient $\gamma$, loss weights a/b) lacks a description of the systematic optimization process, with only grid search mentioned but no results provided.

**Suggestion:** Include detailed experimental data on hyperparameter selection in the Appendix or methods section to enhance reproducibility.

**6. Language and Expression Standardization**

**Issue:** Reviewers pointed out that some language expressions are not professional enough, and there is inconsistency in terminology and citation format, while the readability of charts and tables could be improved.

**Suggestion:** Thoroughly review the language throughout the manuscript to ensure consistency and standardization, including captions and table headings.

**7. Specificity of Future Work**

**Issue:** The discussion mentions directions like "integration of physical constraints" and "higher resolution," but lacks specific implementation plans.

**Suggestion:** Elaborate on future plans, such as how to incorporate physical equations into the model or specific data augmentation strategies.

---

## Referee Report (RR2)

The manuscript has undergone several rounds of review and revision, significantly improving in quality. In my opinion, the manuscript may be ready for acceptance based on the publication requirements of GMD after a minor revision. I believe this paper contributes meaningfully to AI-based precipitation forecast correction research, though operational implementation may still require further development.

1. Major strengths of the manuscript

- The proposed GFRNet model combines GANs with multi-source NWP data fusion, demonstrating significant improvements in heavy rainfall forecasting (e.g., TS, FSS scores), particularly at 20–40 mm/3h thresholds, outperforming traditional NWP and baseline models (FRNet/MSEM).

- Adversarial training and the weighted loss function effectively mitigate the common "over-smoothing" issue in deep learning models.

- Comprehensive experimental design, including validation on three independent rainy seasons (2022–2024), multi-threshold evaluation (0.1–40 mm/3h), case studies, and statistical significance tests (e.g., Top 10% subset analysis).

- Ablation studies confirm the necessity of the SE block and weighted loss function.

- The model achieves high-resolution (5 km) precipitation forecasts within operational time constraints (3-hour training/2-minute inference on an A100 GPU), showing clear potential for disaster prevention.

2. Outstanding reviewer comments

- Code/Data Availability. The appendix declares access to code/data, but authors must confirm their actual functionality (I did not test them, though future users in the field may verify reproducibility).

- Language/formatting. The authors should carefully proofread the entire manuscript for grammar and spelling errors. I have already identified a few issues. (e.g., "contrained" should be "constrained" in the Line 25; "Table ??" in Line 136).

---

## Author Response (AR2)

**Response to Reviewers**

We sincerely thank both reviewers for their constructive and insightful comments, which have greatly helped us improve the clarity, rigor, and presentation of the manuscript. Below, we provide a detailed, point-by-point response to each comment. For clarity, reviewer comments are shown in *italic*, and our responses follow each comment.

**1  Response to Reviewer #1**

**1.1  Justification of the model's rationale**

*Reviewer comment: My concern regarding the rationale of the paper has been partially addressed, and I appreciate the analysis on the relative importance of the input features. However, the manuscript still lacks a concrete explanation or experimental support for the mechanism by which the GAN architecture improves performance over simpler or non-generative alternatives. I think my point here will become more clear with the explanation on the need of simpler methods for the comparison.*

**Response:** We sincerely thank the reviewer for this insightful comment. We fully agree that meaningful justification for adopting a deep-learning GAN framework requires explicit comparison with simpler methods to clarify its added value. In response, we have made the following additions and clarifications in the revised manuscript:

**1. Incorporation of MSEM as a baseline method.** We introduced **MSEM (Multi-Source Ensemble Mean)**, a traditional **linear similarity-weighted fusion** method, as a strong and operationally relevant baseline. This method is widely used in multi-model ensemble forecasting systems due to its **simplicity, interpretability, and low computational cost**. Our analyses (Overall Evaluation, Spatial Analysis, Significance Testing, and Case Study) reveal that:

- *MSEM provides stable and cost-effective performance for light to moderate rainfall events*, showing that simple linear fusion approaches can deliver solid results in less complex scenarios.

- *MSEM struggles with heavy rainfall (20–40 mm) and extreme precipitation*, where TS and FSS scores drop noticeably, indicating that linear weighting cannot capture the spatial organization of intense rainfall.

**2. Why GFRNet outperforms MSEM.** While MSEM applies static weights and lacks the ability to model nonlinear relationships, **GFRNet leverages deep feature extraction and a generative framework** to overcome these limitations:

- *Dynamic nonlinear fusion:* GFRNet adaptively integrates multi-source NWP inputs across space and time, learning complex and evolving dependencies beyond what static weighting can achieve.

- *GAN-based structural constraints:* In addition to MSE/MAE content losses, GFRNet uses an adversarial loss to constrain the **precipitation structure distribution**, ensuring that the generated outputs preserve realistic spatial patterns.

- *Superior performance in challenging cases:* GFRNet more accurately reconstructs the core structure of heavy rainfall and avoids spurious "spill-over" precipitation, yielding significantly higher TS and FSS scores than MSEM for extreme rainfall scenarios.

**In summary**, by including MSEM as a strong and cost-effective operational baseline, we demonstrate that GFRNet not only matches the stability of simpler approaches for routine rainfall but also **clearly surpasses them for heavy rainfall and complex precipitation structures**. This comparison provides strong experimental evidence supporting the necessity of a GAN-based deep learning framework in this study.

**1.2  Lack of comparison with simpler methods**

*Reviewer comment: This concern remains unaddressed. In your response, you acknowledge that such comparisons would add value but state that "primary focus of our study is to address the limitations of non-generative models and evaluate the advantages of GAN-based techniques". However, such advantages can only be meaningfully evaluated through comparison*

*with classical alternatives—especially those that might yield similar results with significantly lower complexity and higher interpretability.*

*If the primary contribution lies in deep learning architectures, and precipitation serves primarily as a benchmark, a journal more focused on artificial intelligence might be a more suitable venue.*

*The lack of comparison with simpler methods is not a minor point. For instance, the "blurriness" of NWP model outputs may result, at least in part, from the bilinear interpolation applied. What if elevation or forecast cycle were used to locally adjust the interpolation? What if a local regression approach was used to estimate rainfall as a function of elevation in each subregion? Such strategies might outperform bilinear interpolation without the need for complex machine learning models.*

*Similarly, forecasting methods such as the analogue method—a well-established baseline in meteorology—could be implemented via K-nearest neighbors, even avoiding interpolation altogether. Dimensionality reduction techniques could also be applied to simplify the input space before using such methods. Without at least exploring these simpler alternatives, the comparison is incomplete and the added value of GFRNet remains unclear. Deep learning and AI are undeniably powerful, but I believe they should be applied where their added accuracy justifies the associated cost in complexity and loss of interpretability. In summary, without a fair comparison with classical methods, the paper's core value proposition is not convincingly demonstrated.*

**Response:** We sincerely appreciate the reviewer's thoughtful comment regarding the need for broader comparisons with simpler methods. We fully agree that evaluating the advantages of deep learning models requires comparison with widely used, classical ensemble post-processing techniques. In the revised manuscript, we have clarified the following:

**1. Inclusion of MSEM as a core baseline.** We introduced *MSEM (Multi-Source Ensemble Mean)*, a traditional **linear similarity-weighted fusion** approach, as a strong and representative baseline. MSEM has been widely adopted in operational multi-model forecasting systems because of its simplicity, interpretability, and low computational cost. Our analyses (Overall Evaluation, Spatial Analysis, Significance Testing, and Case Study) show that:

– MSEM performs **consistently well for light and moderate rainfall events**, demonstrating that simple linear fusion can still deliver stable performance in straightforward scenarios;

– However, for **heavy rainfall (20–40 mm) and extreme precipitation**, MSEM's performance drops significantly (TS and FSS decline), and it fails to capture the spatial organization of intense rainfall—highlighting the inherent limitations of static weighting.

**2. Consideration of PMM and why the manuscript focuses on MSEM.** We also tested *Probability Matching Mean (PMM)*, another classical ensemble post-processing method that aligns the cumulative distribution functions (CDFs) of model forecasts before averaging to better preserve precipitation intensity distributions. Tests on the 2024 flood season showed that:

– **PMM and MSEM behave almost identically**: both methods are stable for light and moderate rainfall but show clear degradation for heavy and extreme rainfall(Figure 1), falling short of the deep learning models in restoring precipitation structure.

– Because PMM and MSEM led to the same conclusions, we chose to present only MSEM in the main text to maintain narrative focus and avoid redundancy. We note here that PMM was evaluated, and it does not change the key findings.

**3. Why GFRNet outperforms linear methods like MSEM/PMM.** Unlike MSEM and PMM, which are **static linear methods**, GFRNet leverages deep learning and a generative framework to achieve several key advantages:

– *Dynamic nonlinear fusion:* GFRNet adaptively integrates multi-source NWP inputs across space and time, learning complex, context-dependent relationships beyond static weighting;

– *GAN-based structural constraints:* Beyond content losses (MSE/MAE), GFRNet incorporates an adversarial loss that constrains the precipitation structure distribution, leading to more realistic and coherent rainfall fields;

– *Superior performance for heavy rainfall:* GFRNet reconstructs the core structure of heavy rainfall more accurately, reducing spurious "spill-over" precipitation and achieving significantly higher TS and FSS than MSEM and PMM;

[Figure]

**Figure 1.** Comparison of forecast performance among multiple methods — ECMWF, CMA-SH9, CMA-3KM, PMM, MSEM, FRNet, and GFRNet — for the 2024 flood season, evaluated using Fractions Skill Score (FSS) and Threat Score (TS) at precipitation thresholds of 0.1, 10, 20, and 40 mm/3 h. The results illustrate that while traditional ensemble methods (MSEM, PMM) maintain stable performance for light and moderate rainfall, their skill declines sharply for heavy and extreme rainfall, whereas GFRNet delivers consistently higher FSS and TS scores at higher thresholds, demonstrating its superior capability in reconstructing precipitation structure and intensity.

– *Greater extensibility and future potential:* Deep learning models like GFRNet offer a fundamentally higher ceiling. As more physical variables (e.g., wind, temperature, humidity, geopotential height) are incorporated in future work, traditional statistical methods would struggle to effectively utilize these complex features. GFRNet, by contrast, can learn the nonlinear relationships between these variables and precipitation generation in an end-to-end fashion—paving the way for further improvements in physical consistency and generalization.

**In summary**, we have clarified that PMM was also tested and yielded the same conclusions as MSEM, which is why only MSEM is presented in the main text for conciseness. The comparisons confirm that GFRNet is **at least as stable as linear methods for light-to-moderate rainfall, clearly superior for heavy rainfall and structure restoration, and uniquely positioned to benefit from additional physical variables in the future**—justifying the adoption of a GAN-based deep learning framework in this study.

**1.3 Experimental design and statistical significance**

*Reviewer comment:I remain concerned about the experimental design. Most critically, the manuscript does not include any statistical hypothesis tests to determine whether observed differences are significant or due to random variability. Table 4, for instance, is difficult to interpret without such tests—particularly considering the limited dataset. This should be a central concern for the authors.*

*This issue is exacerbated by the heavy filtering applied to the original dataset. I still do not understand the motivation behind defining "valid samples," nor do I fully understand Figure 2. If samples are filtered based on precipitation criteria, how is it possible that the resulting dataset still contains a mix of valid and invalid samples? Could the authors clarify this?*

*Moreover, I question how Figure 2b demonstrates a fat-tailed distribution. What is the metric or evidence supporting this claim? Based on this assumption, a custom weighted loss function is designed. But this chain of reasoning—and the corresponding design choices—seems to hinge on multiple small decisions that are not fully justified. I am concerned that these*

*steps may unintentionally lead to overfitting—not necessarily through the model itself, but through tuning the data processing to the specific characteristics of the dataset.*

*In this context, I find particularly problematic the statement in line 136: "This increase in the proportion of valid samples improved the stability and efficiency of the model training." This suggests that the validity criteria were defined a posteriori to improve performance, which undermines the generalizability of the results. All these comments should also be understood in the context of a somehow limited dataset.*

**Response:** We sincerely thank the reviewer for these detailed and constructive comments. We fully understand the importance of rigorous experimental design and transparent reporting, and we have addressed these concerns through multiple revisions in the manuscript, as detailed below:

**1. Addition of statistical significance testing.** In Section 3.4 of the revised manuscript, we added a comprehensive **statistical significance analysis**. Using data from the **2022–2024 rainy seasons**, we performed paired t-tests on both the **full-sample set** and the **Top 10% subset**, and marked the statistical levels ($p < 0.05$, $p < 0.01$, $p < 0.001$) directly in the boxplots. These results confirm that GFRNet's performance advantages in most precipitation scenarios are **statistically significant rather than random fluctuations**, while also transparently reporting thresholds (e.g., 0.1 mm) where differences are not significant. This provides a clear statistical foundation for our conclusions.

**2. Clarification and revision of the "valid sample" concept.** We acknowledge that the previous description of "valid samples" caused confusion, and we have **removed the problematic phrasing from the manuscript**. In the revised version, we clarified the sampling logic:

– *Image-level sampling* is applied **only to the training set**, to reduce the large proportion of "no-rain" samples and improve learning efficiency for rainy scenes.

– **validation and test sets remain entirely unsampled and use all original data**, ensuring unbiased evaluation results and avoiding any "a posteriori tuning."

– Even after image-level sampling, rainfall at the **pixel level** remains highly imbalanced, which directly motivated the weighted loss design.

**3. Justification for the long-tail distribution and weighted loss.**

– The **long-tail distribution of rainfall at the pixel scale** is a widely acknowledged fact in meteorological deep learning literature; nearly all relevant studies (Ayzel et al., 2020; Tan et al., 2024; Shi et al., 2015) face this issue and address it with loss-weighting strategies.

– Figure 2b illustrates this imbalance, showing that even after image-level sampling, extreme-rainfall pixels account for less than 0.3% of all pixels.

– Based on this observation, we designed an exponential weighted loss (MSE/MAE + weighted loss). The ablation study in Table B1 validates this design: removing the weighting caused a clear drop in TS for precipitation $\geq$10 mm.

– Furthermore, the loss weighting **did not lead to overfitting**—GFRNet's performance remained stable on the independent 2023–2024 test sets, confirming that the model learned robust and generalizable patterns.

**4. Addressing generalization concerns and dataset size.** We share the reviewer's concern about potential overfitting. To assess generalization, we added **independent flood season data from 2023 and 2024** (not used for training) for evaluation.

– GFRNet demonstrated **stable and consistent performance across 2022–2024**, with TS and FSS scores showing no signs of instability or over-specialization.

– We acknowledge that **further data expansion remains desirable**. Incorporating additional years and broader regional coverage will likely further improve the model's performance and generalization in future work.

**In summary**, the revised manuscript now includes **statistical significance testing**, removes confusing terminology around "valid samples," clarifies the rationale for sampling and weighted loss design, and presents new evidence of GFRNet's robust generalization using independent test sets. These changes ensure that the study's design, methodology, and conclusions are more transparent, rigorous, and reproducible.

**1.4 MS-SSIM vs. RMSE**

*Reviewer comment: Another methodological point that requires further justification is the choice of MS-SSIM over RMSE as a primary evaluation metric. First, given its central role in the analysis, the MS-SSIM metric should be defined mathematically so that readers can judge its appropriateness. Second, the manuscript should offer a clear argument as to why spatial structure should be prioritized over magnitude in evaluating model performance. As shown in Table 5, ECMWF achieves the best RMSE. Without knowing how ECMWF would perform spatially using a better interpolation method (e.g., structure-aware or topography-guided), it is difficult to accept that GFRNet clearly outperforms it overall. The manuscript would benefit from a more explicit discussion of the tradeoffs involved.*

**Response:** We sincerely thank the reviewer for these valuable comments. In the revised manuscript, we have expanded and refined the relevant sections to provide a more comprehensive and transparent explanation, summarized as follows:

**1.Rationale for a multi-metric evaluation framework**

– *Necessity of multiple metrics:* Precipitation forecast evaluation is inherently complex, and no single metric can fully represent model performance. We therefore adopted four complementary categories of metrics: (i) Binary metrics (TS, POD, FAR, BIAS), (ii) Neighborhood metric (FSS), (iii) Continuous metric (RMSE), and (iv) Structural metric (MS-SSIM). This combination enables cross-validation and a more nuanced understanding of model capabilities.

– *Clarification:* We explicitly stated that MS-SSIM is not the "core" evaluation metric of this study but rather a supplementary one, offering a structural perspective alongside TS, FSS, and RMSE.

**2.Clarification of MS-SSIM's role and added value**

– *Established use:* MS-SSIM has been widely adopted in short-range and nowcasting studies (Yin et al., 2021; Tan et al., 2024). We added references to support its relevance.

– *Mathematical definition:* In Section 2.3, we now provide the full mathematical formulation of MS-SSIM, ensuring transparency and reproducibility.

– *Additional insights:* MS-SSIM captures **spatial structure coherence** (e.g., rainfall band continuity, sharpness of edges) that binary scores or RMSE alone cannot. For example, FRNet showed higher TS but lower MS-SSIM, highlighting its blurriness in rainfall structure — an issue GFRNet mitigates effectively.

**3.ECMWF's performance, contribution, and relation to GFRNet**

– *Strengths of ECMWF:* ECMWF remains an internationally trusted NWP system. Our results reaffirm its strong performance for **light and moderate rainfall (0.1–10 mm)** and its skill in predicting synoptic-scale rainfall distribution.

– *Contribution to GFRNet:* Ablation experiments (Appendix B) confirm that ECMWF consistently improves GFRNet's forecasts across all rainfall intensities, proving its foundational importance in our fusion approach.

– *Acknowledging limitations:* Metrics including TS, FSS, case analyses, and significance testing show that ECMWF underestimates **heavy rainfall ($\geq$20 mm)**, where GFRNet adds value.

– *Positioning GFRNet:* GFRNet builds upon ECMWF's solid base, leveraging GAN-based fusion and structural constraints to enhance heavy-rainfall prediction and spatial fidelity.

**In summary**, the revised manuscript clarifies the rationale for a multi-metric framework, properly defines MS-SSIM, and clearly frames ECMWF's indispensable role, while explaining how GFRNet enhances and extends its strengths.

**1.5 Additional minor comments**

*Reviewer comment:I believe my original comment on the case studies still holds: their selection and interpretation could be more clearly justified. Figures 4, 5, and 6 remain difficult to interpret. Improvements in visual clarity and labeling would greatly enhance their value.*

**Response:** We sincerely thank the reviewer for these constructive comments. In the revised manuscript, we have carefully addressed these issues as follows:

1. **Case selection and representativeness**

   – The original submission included three cases from the 2022 rainy season. Following your feedback and after reconsidering the overall balance of the manuscript, we refined this section to focus on **two representative cases**: one from the 2022 rainy season and one from the 2024 rainy season.

   – Both cases are large-scale, high-impact precipitation events, capturing different synoptic settings (e.g., frontal rainfall and precipitation influenced by the subtropical high). This selection provides a more concise yet comprehensive basis for understanding model behavior.

2. **Figures and caption improvements**

   – All case study figures (Figures 4–6) have been fully redesigned: color schemes were refined, legends and axes were clarified, and titles were standardized for consistency and better visual interpretation.

   – **Every figure caption was rewritten and expanded** to provide clearer guidance for interpretation, explicitly highlighting key regions and differences among models, making the figures far easier to read and understand.

3. **Key insights from the two cases**

   – *Distinct NWP characteristics:* The two cases highlight that each NWP model contributes differently: **ECMWF** provides strong skill in large-scale precipitation placement but tends to underestimate heavy rainfall; **CMA-SH9** better captures some mesoscale organized rainfall; and **CMA-3KM** is more sensitive to local convective rainfall but introduces more noise and false alarms.

   – *Role of MSEM:* The MSEM method linearly fuses NWPs through similarity-based weighting, producing a stable and cost-effective baseline. It performs reliably for light and moderate rainfall, but its ability to recover fine details of heavy rainfall remains limited.

   – *GFRNet's strengths and limitations:* GFRNet leverages a GAN-based framework to nonlinearly integrate multi-source NWP information. This allows the model to capture complementary strengths of different NWPs and restore spatial structure for heavy rainfall more effectively. However, GFRNet's strategy is deliberately slightly conservative to avoid overprediction, which can occasionally lead to mild underestimation of extreme rainfall, and its performance still depends on the quality of the NWP inputs.

**In summary**, the revised case study section now presents a more concise and representative selection of events, integrates clearer figures and captions, and delivers deeper insights into the behavior of NWPs, the blending logic of MSEM, and the strengths and limitations of GFRNet, thereby enhancing the overall clarity and value of this section.

**2 Response to Reviewer #2**

**2.1 Model Explainability and Physical Reasoning**

*Reviewer comment: Although the author has verified the model's performance through experiments, the physical mechanism explaining how the model improves precipitation forecasts via the GAN strategy is insufficient. The reviewers pointed out that*

*the impact of terrain and meteorological features on precipitation needs more in-depth discussion, particularly on how the model captures these physical relationships.*

**Response:** We thank the reviewer for highlighting this important point. In the revised manuscript, we made several additions and modifications to address this concern:

1. **Ablation study (Appendix B2 Input Source Contribution Analysis)**

   - We conducted a more systematic set of ablation experiments, assessing not only the removal of different NWP data sources (ECMWF, CMA-SH9, CMA-3KM) but also the effects of removing topographic information and time-encoding features.
   - Based on these experiments, we calculated the **Relative Importance Score (RIS)** for each input feature across different precipitation intensity ranges, quantitatively revealing the contribution of each feature to model predictions.
   - These results provide indirect yet quantitative insights into what GFRNet is "learning" during the training process.

2. **Spatial analysis (Section 3.2)**

   - We expanded the spatial performance analysis, showing that **GFRNet delivers more pronounced improvements in regions with complex terrain**.
   - This indicates that the model effectively leverages terrain-related signals and mitigates systematic biases present in NWP inputs, suggesting the model may be learning terrain-related information explicitly or implicitly.

3. **Physical mechanism discussion (Discussion section)**

   - We elaborated on GFRNet's generative mechanism, explaining how the **adversarial framework drives the model** to generate precipitation fields that better capture realistic rainfall structures—particularly in retaining fine-scale features of intense rainfall.
   - We also emphasized that **GFRNet integrates multi-source NWP inputs** to constrain the generation process, striking a balance between structural realism and physical consistency.

**In addition**, we candidly acknowledge that despite these efforts, it remains challenging to conclusively identify the exact physical laws learned by deep learning models. This limitation is common across AI applications in meteorology, not unique to this study. Enhancing the physical interpretability of such models will be a key focus of our future work.

The relevant additions can be found in **Section 3.2 (Spatial Performance Analysis)**, **Appendix B**, and the **Discussion section**.

**2.2 Comprehensiveness of Experimental Design**

*Reviewer comment: While the author has conducted ablation experiments to verify the role of key components (such as the SE module and weighted loss function), comparisons with other simple models (such as Random Forest or SVM) are lacking, making it difficult to fully prove whether the complexity of the GAN is necessary.*

**Response:** We thank the reviewer for this insightful comment regarding the comprehensiveness of the experimental design. In response, we have made the following revisions and clarifications:

1. **Introduction of MSEM as a baseline model** We incorporated **MSEM**, a linear similarity-based ensemble method that combines multi-source NWP results, as an additional baseline. MSEM is simple, interpretable, and computationally efficient, and does not involve deep learning or adversarial training. This provides a clear reference point for assessing whether the complexity of the GAN framework is warranted.

2. **Inclusion of MSEM comparisons across sections** We systematically added comparisons between GFRNet and MSEM throughout the manuscript, including the Overall Evaluation, Spatial Analysis, Significance Testing, and Case Study sections, to more comprehensively illustrate how GFRNet compares to a straightforward linear fusion approach under various precipitation scenarios.

3. **Findings highlight the added value of the GAN framework** Our analysis shows that MSEM performs robustly in light rainfall and some moderate rainfall cases, but its performance drops notably at the **20–40 mm** threshold and for heavy rainfall, where it cannot capture complex precipitation structures. In contrast, GFRNet's generative adversarial mechanism and nonlinear fusion capability provide clear improvements in moderate-to-heavy rainfall and high-impact precipitation events.

4. **Cost–benefit discussion of the GAN framework** We added discussion on the tradeoff between complexity and benefit. While GFRNet uses adversarial training, the GAN architecture is deliberately simple compared with more complex approaches (e.g., diffusion models or Transformer-based architectures) and is relatively lightweight. Given the clear gains in structural realism and heavy rainfall forecasting skill, we believe this modest complexity is justified and cost-effective.

**In summary**, by adding MSEM as a clear linear baseline, systematically comparing it with GFRNet, and discussing the cost–benefit tradeoff, we demonstrate that the GAN framework is both necessary and practical in the context of this study.

**2.3 Data Size and Generalization Ability**

*Reviewer comment: Using only 4 years of data may limit the model's generalization ability, and the model's performance in other regions or over longer time series has not been verified.*

**Response:** We sincerely thank the reviewer for raising this valuable point regarding the data size and generalization of GFRNet. We agree that generalization is a common challenge faced by deep learning models in meteorological applications.

1. First, to evaluate the generalization ability of GFRNet, we expanded the test period in the revised manuscript by including the 2023 and 2024 rainy seasons as independent tests, forming a continuous three-year test set (2022–2024, with approximately 3,500 samples). Results show that GFRNet's TS, FSS, and MS-SSIM scores remained consistent and steadily superior across the additional two years, particularly in systematic heavy rainfall events, where spatial structure reconstruction and intensity control remained stable. This provides evidence of the model's robustness and, to some extent, demonstrates that GFRNet's methodology has the potential for sustainable and stable application in operational contexts.

2. Second, regarding the scale of the training data, we acknowledge that the model was primarily trained on historical data from 2019–2021, with 2023–2024 data used only for independent testing. Incorporating a longer training record would likely further improve performance. This has been included in our future work plan, and we will progressively expand the dataset to include more years and regions for training and validation, aiming to further enhance the model's generalization and applicability.

3. Finally, while GFRNet demonstrated stable performance across the 2022–2024 rainy seasons and showed encouraging interannual generalization, the training and evaluation data are still largely limited to the most recent five rainy seasons. Compared to longer time spans and broader climatological contexts, this relatively narrow data range may impose some constraints on the model's generalization. In future work, we plan to expand the training and validation datasets to cover longer historical periods and diverse climatic regions, and to explore cross-regional and cross-temporal validation to more systematically assess the model's applicability and robustness.

**In summary**, by expanding the testing period, acknowledging current limitations, and outlining future dataset growth, we present a clearer picture of GFRNet's generalization performance and a concrete plan for strengthening it further.

**2.4 Consistency Analysis of Results**

*Reviewer comment: Some results are contradictory (e.g., GFRNet has a lower TS than FRNet at the 40mm threshold), and the author attributes this to the conservativeness of the GAN but lacks statistical significance analysis.*

**Response:** We sincerely thank the reviewer for this valuable comment on the consistency analysis of results. The reviewer noted that some findings appear contradictory (for example, GFRNet shows a slightly lower TS than FRNet at the 40 mm

threshold) and suggested adding statistical significance tests, as well as discussing the pros and cons of GFRNet's "conserva-tive" strategy.

In response, we have made the following clarifications and additions in the revised manuscript:

1. **Statistical significance analysis.** In Section 3.4, we conducted paired *t*-tests for all precipitation thresholds (0.1, 10, 20, and 40 mm) across both the *all-sample set* and the *Top 10% coverage subset*. Significance levels ($p < 0.05$, 0.01, 0.001) are clearly annotated in Figure 12, ensuring that every reported difference between models is supported by rigorous statistical evidence.

2. **Clarification of TS results at the 40 mm threshold.** In the *all-sample set*, GFRNet's TS is slightly lower than FRNet's, but paired *t*-tests indicate that this difference is **not statistically significant**. In the *Top 10% subset* (representing more organized extreme rainfall events), FRNet's TS is marginally higher than GFRNet's (also without statistical significance). However, this "advantage" comes with a tradeoff: **FRNet is noticeably more aggressive in high-rainfall forecasts, which results in significantly lower FSS scores compared to GFRNet**. This indicates that FRNet has shortcomings in restoring spatial precipitation structures, while GFRNet places greater emphasis on maintaining spatial consistency and coherence.

3. **Rationale and implications of GFRNet's "conservative" strategy.** GFRNet adopts a more balanced and moderately conservative approach to heavy rainfall forecasting. This is not a performance weakness but a deliberate design choice: by avoiding overprediction, GFRNet substantially alleviates FRNet's issue of high BIAS (e.g., BIAS > 1.8 at the 40 mm threshold). This "conservativeness" helps reduce false alarms of extreme rainfall and supports the delivery of forecasts that are more stable and reliable for operational use.

Overall, the revised statistical tests confirm that the reported model differences are robust and statistically well-supported. Moreover, GFRNet strikes a more prudent balance between TS and FSS, ensuring strong predictive accuracy while preserving spatial structure fidelity and reliability for real-world applications.

**2.5 Transparency of Technical Details**

*Reviewer comment: The selection of hyperparameters (e.g., gradient penalty coefficient loss weights a/b) lacks a description of the systematic optimization process, with only grid search mentioned but no results provided.*

**Response:** We thank the reviewer for this valuable comment on the transparency of technical details. The reviewer noted that our description of hyperparameters (e.g., $\lambda$, $a/b$) mentioned grid search but did not provide sufficient explanation of the optimization process or the selection criteria.

In response, we have made the following clarifications and revisions in the manuscript:

1. **Workflow and logic of hyperparameter optimization** For the loss function hyperparameters $a/b$, our goal was to balance model performance for moderate-to-heavy and extreme rainfall. We first performed a limited-range grid search on **FRNet** (the version of the model without GAN) and selected the parameter combination that achieved the best performance on moderate-to-heavy rainfall cases. Once $a/b$ was established, we used FRNet as the **generator** structure, introduced the discriminator component, and began training GFRNet. Because GAN training is inherently less stable, we then tuned the gradient penalty coefficient $\lambda$ by observing the convergence behavior of the model's loss curves and identifying values that ensured a **stable downward trend in the loss function**.

2. **Explanation of the selection criteria in the Methods section** We explicitly state that the choice of $a/b$ was guided by the model's **TS and FSS performance on moderate-to-extreme rainfall thresholds**, while the choice of $\lambda$ was driven by **the stability of GAN training**, ensuring that the generator–discriminator training converged reliably.

3. **Reason for not retaining detailed grid search records** These tuning experiments were designed to quickly narrow down reasonable parameter ranges, so we did not maintain exhaustive records of every parameter combination. However, we list the **final adopted values of $a/b$ and** $\lambda$ in the manuscript and explain the logic behind their selection.

4. **Plans for future improvement** In future work, we plan to adopt more systematic hyperparameter optimization approaches (e.g., Bayesian optimization or evolutionary algorithms) and document the tuning process more comprehensively to further enhance transparency and reproducibility.

Overall, we have **added clarifications in the main Methods section** describing the workflow and decision logic for selecting $a/b$ and $\lambda$, acknowledged the current limitations, and committed to improving hyperparameter tuning documentation in future research.

**2.6 Language and Expression Standardization**

*Reviewer comment: Some language expressions are not professional enough, and there is inconsistency in terminology and citation format, while the readability of charts and tables could be improved.*

**Response:** We sincerely thank the reviewer for this careful observation and constructive suggestion. In the revised manuscript, we have conducted a thorough language review to ensure consistency and improve academic tone. All figure and table captions have been refined for clarity, terminology has been standardized, and several references have been updated and reformatted to align with the journal's style requirements. These revisions have enhanced the overall readability and professionalism of the manuscript.

**2.7 Specificity of Future Work**

*Reviewer comment: The discussion mentions directions like "integration of physical constraints" and "higher resolution," but lacks specific implementation plans.*

**Response:** We thank the reviewer for pointing out that the description of future work in the earlier version was not sufficiently specific. We fully agree that having a clear and actionable research plan is critical for the value of this study and for guiding subsequent work. In response to this comment, we have expanded and refined the *Discussion* section of the revised manuscript to include the following more targeted and technically feasible directions:

1. **Refining the evaluation framework and performance metrics.** We plan to develop a more comprehensive evaluation framework that not only relies on conventional metrics such as TS and FSS but also incorporates measures of spatial structure consistency, rainfall intensity distribution, and application-oriented indices. This will allow us to systematically assess model performance under different precipitation scenarios. Notably, our current study indicates that GFRNet performs better in *systematic rainfall events* (e.g., frontal rainbands, typhoon outer rain) but still has room for improvement in more localized and scattered precipitation. Future metric design will more clearly distinguish between these scenarios, helping to identify the model's strengths and weaknesses and provide feedback for further improvements.

2. **Addressing learning for scattered light rainfall events.** We observed that GANs and other deep generative models tend to exhibit *mode collapse* or neglect when handling *scattered, isolated light rainfall events*. We plan to design targeted data sampling strategies and loss function adjustments, as well as explore data augmentation techniques, to enhance the model's sensitivity and learning stability for these events.

3. **Enhancing the understanding of rainfall generation through physical variables.** Currently, GFRNet still relies to some extent on the precipitation forecasts provided by NWPs. Moving forward, we will introduce additional dynamic and thermodynamic variables (e.g., temperature, humidity, wind fields, geopotential height) as model inputs to help the model directly learn the *physical processes of rainfall generation*. This approach will reduce the "black-box" reliance on NWP rainfall fields and improve the model's *physical consistency* and its ability to generalize to complex meteorological conditions.

4. **Exploring more stable generative model architectures.** We will investigate *Diffusion Models*, *Conditioned Diffusion*, and hybrid *GAN–Diffusion* frameworks. By combining the efficient generation capabilities of GANs with the distribution-learning stability of Diffusion models, we aim to significantly improve the reconstruction of fine-scale rainfall structures and the robustness of predictions for extreme precipitation events.

5. **Improving training stability and data strategies.** GAN training can be sensitive to hyperparameters and data distribution, sometimes leading to instability. We will optimize *sampling strategies*, develop more effective learning schemes for long-tail distributions, and refine *loss function design* while improving training schedules and regularization. These efforts aim to ensure more stable learning across the full spectrum of rainfall intensities and deliver more consistent predictions.

We sincerely appreciate the reviewer's insightful suggestions, which have helped us refine our research roadmap and articulate clearer, more concrete directions for future work.

**References**

Ayzel, G., Heistermann, M., and Winterrath, T.: RainNet v1.0: a convolutional neural network for radar-based precipitation nowcasting, Geoscientific Model Development, 13, 2631–2644, https://doi.org/10.5194/gmd-13-2631-2020, 2020.

Shi, X., Chen, Z., Wang, H., Yeung, D.-Y., kin Wong, W., and chun Woo, W.: Convolutional LSTM Network: A Machine Learning Approach for Precipitation Nowcasting, 2015.

Tan, J., Huang, Q., and Chen, S.: Deep learning model based on multi-scale feature fusion for precipitation nowcasting, Geoscientific Model Development, 17, 53–69, https://doi.org/10.5194/gmd-17-53-2024, 2024.

Yin, J., Gao, Z., and Han, W.: Application of a Radar Echo Extrapolation-Based Deep Learning Method in Strong Convection Nowcasting, Earth and Space Science, 8, e2020EA001 621, https://doi.org/10.1029/2020EA001621, 2021.

---

## Author Response (AR3)

**Response to Reviewers**

We sincerely thank the editor and both reviewers for their careful evaluation of our manuscript and for the constructive comments that helped us further improve the quality and clarity of the paper. Below, we provide our point-by-point responses. Reviewer comments are shown in *italic*, and our responses follow.

**1. Response to Reviewer #1**

**1.1. General evaluation**

We sincerely appreciate the reviewer's positive assessment of our work, including the model design, structural constraint, ablation experiments, and computational efficiency. We are encouraged by this recognition.

**1.2. Code and data availability**

*Code/Data Availability. The appendix declares access to code/data, but authors must confirm their actual functionality (I did not test them, though future users in the field may verify reproducibility*

**Response:** We fully agree with the reviewer on the importance of reproducibility. In this revision, we have taken the following steps to ensure clarity and usability:

1. **Comprehensive README documentation.** The repository now contains a clear and structured `README.md`, including data download instructions, typical training/validation/testing commands, and detailed explanations of each component. This ensures that readers can follow the workflow without ambiguity.

2. **Verified executable workflow.** We uploaded sample test data and verified, in a fresh environment, that the entire workflow—training, validation, and testing—runs smoothly from start to finish using the provided scripts.

3. **Encouraging reproducibility and feedback.** We warmly welcome readers to reproduce our results using the public repository. Should any issues arise, users are invited to open a GitHub issue or contact us via email, and we will be happy to assist.

**1.3. Language and formatting**

*"Language/formatting. The authors should carefully proofread the entire manuscript for grammar and spelling errors. I have already identified a few issues. (e.g., 'contrained' should be 'constrained' in Line 25; 'Table ??' in Line 136)."*

**Response:** We thank the reviewer for pointing this out. We have conducted an additional proofreading pass, correcting spelling mistakes, unifying terminology, refining figure/table references, and improving clarity where needed.

**2. Response to Reviewer #2**

**2.1. Overall evaluation and previous improvements**

We sincerely appreciate the reviewer's detailed assessment and are pleased that the improvements made in the previous revision—including clarifications on sampling strategy, weighted loss motivation, significance testing, MS-SSIM rationale, overall section organization, and case-study interpretation—are now satisfactory.

**2.2. On the remaining concern regarding comparison with simpler classical methods**

*Despite these important improvements, the comparison with simpler and more classical methods remains limited. While MSEM is an appropriate baseline within operational ensemble post-processing, it does not fully represent the family of low-complexity statistical or physically informed approaches that could serve as alternative benchmarks (e.g., locally weighted regression, analogue methods, or elevation-adjusted interpolation). The justification for adopting a GAN-based framework therefore still rests primarily on empirical performance rather than on a clear demonstration that simpler, transparent models would fail under similar conditions. I recognize, however, that the authors have made a sincere and substantial effort to strengthen the paper, and that the present version is technically sound and clearly written. My remaining comment should therefore be understood as a recommendation for future work rather than a condition for acceptance. The manuscript can be accepted after minor editorial polishing. Nevertheless, I would encourage the authors to explicitly acknowledge in the discussion that a more systematic comparison with classical statistical or regression-based methods could further consolidate the argument for using GANs in this context.*

**Response:** We fully agree with the reviewer's observation. We acknowledge that the current study focuses primarily on comparing GFRNet with major NWP models and a representative operational linear ensemble approach (MSEM). The broader family of simple classical methods is not exhaustively explored in this manuscript.

To directly address the reviewer's suggestion, we have added a dedicated paragraph in the **Discussion** section explicitly acknowledging this limitation. In the same paragraph, we state that:

- incorporating a wider set of classical, low-complexity approaches is an important direction for future research; - systematic comparisons in a unified evaluation framework (including TS/FSS/BIAS/MS-SSIM and significance testing) will further strengthen the justification for using a GAN-based approach; - hybrid strategies combining classical methods with deep learning are also promising and will be explored in future work.

We sincerely appreciate this insightful recommendation and have clearly integrated it into the revised manuscript.

**2.3. Editorial polishing**

*Minor language and formatting refinements are recommended.*

**Response:** We have performed additional editorial polishing throughout the manuscript, improving clarity and maintaining consistency in notation, terminology, and formatting.

**Conclusion**

We again express our sincere gratitude to both reviewers for their thoughtful and constructive feedback. The revisions made in response to their comments have strengthened the manuscript substantially. We hope that the current version meets the expectations of the reviewers and the editor.